# Neural Networks for Learning Counterfactual G-Invariances from Single Environments

**S Chandra Mouli**
Department of Computer Science
Purdue University
chandr@purdue.edu

**Bruno Ribeiro**
Department of Computer Science
Purdue University
ribeiro@cs.purdue.edu

## Abstract

Despite —or maybe because of— their astonishing capacity to fit data, neural networks are believed to have difficulties extrapolating beyond training data distribution. This work shows that, for extrapolations based on finite transformation groups, a model's inability to extrapolate is unrelated to its capacity. Rather, the shortcoming is inherited from a learning hypothesis: *Examples not explicitly observed with infinitely many training examples have underspecified outcomes in the learner's model.* In order to endow neural networks with the ability to extrapolate over group transformations, we introduce a learning framework counterfactually-guided by the learning hypothesis that *any group invariance to (known) transformation groups is mandatory even without evidence, unless the learner deems it inconsistent with the training data.* Unlike existing invariance-driven methods for (counterfactual) extrapolations, this framework allows extrapolations from a single environment. Finally, we introduce sequence and image extrapolation tasks that validate our framework and showcase the shortcomings of traditional approaches.

## 1 Introduction

Neural networks are widely praised for their ability to interpolate the training data. However, in some applications, they have also been shown to be unable to learn patterns that can provably extrapolate out-of-distribution (beyond the training data distribution) (Arjovsky et al., 2019; D'Amour et al., 2020; de Haan et al., 2019; Geirhos et al., 2020; McCoy et al., 2019; Schölkopf, 2019).

Recent counterfactual-based learning frameworks for extrapolation tasks —such as ICM and IRM (Arjovsky et al., 2019; Besserve et al., 2018; Johansson et al., 2016; Louizos et al., 2017; Peters et al., 2017; Schölkopf, 2019; Krueger et al., 2020) detailed in Section 2— assume the learner is given data from multiple environmental conditions (say environments E1 and E2) and is expected to learn patterns that work well over an unseen environment E3. In particular, the key idea behind IRM is to force the neural network to learn an internal representation of the input data that is invariant to environmental changes between E1 and E2, and, hence, hopefully also invariant to E3, which may not be true for nonlinear classifiers (Rosenfeld et al., 2020). While successful for a class of extrapolation tasks, these frameworks require multiple environments in the training data. *But, are we asking the impossible? Can humans even perform single-environment extrapolation?*

Young children, unlike monkeys and baboons, assume that a conditional stimulus F given another stimulus D extrapolates to a symmetric relation D given F without ever seeing any such examples (Sidman et al., 1982). E.g., if given D, action F produces a treat, the child assumes that given F, action D also produces a treat. Young children differ from primates in their ability to use symmetries to build conceptual relations beyond visual patterns (Sidman and Tailby, 1982; Westphal-Fitch et al., 2012), allowing extrapolations from intelligent reasoning. However, forcing symmetries against data evidence is undesirable, since symmetries can provide valuable evidence when they are broken.

Unfortunately, single-environment extrapolations have not been addressed in the literature. The challenge comes from a learning framework where *examples not explicitly observed with infinitely many independent training examples are underspecified in the learner's statistical model*, which is shared by both objective (frequentist) and subjective (Bayesian) learner's frameworks. For instance,

consider a supervised learning task where the training data contains infinitely many sequences $x^{(\text{tr})} = (A,B)$ associated with label $y^{(\text{tr})} = C$, but no examples of a sequence $x^{(\text{tr})} = (B,A)$. If given a test example $x^{(\text{te})} = (B,A)$, the hypothesis considers it to be out of distribution and the prediction $P(Y^{(\text{te})} = C | X^{(\text{te})} = (B,A))$ is undefined, since $P(X^{(\text{tr})} = (B,A)) = 0$. This happens regardless of a prior over $P(X^{(\text{tr})})$. This *unseen-is-underspecified* learning hypothesis is not guaranteed to push neural networks to assume symmetric extrapolations without evidence.

**Contributions.** Since symmetries are intrinsically tied to human single-environment extrapolation capabilities, *this work explores a learning framework that modifies the learner's hypothesis space to allow symmetric extrapolation (over known groups) without evidence, while not losing valuable antisymmetric information if observed to predict the target variable in the training data.* Formally, a symmetry is an invariance to transformations of a group, known as a *G-invariance*. In Theorem 1 we show that the counterfactual invariances needed for symmetry extrapolation —denoted Counterfactual G-invariances (CG-invariances)— are stronger than traditional G-invariances. Theorem 2, then, introduces a condition in the structural causal model where G-invariances of linear automorphism groups are safe to use as CG-invariances. With that, Theorem 3 defines a partial order over the appropriate invariant subspaces that we use to learn the correct G-invariances from a single environment without evidence, while retaining the ability to be sensitive to antisymmetries shown to be relevant in the training data. Finally, we introduce sequence and image counterfactual extrapolation tasks with experiments that validate the theoretical results and showcase the advantages of our approach.

## 2 RELATED WORK

**Counterfactual inference and invariances.** Recent efforts have brought counterfactual inference to machine learning models. *Independent causal mechanism (ICM) and Invariant Risk Minimization (IRM)* methods (Arjovsky et al., 2019; Besserve et al., 2018; Johansson et al., 2016; Parascandolo et al., 2018; Schölkopf, 2019), *Causal Discovery from Change (CDC)* methods (Tian and Pearl, 2001), and *representation disentanglement* methods (Bengio et al., 2020; Goudet et al., 2017) broadly look for representations, classifiers, or mechanism descriptions, that are invariant across multiple environments observed in the training data or inferred from the training data (Creager et al., 2020). They rely on multiple environment samples in order to reason over new environments. To the best of our knowledge there is no clear effort for extrapolations from a single environment. The key similarity between the ICM framework and our framework is the assumption of independently sampled mechanisms (the transformations) and causes.

**Domain adaptation and domain generalization.** Domain adaptation and domain generalization (e.g. (Long et al., 2017; Muandet et al., 2013; Quionero-Candela et al., 2009; Rojas-Carulla et al., 2018; Shimodaira, 2000; Zhang et al., 2015) and others) ask questions about specific —observed or known— changes in the data distribution rather than counterfactual questions. A key difference is that counterfactual inference accounts for hypothetical interventions, not known ones.

**Forced G-invariances.** *Forcing a G-invariance may contradict the training data, where the target variable is actually influenced by the transformation of the input.* For instance, handwritten digits are not invariant to $180^o$ rotations, since digits 6 and 9 would get confused. Data augmentation is a type of forced G-invariance (Chen et al., 2020; Lyle et al., 2020) and hence, will fail to extrapolate. Other works forcing G-invariances that will also fail include (not an extensive list): Zaheer et al. (2017) and Murphy et al. (2019a;b) for permutation groups over set and graph inputs; Cohen and Welling (2016), Cohen et al. (2019) for dihedral and spherical transformation groups over images.

**Learning invariances from training data.** The parallel work of Benton et al. (2020) considers learning image invariances from the training data, however does not consider extrapolation tasks. Moreover, it does not provide a concrete theoretical proof of invariance, relying on experimental results over interpolation tasks for validation. Another parallel work (Zhou et al., 2021) uses meta-learning to learn symmetries that are shared across several tasks (or environments). The works of van der Wilk et al. (2018) and Anselmi et al. (2019) focus on learning invariances from training data for better generalization error of the training distribution. However, none of these works consider the extrapolation task. In contrast, our framework formally considers *counterfactual* extrapolation, for which we provide both theoretical and experimental results.

## 3 EXTRAPOLATIONS FROM A SINGLE ENVIRONMENT

Geometrically, extrapolation can be thought as reasoning beyond a convex hull of a set of training points (Haffner, 2002; Hastie et al., 2012; Xu et al., 2021). However, for neural networks —with their arbitrary representation mappings— this geometric interpretation can be insufficient. Rather, we believe extrapolations are better described through counterfactual reasoning (Neyman, 1923; Rubin, 1974; Pearl, 2009; Schölkopf, 2019). Specifically in our task, we ask: *After seeing training data from environment A, the learner wants to extrapolate and predict what would have been the output if the training environment were B.* Extrapolations differ from traditional domain adaptation due to its counterfactual nature —a *what-if* question of an intervention that can only be imagined if given offline data (Bareinboim et al., 2020; Pearl and Mackenzie, 2018), rather than a known distributional change.

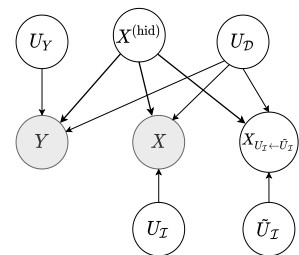

Figure 1: Illustration of our structural causal model (SCM), where gray nodes indicate observed variables (in training). $X$ and $X_{U_\mathcal{I} \leftarrow \widetilde{U}_\mathcal{I}}$ are obtained from $X^{(\mathrm{hid})}$ and are coupled by sharing $U_\mathcal{D}$. However, $U_\mathcal{I}$ and $\widetilde{U}_\mathcal{I}$ can have different support, resulting in different distributions over $X$ and $X_{U_\mathcal{I} \leftarrow \widetilde{U}_\mathcal{I}}$.

Specifically, our framework follows the independent causal mechanism principle (Schölkopf, 2019; Peters et al., 2017): A mechanism describing a variable given its causes is independent of all other mechanisms describing other variables. For instance, in the causal model $U_X \rightarrow X \rightarrow Y \leftarrow U_Y$, this implies that the conditional distribution $P(Y|X)$ is not influenced by any change in $P(X)$.

### 3.1 TRANSFORMATION GROUPS

We focus on extrapolations tied to finite linear automorphism groups acting on the input data. We start with an example. Consider an input $\boldsymbol{x} \in \mathcal{X} = \mathbb{R}^{3n^2}$ representing a vectorized $n \times n$ RGB image. We can define at least three linear automorphism groups: (1) $\mathcal{G}_{\mathrm{rot}} \equiv \{T^{(k)}\}_{k \in \{0°, 90°, 180°, 270°\}}$, which rotates the image by $k$ degrees, (2) $\mathcal{G}_{\mathrm{color}} \equiv \{T^{(\alpha)}\}_{\alpha \in \mathbb{S}_3}$, which permutes the RGB channels of the image, and (3) $\mathcal{G}_{\mathrm{vflip}} \equiv \{T^{(v)}, T^{(0)}\}$, which flips the image vertically. More generally, a linear automorphism group $\mathcal{G}$ satisfies six properties: (automorphism) $\forall T \in \mathcal{G}, T : \mathcal{X} \rightarrow \mathcal{X}$; (identity) $I(x) = x$, $I \in \mathcal{G}$; (is closed under composition) $\forall T, T' \in \mathcal{G}, T \circ T' \in \mathcal{G}$, where $T \circ T'(x) = T(T'(x))$; (associative) $\forall T, T', T^\dagger \in \mathcal{G}, T \circ (T' \circ T^\dagger) = (T \circ T') \circ T^\dagger$; (has inverses) $\forall T \in \mathcal{G}, \exists T^{-1} \in \mathcal{G}$ s.t. $T^{-1} \circ T = I$; and (is linear) $T \in \mathcal{G}$ is a linear function.

Besides images, sequences $\boldsymbol{x} = (x_1, x_2, \ldots)$ are another input of interest, where $\boldsymbol{x} \in \mathcal{X}$ for some appropriately defined set $\mathcal{X}$. Here, the symmetric group (permutation group) $\mathbb{S}_n$, is the set of all permutations $\mathbb{S}_n = \{\pi \mid \pi : \{1, \ldots, n\} \rightarrow \{1, \ldots, n\}$ is a bijection$\}$ equipped with the composition operator. Attributed graphs $(\mathbf{A}, \boldsymbol{X}) \in \mathcal{X}$, where $\mathbf{A}$ is tensor of edge properties and $\boldsymbol{X}$ is a matrix of node attributes, are also of interest for the permutation group $\mathbb{S}_n$.

*Subgroups and overgroups.* Just as we can compose image transformations to make new image transformations, we can also compose automorphism groups into larger automorphism groups (overgroups). For instance, we can compose rotations and image flips to form a linear automorphism group $\mathcal{G}_{\{\mathrm{rot}, \mathrm{vflip}\}} = \langle \mathcal{G}_{\mathrm{rot}} \cup \mathcal{G}_{\mathrm{vflip}} \rangle$ containing all such compositions, where $\langle \cdot \rangle$ is the group join operator. Following standard notation, we say $\mathcal{G}_{\mathrm{rot}} \leq \mathcal{G}_{\{\mathrm{rot}, \mathrm{vflip}\}}$ to indicate that $\mathcal{G}_{\mathrm{rot}}$ is a subgroup of $\mathcal{G}_{\{\mathrm{rot}, \mathrm{vflip}\}}$, or, equivalently, $\mathcal{G}_{\{\mathrm{rot}, \mathrm{vflip}\}}$ is an overgroup of $\mathcal{G}_{\mathrm{rot}}$. Henceforth, we use $\mathcal{G}_{\{1, \ldots, m\}} \equiv \langle \cup_{i=1}^m \mathcal{G}_i \rangle$ to denote the group generated by the groups $\mathcal{G}_1, \ldots, \mathcal{G}_m$.

### 3.2 THE CAUSAL MECHANISM AND AN ECONOMICAL DATA GENERATION PROCESS

We assume that a fundamentally economical process created the training data, where the focus was on sampling diverse environments in a way that mattered to the task. For instance, image datasets will contain mostly upright pictures, rather than images over all possible orientations, but we will assume the dataset curators strive for a somewhat diverse set of subjects for each label (e.g., a good representation of different types of subjects and environmental conditions). *Hence, the absence of variation over image orientations in the dataset can be counted as evidence against its effect on the image labels.*

We describe the data generation with the help of a structural causal model (SCM) (Pearl, 2009, Definition 7.1.1) illustrated in Figure 1. Consider a supervised task over inputs $X$ and their corresponding outputs $Y$, which are random variables defined over a suitable space. The hidden random variable

$$X^{(\text{hid})} := g(U_u), \tag{1}$$

where $g : \mathcal{U} \to \mathcal{X}$ is a measurable map (deterministic function) that describes the input $X$ in some unknown canonical form, where $U_u$ is a random variable (e.g., $U_u \sim \text{Uniform}(0,1)$). Next, we define how $X^{(\text{hid})}$ is modified by transformations into the observed input $X$.

**Transformation of $X^{(\text{hid})}$ into $X$.** Consider a collection of finite linear automorphism groups $\mathcal{G}_1, \dots, \mathcal{G}_m$. Let $\mathcal{I} \subseteq \{1, \dots, m\}$ be a subset and $\mathcal{D} \subseteq \{1, \dots, m\} \backslash \mathcal{I}$ be a subset of its complement. We will later define the target variable to be dependent only on the groups indexed by $\mathcal{D}$. Consider independent and identically distributed random variables $U_\mathcal{D}$ and $U_\mathcal{I}$ that select transformations in the respective overgroups $\mathcal{G}_\mathcal{D} = \langle \cup_{j \in \mathcal{D}} \mathcal{G}_j \rangle$ and $\mathcal{G}_\mathcal{I} = \langle \cup_{i \in \mathcal{I}} \mathcal{G}_i \rangle$. We note in passing that we allow $\mathcal{G}_\mathcal{D} \cap \mathcal{G}_\mathcal{I} \neq \{T_{\text{identity}}\}$ even though $\mathcal{G}_\mathcal{D} \cap \mathcal{G}_\mathcal{I} = \{T_{\text{identity}}\}$ makes the counterfactual task easier. The observed input is defined as

$$X := T_{U_\mathcal{D}, U_\mathcal{I}} \circ X^{(\text{hid})}, \tag{2}$$

where $T_{U_\mathcal{D}, U_\mathcal{I}}$ is a transformation in $\mathcal{G}_{\mathcal{D} \cup \mathcal{I}}$ indexed by two independent *hidden* environment background random variables $U_\mathcal{D}, U_\mathcal{I}$. The reader can roughly interpret $U_\mathcal{D}$ and $U_\mathcal{I}$ as the random seeds of a random number generator that gives ordered sequences of transformations from $\mathcal{G}_\mathcal{D}$ and $\mathcal{G}_\mathcal{I}$ respectively. If these ordered sequences are, say, $T_\mathcal{D}^{(1)}, \dots, T_\mathcal{D}^{(a)}$ and $T_\mathcal{I}^{(1)}, \dots, T_\mathcal{I}^{(b)}$, then $T_{U_\mathcal{D}, U_\mathcal{I}}$ is the transformation obtained after interleaving the two sequences of transformations and composing them in order: $T_{U_\mathcal{D}, U_\mathcal{I}} = T_\mathcal{I}^{(1)} \circ T_\mathcal{D}^{(1)} \circ T_\mathcal{I}^{(2)} \circ \dots$. Note that $T_\mathcal{I}^{(i)}$ or $T_\mathcal{D}^{(i)}$ could be identity transformations. Appendix B.1 shows that this indexing is surjective, i.e., it can index every transformation in $\mathcal{G}_{\mathcal{D} \cup \mathcal{I}}$.

**Target variable.** The output $Y$ associated with $X$ is given by

$$Y := h(X^{(\text{hid})}, U_\mathcal{D}, U_Y), \tag{3}$$

where $h$ is a deterministic function and $U_Y$ is an independent random variable.

A distribution over the set of background random variables $\mathcal{U}_{\text{all}} = \{U_u, U_Y, U_\mathcal{D}, U_\mathcal{I}\}$ along with Equations (2) and (3) *induces* a joint distribution $P(Y, X)$. If the support of $U_\mathcal{I}$ is a singleton set $\{c\}$ for some constant $c$, then $(Y, X)$ are said to be sampled using an **economical data generation process**. In other words, the training data can contain just one value for the variable $U_\mathcal{I}$ since the outputs $Y$ do not depend on $U_\mathcal{I}$. For instance, if $\mathcal{G}_\mathcal{I}$ is the rotation group, and the image label $Y$ does not depend on image rotation, then the observed images can be all upright since the sampling is economical. This is not a required condition for our method to work, however.

**Extrapolation as counterfactual reasoning.** We can now ask "what would have happened to $Y$ if we had given specific values of $U_\mathcal{I}$ to the data generation process in Equations (2) and (3) rather than sampling from $P(U_\mathcal{I})$". For instance, would the class of an image change if we had flipped the image along the vertical axis? Would we re-classify outlier events if we changed the order of events in a stationary time series? These are counterfactual queries over environment background variables $U_\mathcal{I}$.

We now describe the counterfactual variable in our task via variable coupling (Pitman, 1976; Propp and Wilson, 1996), which we believe gives a standard-statistics-friendly description of counterfactual SCMs (Shpitser and Pearl, 2007). The coupling of two independent variables $D_1$ and $D_2$ is a proof technique that creates a random vector $(D_1^\dagger, D_2^\dagger)$, such that $D_i$ and $D_i^\dagger$ have the same marginal distributions, $i = 1, 2$, but makes $D_1^\dagger$ and $D_2^\dagger$ structurally dependent. For instance, consider independent 6-sided and 12-sided dice, denoted $D_1$ and $D_2$ respectively. Let $D_1^\dagger = (U + \epsilon_1) \bmod 6 + 1$ and $D_2^\dagger = (U + \epsilon_2) \bmod 12 + 1$, where $U$ is a 12-sided die roll and $\epsilon_1, \epsilon_2 \in \{0, 1\}$ are two independent coin flips. Then, the tuple $(D_1^\dagger, D_2^\dagger)$ has coupled the variables $D_1$ and $D_2$ via the common random variable $U$.

**Definition 1** (Counterfactual coupling (CFC)). *The counterfactual coupling of the observed data* $(Y, X)$ *is a vector* $(Y, X, X_{U_\mathcal{I} \leftarrow \widetilde{U}_\mathcal{I}})$, *where* $Y = h(X^{(hid)}, U_\mathcal{D}, U_Y)$, $X = T_{U_\mathcal{D}, U_\mathcal{I}} \circ X^{(hid)}$, *and* $X_{U_\mathcal{I} \leftarrow \widetilde{U}_\mathcal{I}} = T_{U_\mathcal{D}, \widetilde{U}_\mathcal{I}} \circ X^{(hid)}$, *for appropriately defined* $U_u, U_\mathcal{D}, U_Y, U_\mathcal{I}, \widetilde{U}_\mathcal{I}$. *The subscript* $U_\mathcal{I} \leftarrow \widetilde{U}_\mathcal{I}$ *denotes the counterfactual variable to $X$ when* $\widetilde{U}_\mathcal{I}$ *replaces* $U_\mathcal{I}$ *in the data generation process. For a constant $u$,* $X_{U_\mathcal{I} \leftarrow u}$ *gives the same definition as the twin network method of Balke and Pearl (1994).*

The support of $\widetilde{U}_{\mathcal{I}}$ in Definition 1 can be very different from that of $U_{\mathcal{I}}$, potentially inducing a different distribution over $X_{U_{\mathcal{I}} \leftarrow \widetilde{U}_{\mathcal{I}}}$ than $X$ even if the variables $X_{U_{\mathcal{I}} \leftarrow \widetilde{U}_{\mathcal{I}}}$ and $X$ are structurally dependent via $U_{\mathcal{D}}$. Armed with Definition 1, we are now ready to describe our task.

## 3.3 EXTRAPOLATION MODEL

We start by defining counterfactual G-invariant (CG-invariant) representations.

**Definition 2** (CG-invariant representations)**.** *Let the vector* $(X, X_{U_{\mathcal{I}} \leftarrow \widetilde{U}_{\mathcal{I}}})$ *denote the counterfactual coupling of the random variable* $X$ *given in Definition 1 for any* $\widetilde{U}_{\mathcal{I}}$*. A representation function* $\Gamma : \mathcal{X} \rightarrow \mathbb{R}^d$*,* $d \geq 1$*, is deemed CG-invariant if*

$$\Gamma(X) = \Gamma(X_{U_{\mathcal{I}} \leftarrow \widetilde{U}_{\mathcal{I}}}), \tag{4}$$

*where the equality implies that* $\Gamma(X_{U_{\mathcal{I}} \leftarrow u}) = \Gamma(X_{U_{\mathcal{I}} \leftarrow u'}), \forall u \in supp(U_{\mathcal{I}}), \forall u' \in supp(\widetilde{U}_{\mathcal{I}})$ *and* $supp(A)$ *is the support of random variable* $A$*.*

**Extrapolated model from training to test data.** Let $(Y, X^{(\text{tr})}) \sim P(Y, X)$ and $(Y, X^{(\text{te})}) \sim P(Y, X_{U_{\mathcal{I}} \leftarrow \widetilde{U}_{\mathcal{I}}})$, for some appropriately defined $\widetilde{U}_{\mathcal{I}} \sim P(\widetilde{U}_{\mathcal{I}})$, be the random variables describing the training and test data, respectively. We do not have access to test data at training time. Let $\Gamma_{\text{true}} : \mathcal{X} \rightarrow \mathbb{R}^d$, $d \geq 1$, be a representation of the input data. Consider a function $g_{\text{true}} : \mathbb{R}^d \rightarrow \text{Im} \, P(Y = y|X^{(\text{tr})})$ —where $\text{Im} \, P(\cdot)$ is the image of $P(\cdot)$— (e.g., $g_{\text{true}}$ could be a feedforward network with softmax output) and

$$Y|X^{(\text{tr})} \stackrel{d}{=} \hat{Y}|X^{(\text{tr})}, \text{ with } \hat{Y}|X^{(\text{tr})} \sim g_{\text{true}}(\Gamma_{\text{true}}(X^{(\text{tr})})), \tag{5}$$

where $\stackrel{d}{=}$ means the random variables have the same distribution. Then, **if** $\Gamma_{\text{true}}(X) = \Gamma_{\text{true}}(X_{U_{\mathcal{I}} \leftarrow \widetilde{U}_{\mathcal{I}}})$, **then** we have that, by our definition of $X^{(\text{te})}$ and $X^{(\text{tr})}$, $g_{\text{true}} \circ \Gamma_{\text{true}}$ extrapolates:

$$Y|X^{(\text{te})} \stackrel{d}{=} \hat{Y}|X^{(\text{te})}, \text{ with } \hat{Y}|X^{(\text{te})} \sim g_{\text{true}}(\Gamma_{\text{true}}(X^{(\text{te})})). \tag{6}$$

Alas, learning $\Gamma_{\text{true}}$ is the real challenge: (i) We do not know $\mathcal{I}$ (and, hence, we do not know the group $\mathcal{G}_{\mathcal{I}}$ which is related to the CG-invariance); (ii) this would also require knowing $P(\widetilde{U}_{\mathcal{I}})$, which we don't. Without an observed $X_{U_{\mathcal{I}} \leftarrow \widetilde{U}_{\mathcal{I}}}$, the statistical assumption that *examples not explicitly observed with infinitely large training data have underspecified outcomes in the learner's statistical model* does not push the model towards learning $\Gamma_{\text{true}}$. We must change this assumption.

## 4 CG-INVARIANCES FOR EXTRAPOLATION

In this section we introduce our learning framework, which seeks to use the training data to approximate $\Gamma_{\text{true}}$ and $g_{\text{true}}$ of Equation (6). Our framework regularizes neural network weights towards representations that are invariant to groups that negligibly impact training data accuracy. We overcome some key challenges: **(a)** Theorem 1 below shows that CG-invariances (Definition 2) are stronger than G-invariances. After that, Theorem 2 defines conditions under which G-invariances suffice as CG-invariances, and **(b)** We derive an optimization objective where *all G-invariances are mandatory, except the ones deemed inconsistent with the training data*, replacing the traditional *unseen-is-underspecified* learning hypothesis.

Our first question is whether CG-invariances are just G-invariances. Theorem 1 shows they are not.

**Theorem 1** (CG-invariance is stronger than G-invariance)**.** *Let the vector* $(X, X_{U_{\mathcal{I}} \leftarrow \widetilde{U}_{\mathcal{I}}})$ *denote the counterfactual coupling of the observed variable* $X$ *given in Definition 1. For a representation* $\Gamma : \mathcal{X} \rightarrow \mathbb{R}^d$*,* $d \geq 1$*, let*

$$G\text{-}inv : \ \forall T_{\mathcal{I}} \in \mathcal{G}_{\mathcal{I}}, \ \ \Gamma(X) = \Gamma(T_{\mathcal{I}} \circ X),$$
$$CG\text{-}inv : \ \Gamma(X) = \Gamma(X_{U_{\mathcal{I}} \leftarrow \widetilde{U}_{\mathcal{I}}}),$$

*denote the conditions on* $\Gamma$ *for* $\mathcal{G}_{\mathcal{I}}$*-invariance and CG-invariance respectively. Then, CG-inv* $\Longrightarrow$ *G-inv, but G-inv* $\not\Longrightarrow$ *CG-inv.*

The proof in Appendix B.2 constructs a task over images and a representation $\Gamma$ that is $\mathcal{G}_{\mathcal{I}}$-invariant but is not CG-invariant (for appropriately chosen $\mathcal{G}_{\mathcal{I}}$ and $\mathcal{G}_{\mathcal{D}}$). The following condition ensures that a $\mathcal{G}_{\mathcal{I}}$-invariance is also a CG-invariance.

**Theorem 2.** *If $\mathcal{G}_\mathcal{I}$ is a normal subgroup of $\mathcal{G}_{\mathcal{D}\cup\mathcal{I}}$, then CG-inv $\iff$ G-inv.*

A subgroup $H$ of a group $G$ is called normal (denoted $H \trianglelefteq G$) if for all $h \in H$ and $g \in G$, $ghg^{-1} \in H$. Proof in the Appendix B.2 utilizes the fact that if $\mathcal{G}_\mathcal{I} \trianglelefteq \mathcal{G}_{\mathcal{D}\cup\mathcal{I}}$, then any $T \in \mathcal{G}_{\mathcal{D}\cup\mathcal{I}}$ can be written as $T = T_\mathcal{I} \circ T_\mathcal{D}$ for some $T_\mathcal{I} \in \mathcal{G}_\mathcal{I}, T_\mathcal{D} \in \mathcal{G}_\mathcal{D}$. Throughout the rest of the paper, we will assume that $\mathcal{G}_\mathcal{I}$ is a normal subgroup of $\mathcal{G}_{\mathcal{D}\cup\mathcal{I}}$ in the SCM Equation (2).

## 4.1 CONSTRUCTING SUBSPACES OF VEC($\mathcal{X}$) PARTIALLY ORDERED BY INVARIANCE STRENGTH

As discussed before, we do not know $\mathcal{G}_\mathcal{I}$. In this subsection, we build neural network weights that are invariant to $\mathcal{G}_M$ for different subsets $M \subseteq \{1, \ldots, m\}$. A detailed step-by-step example of this construction for $3 \times 3$ images is shown in Appendix C. We start by restating the Reynolds operator, which has been extensively used in the literature of G-invariant representations without attribution:

**Lemma 1** (Reynolds operator (Mumford et al. (1994), Definition 1.5))**.** *Let $\mathcal{G}$ be a (finite) linear automorphism group over $vec(\mathcal{X})$. Then,*

$$\overline{T} = \frac{1}{|\mathcal{G}|} \sum_{T \in \mathcal{G}} T \tag{7}$$

*is a $\mathcal{G}$-invariant linear automorphism, i.e., $\forall T_\dagger \in \mathcal{G}$ and $\forall \boldsymbol{x} \in vec(\mathcal{X})$, it must be that $\overline{T}(T_\dagger \boldsymbol{x}) = \overline{T}\boldsymbol{x}$.*

Since $\overline{T}$ is a projection operator (i.e., $\overline{T}^2 = \overline{T}$), all the eigenvalues of $\overline{T}$ are either 0 or 1. Using this fact, we now describe G-invariant neurons using the left eigenspace of $\overline{T}$ corresponding to the eigenvalue 1.

**Lemma 2.** *If $\mathcal{W}$ denotes the left eigenspace corresponding to the eigenvalue 1 of the Reynolds operator $\overline{T}$ for the group $\mathcal{G}$, **then** $\forall b \in \mathbb{R}$, the linear transformation $\gamma(\boldsymbol{x}; \boldsymbol{w}, b) = \boldsymbol{w}^T \boldsymbol{x} + b$ is invariant to all transformations $T \in \mathcal{G}$, i.e., $\gamma(T\boldsymbol{x}; \boldsymbol{w}, b) = \gamma(\boldsymbol{x}; \boldsymbol{w}, b)$, **if and only if** $\boldsymbol{w} \in \mathcal{W}$.*

The above property of the Reynolds operator can be leveraged to build neural networks that adhere to particular group symmetries, as done by Yarotsky (2018) and van der Pol et al. (2020). If we knew $\mathcal{G}_\mathcal{I}$, restricting the parameters of each neuron to the left 1-eigenspace of the Reynolds operator of $\mathcal{G}_\mathcal{I}$ would give us a way to build a $\mathcal{G}_\mathcal{I}$-invariant neural network.

Alas, we do not know $\mathcal{I}$, and consequently we do not know $\mathcal{G}_\mathcal{I}$. Instead, we want to construct bases for the complete space vec($\mathcal{X}$) such that they are partially ordered by their invariance strength: From most invariant bases to least. In other words, we construct bases for subspaces $\mathcal{B}_M$ for $M \subseteq \{1, \ldots, m\}$ such that any weight vector $\mathbf{w} \in \mathcal{B}_M$ is **(a)** invariant to the groups $\mathcal{G}_i$ for $i \in M$, and **(b)** not invariant to any group $\mathcal{G}_j$ for $j \in \{1, \ldots, m\} \setminus M$. Later, we will use this partial order to define a regularization term for our method. Theorem 3 shows how these bases can be constructed inductively, where we start with the most invariant subspace (when $M = \{1, \ldots, m\}$) and judiciously work our way over increasingly less invariant subspaces. A reader more interested in the algorithm can first refer to the pseudocode in Appendix D or the example in Appendix C (Step 2).

**Theorem 3** (G-invariant subspace bases can be partially ordered by invariance strength)**.** *Let $\mathcal{W}_i \subseteq vec(\mathcal{X})$ be the left eigenspace corresponding to the eigenvalue 1 of the Reynolds operator $\overline{T}_i$ for group $\mathcal{G}_i$, $i = 1, \ldots, m$. We construct the invariant subspace partitions*

$$\widetilde{\mathcal{B}}_M = \bigcap_{i \in M} \mathcal{W}_i \, ; \quad \mathcal{B}_M = orth_{\mathcal{B}_{\supsetneq M}}(\widetilde{\mathcal{B}}_M), \quad \forall M \in \wp(\{1, \ldots, m\}) \setminus \emptyset, \tag{8}$$

*where $\wp$ is the power set, $\mathcal{B}_{\supsetneq M} = \bigoplus_{N \supsetneq M} \mathcal{B}_N$, $orth_{\mathcal{A}_1}(\mathcal{A}_2)$ removes from the subspace $\mathcal{A}_2$ its orthogonal projection onto the subspace $\mathcal{A}_1$, and $\bigoplus$ is the direct sum operator. **Then**, the linear transformation $\gamma(\boldsymbol{x}; \boldsymbol{w}, b) = \boldsymbol{w}^T \boldsymbol{x} + b$, $b \in \mathbb{R}$, $\forall \boldsymbol{w} \in \mathcal{B}_M \setminus \{\boldsymbol{0}\}$, is $\mathcal{G}_M$-invariant but not $\mathcal{G}_j$-invariant $\forall j \in \{1, \ldots, m\} \setminus M$.*

The proof in Appendix B.3 shows that $\widetilde{\mathcal{B}}_M$ contains all the vectors $\boldsymbol{w}$ that are invariant to $\mathcal{G}_M$ but could also contain vectors that are invariant to some overgroup of $\mathcal{G}_M$. Thus, each step of our inductive method performs a Gram-Schmidt orthogonalization in order to satisfy condition **(b)** above: we need to remove from $\widetilde{\mathcal{B}}_M$ all weight vectors that are invariant to more groups in addition to those indexed by $M$ (i.e., supersets of $M$). In addition, if needed, we obtain the basis for the rest of the space through $\mathcal{B}_\emptyset = orth_{\mathcal{B}_{\supsetneq\emptyset}}(vec(\mathcal{X}))$, the orthogonal complement of $\mathcal{B}_{\supsetneq\emptyset}$.

Note that if $\boldsymbol{w} \in \mathcal{B}_N$, then $\boldsymbol{w}$ is never $\mathcal{G}_H$-invariant for $H \supsetneq N$ as we remove all such $\boldsymbol{w}$ from $\mathcal{B}_N$. Hence, the partial order of nested subsets in $\wp(\{1, \ldots, m\})$ induces a partial order of invariance strengths in the bases of the input domain $\mathrm{vec}(\mathcal{X})$ (see Figure 5 for an example). We define level of invariance (or invariance strength) of a subspace $\mathcal{B}_M$ as the size of $M$ (i.e., $|M|$).

*Practical aspects.* Our algorithm should output $d_{\mathcal{X}} = \dim(\mathrm{vec}(\mathcal{X}))$ basis vectors covering the entire space (i.e., our new neuron, described later in Equation (10), still has $d_{\mathcal{X}} + 1$ parameters as the original one). Thus we stop the algorithm in Theorem 3 once $d_{\mathcal{X}}$ basis vectors are found. Moreover, the algorithm needs to run only once for groups $\mathcal{G}_1, \ldots, \mathcal{G}_m$, and the results can be reused for other neural architectures. While the worst-case runtime of finding the bases could be exponential in $m$, it is unclear whether this exponential runtime can actually happen in practice (all of our experimental runtimes take less than one minute in commodity machines).

## 4.2   Learning CG-invariant representations without knowledge of $\mathcal{G}_{\mathcal{I}}$.

We are now ready to learn a CG-invariant representation using neural networks $\Gamma$ and $g$. Let $\mathcal{G}_1, \ldots, \mathcal{G}_m$ be known linear automorphism groups. Under the assumption of Theorem 2, we just need $\Gamma$ to be $\mathcal{G}_{\mathcal{I}}$-invariant, with $\mathcal{G}_{\mathcal{I}} \equiv \langle \cup_{i \in \mathcal{I}} \mathcal{G}_i \rangle$, but $\mathcal{I} \subseteq \{1, \ldots, m\}$ is unknown to us. We achieve the correct $\mathcal{G}_{\mathcal{I}}$-invariance by redefining the neuron weights of $\Gamma$ using the subspaces of Theorem 3 and proposing a regularized objective that pushes $\Gamma$ towards the strongest overgroup G-invariance that does not *significantly* hurt the training data, where *significantly* is controlled by a regularization strength $\lambda > 0$.

More formally, let $\Gamma : \mathrm{vec}(\mathcal{X}) \times \mathbb{R}^{d_{\mathcal{X}} \times H} \times \mathbb{R}^H \to \mathbb{R}^d$, $H \geq 1, d \geq 1$, be a neural network layer with $H$ neurons, parameterized by free parameters $\boldsymbol{\Omega} \in \mathbb{R}^{d_{\mathcal{X}} \times H}$ and $\boldsymbol{b} \in \mathbb{R}^H$. The $H$ neurons are arranged in an appropriate architecture as described in Section 5, but reader can imagine a feedforward layer for now. Let $g : \mathbb{R}^d \to \mathrm{Im}P(Y|X)$ be a link function. The training data $\mathcal{D}^{(\mathrm{tr})} = \{(y_i^{(\mathrm{tr})}, \boldsymbol{x}_i^{(\mathrm{tr})})\}_{i=1}^N$ is assumed to be sampled according to the SCM data generation process in Equations (1) to (3), with the hidden $\mathcal{G}_{\mathcal{I}}$ satisfying the conditions in Theorem 2.

Let $\boldsymbol{B}_M \in \mathbb{R}^{d_{\mathcal{X}} \times d_M}$ be a matrix whose columns are the orthogonal basis of subspace $\mathcal{B}_M \neq \{\mathbf{0}\}$ (from Theorem 3) with dimension $d_M$. Any vector $\boldsymbol{w} \in \mathcal{B}_M$ can be expressed as a linear combination of these basis columns. The coefficients of the linear combination form our learnable parameters. These neuron weights $\boldsymbol{\Omega}$ have a correspondence to the nonzero subspace bases $\boldsymbol{B}_{M_1}, \boldsymbol{B}_{M_2}, \ldots, \boldsymbol{B}_{M_B}$:

$$\boldsymbol{\Omega} = \begin{bmatrix} \boldsymbol{\omega}_{M_1,1} & \cdots & \boldsymbol{\omega}_{M_1,H} \\ \cdots & \cdots & \vdots \\ \boldsymbol{\omega}_{M_B,1} & \cdots & \boldsymbol{\omega}_{M_B,H} \end{bmatrix}, \quad \text{where } B \leq d_{\mathcal{X}}, \tag{9}$$

and $\boldsymbol{\omega}_{M_i,h} \in \mathbb{R}^{d_{M_i} \times 1}$ represents the learnable parameters for the subspace $\mathcal{B}_{M_i}$ and the $h$-th neuron. The $h$-th neuron in $\Gamma$, $h \in \{1, \ldots, H\}$, has the form

$$\Gamma^{(h)}(\boldsymbol{x}) = \sigma\left( \boldsymbol{x}^\top \left( \sum_{i=1}^B \boldsymbol{B}_{M_i} \boldsymbol{\omega}_{M_i,h} \right) + b_h \right), \tag{10}$$

$\sigma(\cdot)$ is a nonpolynomial activation function, and $b_h \in \mathbb{R}$ is a bias parameter. Our optimization objective is then

$$\widehat{\boldsymbol{\Omega}}, \widehat{\boldsymbol{b}}, \widehat{\boldsymbol{W}}_g = \underset{\boldsymbol{\Omega}, \boldsymbol{b}, \boldsymbol{W}_g}{\arg\min} \sum_{(y^{(\mathrm{tr})}, \boldsymbol{x}^{(\mathrm{tr})}) \in \mathcal{D}^{(\mathrm{tr})}} \mathcal{L}\left( y^{(\mathrm{tr})}, g(\Gamma(\boldsymbol{x}^{(\mathrm{tr})}; \boldsymbol{\Omega}, \boldsymbol{b}); \boldsymbol{W}_g) \right) + \lambda R(\boldsymbol{\Omega}) \tag{11}$$

where $\mathcal{L} : \mathcal{Y} \times \mathrm{Im}P(Y|X) \to \mathbb{R}_{\geq 0}$ is a nonnegative loss function, and $\lambda > 0$ is a regularization strength. The regularization penalty $R(\boldsymbol{\Omega})$ is given by,

$$R(\boldsymbol{\Omega}) = |\{M_i : |M_i| > l, \ 1 \leq i \leq B\}| + \sum_{\substack{i:|M_i|=l \\ 1 \leq i \leq B}} \mathbf{1}\{\|\boldsymbol{\omega}_{M_i,\cdot}\|_2^2 > 0\}, \tag{12}$$

where $l = \min\{|M_i| \cdot \mathbf{1}\{\|\boldsymbol{\omega}_{M_i,\cdot}\|_2^2 > 0\}, \ 1 \leq i \leq B\}$.

*Intuition behind the penalty in Equation* (12): A subspace $\mathcal{B}_{M_i}$ is said to be *used* in the computation of neuron $h$ (Equation (10)) if the corresponding parameter $\boldsymbol{\omega}_{M_i,h}$ is nonzero. Then, let $\mathcal{B}_{M_k}$ be the least invariant subspace used by any neuron (i.e., $|M_k|$ is the lowest among all used subspaces) and $|M_k| = l$. The **first term** in the penalty counts the number of subspaces $\mathcal{B}_{M_i}$ (used or unused) that

are invariant to more groups than $\mathcal{B}_{M_k}$ (i.e., $|M_i| > |M_k|$). This term ensures that the optimization tries to use subspaces that are higher in the partial order with invariance to more groups. The **second term** in the penalty counts the number of subspaces $\mathcal{B}_{M_i}$ that have the same level of invariance as $\mathcal{B}_{M_k}$ (i.e., $|M_i| = |M_k|$), and also have the corresponding coefficients $\boldsymbol{\omega}_{M_i,h}$ nonzero (i.e., the subspace $\mathcal{B}_{M_i}$ is used). The larger the second term, farther away the optimization is from increasing the least level of invariance from $l$ to $l + 1$. We present a differentiable approximation of the penalty in Appendix F along with an example computation of Equation (12) in Figure 8.

*Limitations of Equation* (12): Recall that we stop the algorithm in Theorem 3 once the basis for $\mathrm{vec}(\mathcal{X})$ is found. In such cases, there could be parameters $\boldsymbol{\Omega}'$ and $\boldsymbol{\Omega}''$ that assign positive weights corresponding to the same subspace bases, but with $\boldsymbol{\Omega}'$ invariant to more groups than $\boldsymbol{\Omega}''$. The penalty in Equation (12) however cannot distinguish between these two sets of weights as they *use* the same subspaces and thus, $R(\boldsymbol{\Omega}') = R(\boldsymbol{\Omega}'')$. We provide an example in the case of sequence inputs in Appendix F.3 and leave the solution as future work.

*Selecting regularization strength* $\lambda$: We use a held-out training set to find the best validation accuracy achieved by any value of $\lambda$. Then, among all the values of $\lambda$ that achieve validation accuracy within 5% of the best validation accuracy, we choose the largest $\lambda$ (i.e., we opt for maximum invariance without significantly affecting validation performance).

## 5 CG-INVARIANT NEURAL ARCHITECTURES

For **image tasks**: We can apply the CG-regularization of Equations (10) and (11) in the convolutional layers of a CNN architecture like VGG (Simonyan and Zisserman, 2014). Mostly, the VGG architecture remains the same with the exception that the convolutional filters are obtained using the subspaces from Theorem 3 for the given groups. Once the filter is obtained as a linear combination of the bases, it is convolved with the image or the feature maps. This will ensure that the model is CG-invariant to the transformations of smaller patches in the image. A sum-pooling layer over the entire channel is applied after all the convolutional layers to ensure that the model can be CG-invariant to the transformations on the whole image. See Appendix E.1 for an example architecture.

For **sequence and array tasks** (sets, graph & tensor tasks), the architecture is more direct: One can simply apply a feedforward network with as many hidden layers as needed. Each neuron of the first layer is as given by Equation (10), ensuring that the first layer can be CG-invariant to the given groups if needed. Other layers can have regular neurons since stacking dense layers after a CG-invariant layer does not undo the CG-invariance. See Appendix E.2 for an example architecture.

## 6 EMPIRICAL RESULTS

We now provide empirical results of 12 different tasks to showcase the properties and advantages of our framework [1]. Due to space limitations, our results are only briefly summarized here, with most of the details described in Appendix G. Appendix A also shows a task where CG-invariance is stronger than G-invariance, showing the practical relevance of Theorem 1.

**Validation of our learning framework (CGreg):** In 12 different image and sequence tasks, we confirmed that our CG-regularization of Equation (11) is able to selectively learn to be invariant to the largest overgroup that doesn't contradict the training data, all of this *without* any evidence in the data supporting the invariance. The results are summarized in Table 1, which also shows that both standard neural networks and forced G-invariant networks do not extrapolate to new environments when $\mathcal{I} \neq \emptyset$ and $\mathcal{I} \subsetneq \{1, \ldots, m\}$, respectively.

$X^{(\mathbf{hid})}$ **and Transformation groups:** $X^{(\mathrm{hid})}$ is the canonically ordered input (e.g., upright images, sorted sequences). Our task considers $m$ linear automorphism groups $\mathcal{G}_1, \ldots, \mathcal{G}_m$. We generate $\mathcal{G}_\mathcal{I}$ from a subset $\mathcal{I} \subseteq \{1, \ldots, m\}$ of the groups, i.e., $\mathcal{G}_\mathcal{I} = \langle \mathcal{G}_{i \in \mathcal{I}} \rangle$. We construct $\mathcal{G}_\mathcal{D}$ using a subset of $\{1, \ldots, m\} \setminus \mathcal{I}$, while ensuring that $\mathcal{G}_\mathcal{I} \trianglelefteq \mathcal{G}_{\mathcal{D} \cup \mathcal{I}}$ in order to fulfill the conditions in Theorem 2.

For *image* tasks, $X^{(\mathrm{hid})}$ is an upright MNIST image and the $m = 3$ groups are $\mathcal{G}_{\mathrm{rot}}, \mathcal{G}_{\mathrm{color}}, \mathcal{G}_{\mathrm{vertical\text{-}flip}}$. For *sequence* tasks, we sample $X^{(\mathrm{hid})}$ as a sequence of $n$ *sorted* integers from a fixed vocabulary and consider $m = \binom{n}{2}$ permutation groups for all the pair-wise permutations: $\mathcal{G}_{1,2}, \mathcal{G}_{2,3}, \mathcal{G}_{1,3}, \ldots, \mathcal{G}_{n-1,n}$, where $\mathcal{G}_{i,j} := \{T_{\mathrm{identity}}, T_{i,j}\}$ and $T_{i,j}$ swaps positions $i$ and $j$ in the sequence.

---

[1]Public code available at: `https://github.com/PurdueMINDS/NN_CGInvariance`

Table 1: Extrapolation accuracy ($\pm$ 95% confidence interval, **bold** means $p < 0.05$ significant)

| | Image transformation groups $\{\mathcal{G}_{\text{rot}}, \mathcal{G}_{\text{vertical-flip}}, \mathcal{G}_{\text{color}}\}$ | | | | | | Sequences $\{\mathcal{G}_{1,2}, \ldots, \mathcal{G}_{n-1,n}\}$ | | | |
| | **Task: Predict digit & which transformations of $\mathcal{G}_{\mathcal{D}}$ was applied to image** | | | | | | **Tasks depend on $\mathcal{I}$ (see Appendix G)** | | | |
| | MNIST $\{3,4\}$ images | | | MNIST all images | | | | Sequence Tasks | | |
| $\mathcal{I}$ | VGG | +G-inv | +CGreg | VGG | +G-inv | +CGreg | $\mathcal{I}$ | Transformer | Best FF+G-inv | FF+CGreg |
| $\emptyset$ | **96.06±0.63** | 15.96±2.17 | 94.49±01.49 | 89.35±0.52 | 15.64±1.55 | **90.89±0.93** | $\emptyset$ | **100.00±0.00** | 23.38±1.88 | 95.70±03.05 |
| color | 15.06±6.70 | 50.05±2.17 | **94.16±06.43** | 4.51±1.36 | 47.61±0.45 | **88.69±2.11** | $\{(i,i+2k)\}_{i,k}$ | 0.85±0.37 | 0.97±0.60 | **71.85±26.61** |
| rot,vflip | 54.87±0.90 | 32.05±1.16 | **95.78±07.11** | 25.91±0.95 | 44.41±3.28 | **62.68±6.02** | $\{(i,j)\}_{j>i\geq 2}$ | 12.15±16.05 | 10.68±1.49 | **42.08±18.99** |
| rot,col,vflip | 49.52±2.37 | **97.19±1.02** | 94.89±07.49 | 11.27±0.34 | **68.46±2.83** | 64.99±2.76 | $\{(i,j)\}_{j>i\geq 1}$ | 20.26±32.08 | **100.00±0.00** | **100.00±00.00** |

**Training data:** The training data is sampled via the SCM equations using an economical data generation process. We decompose the transformation $T_{U_{\mathcal{I}}, U_{\mathcal{D}}}$ into a transformation $T_{U_{\mathcal{D}}} \in \mathcal{G}_{\mathcal{D}}$ followed by another transformation $T_{U'_{\mathcal{I}}|U_{\mathcal{D}}} \in \mathcal{G}_{\mathcal{I}}$ to obtain $X = T_{U'_{\mathcal{I}}|U_{\mathcal{D}}} \circ T_{U_{\mathcal{D}}} \circ X^{(\text{hid})}$. This decomposition is made possible from our assumption that $\mathcal{G}_{\mathcal{I}}$ is a normal subgroup of $\mathcal{G}_{\mathcal{D}\cup\mathcal{I}}$ (Theorem 2). Under the assumption of economic sampling of the training data, in all our experiments we simply set $T_{U'_{\mathcal{I}}|U_{\mathcal{D}}} = T_{\text{identity}} \in \mathcal{G}_{\mathcal{I}}$, whereas $T_{U_{\mathcal{D}}}$ is randomly sampled from $\mathcal{G}_{\mathcal{D}}$. Finally, following Equation (3), the label $Y$ is a combination of the original label of $X^{(\text{hid})}$ and the transformation $T_{U_{\mathcal{D}}}$.

Example (Table 1, row: rot,vflip): For *image tasks*, if $\mathcal{G}_{\mathcal{I}} = \mathcal{G}_{\text{rot, vertical-flip}}$ and $\mathcal{G}_{\mathcal{D}} = \mathcal{G}_{\text{color}}$, then the training data consists of **upright** and **unflipped** images (as $T_{U'_{\mathcal{I}}|U_{\mathcal{D}}} = T_{\text{identity}}$) with different permutations of the color channels (random transformations $T_{U_{\mathcal{D}}} \in \mathcal{G}_{\text{color}}$ are chosen). The task is to predict the original label of the image (i.e., the digit) and the transformation $T_{\mathcal{D}}$ (i.e., the color).

**Extrapolation task:** The extrapolated test data consists of samples from the coupled random variable $X_{U_{\mathcal{I}} \leftarrow \widetilde{U}_{\mathcal{I}}} = T_{\widetilde{U}_{\mathcal{I}}, U_{\mathcal{D}}} \circ X^{(\text{hid})}$ (Definition 1). As before, we decompose $T_{\widetilde{U}_{\mathcal{I}}, U_{\mathcal{D}}} = T_{\widetilde{U}'_{\mathcal{I}}|U_{\mathcal{D}}} \circ T_{U_{\mathcal{D}}}$ with $T_{U_{\mathcal{D}}} \in \mathcal{G}_{\mathcal{D}}$ and $T_{\widetilde{U}'_{\mathcal{I}}|U_{\mathcal{D}}} \in \mathcal{G}_{\mathcal{I}}$. However, there is no economic sampling for the test data: $T_{\widetilde{U}'_{\mathcal{I}}|U_{\mathcal{D}}}$ and $T_{U_{\mathcal{D}}}$ are sampled randomly from $\mathcal{G}_{\mathcal{I}}$ and $\mathcal{G}_{\mathcal{D}}$ respectively. The task is the same as in the training data.

Example (Table 1, row: rot,vflip): For *image tasks*, if $\mathcal{G}_{\mathcal{I}} = \mathcal{G}_{\text{rot, vertical-flip}}$ and $\mathcal{G}_{\mathcal{D}} = \mathcal{G}_{\text{color}}$, then the extrapolation test data consists of images randomly **rotated**, **flipped** and **color permuted**, while the task is the same: predict the digit and its color.

**Results:** Standard neural networks such as CNNs (e.g., VGG (Simonyan and Zisserman, 2014)) (for images) and GRUs/Transformers (Cho et al., 2014; Vaswani et al., 2017) (for sequences) fail whenever the extrapolation task requires some invariance ($\mathcal{I} \neq \emptyset$), but excel at the *interpolation task* ($\mathcal{I} = \emptyset$). Adding forced $\mathcal{G}_{\mathcal{D}\cup\mathcal{I}}$-invariances via G-CNNs (Cohen and Welling, 2016) (for images) and permutation-invariant models (Lee et al., 2019; Murphy et al., 2019a; Zaheer et al., 2017) (for sequences) clearly fails when $\mathcal{D} \neq \emptyset$ but succeeds when $\mathcal{D} = \emptyset$. Our CG-regularized neural network representations, on the other hand, achieve high extrapolation accuracy across all tasks for all choices of $\mathcal{I} \subseteq \{1, \ldots, m\}$ and $\mathcal{D} \subseteq \{1, \ldots, m\} \setminus \mathcal{I}$. These results plainly show that our approach is able to selectively learn to be invariant only to the appropriate groups. Furthermore, this $\mathcal{G}_{\mathcal{I}}$-invariance is achieved without any evidence in the training data, thanks to our novel learning paradigm that considers all G-invariances mandatory unless contradicted by the training data.

## 7 Conclusion

This work studied the task of learning representations that can extrapolate beyond the training data distribution (environment), even when presented with a single training environment. We considered the case of (counterfactual) extrapolation from linear automorphism groups and described a framework where all G-invariances (and CG-invariances via Theorem 2) are mandatory, except the ones deemed inconsistent with the training data (i.e., rather than learning G-invariances, we unlearn them). Our framework reframes the standard statistical learning hypothesis that *unseen-data means underspecified-models* with a learning hypothesis that forces models to have all (known) G-invariances (symmetries) that do not contradict the data, with our empirical results supporting the proposed approach. Finally, this learning paradigm offers a promising novel research direction for neural network extrapolations.

### Acknowledgments

This work was funded in part by the National Science Foundation (NSF) Awards CAREER IIS-1943364 and CCF-1918483, the Purdue Integrative Data Science Initiative, and the Wabash Heartland Innovation Network. Any opinions, findings and conclusions or recommendations expressed in this material are those of the authors and do not necessarily reflect the views of the sponsors.

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

# Supplementary Material of "Neural Networks for Learning Counterfactual G-Invariances from Single Environments"

## A  THE PRACTICAL IMPORTANCE OF THEOREM 1

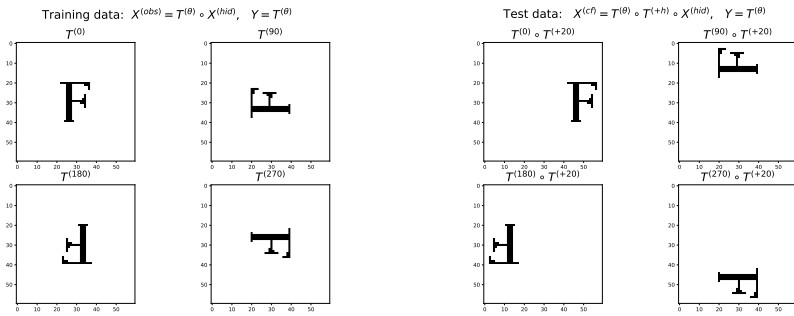

Figure 2: An example task where CG-invariance is stronger than G-invariance. The task is to predict the orientation of the image while being CG-invariant to horizontal translations.

*There are real tasks where CG-invariance is stronger than G-invariance.* We consider a task with $60 \times 60$ image shown in Figure 2 and two transformation groups: the rotation group $\mathcal{G}_{\text{rot}}$ and the cyclic horizontal-translation group $\mathcal{G}_{\text{h-translate}} \cong \mathbb{Z}_{60}$. Each transformation $T^{(\theta^\circ)} \in \mathcal{G}_{\text{rot}}$ rotates the image along its center by $\theta^\circ$, whereas every transformation $T^{(+h)} \in \mathcal{G}_{\text{h-translate}}$ translates the image horizontally by $h$ pixels while wrapping around the edges. Let $\mathcal{G}_{\mathcal{I}} = \mathcal{G}_{\text{rot}}$ and $\mathcal{G}_{\mathcal{D}} = \mathcal{G}_{\text{h-translate}}$. The training data consists of images $X = T^{(\theta^\circ)} \circ X^{(\text{hid})}$ for all $T^{(\theta^\circ)} \in \mathcal{G}_{\text{rot}}$, whereas the test data consists of images $X_{U_{\mathcal{I}} \leftarrow \widetilde{U}_{\mathcal{I}}} = T^{(\theta^\circ)} \circ T^{(+20)} \circ X^{(\text{hid})}$ for all $T^{(\theta^\circ)} \in \mathcal{G}_{\text{rot}}$ The task is to predict the orientation of the image, i.e., degrees of rotation. It is easy to see that the label requires CG-invariance to $\mathcal{G}_{\text{h-translate}}$ but sensitivity to $\mathcal{G}_{\text{rot}}$. We train a *strictly* $\mathcal{G}_{\text{h-translate}}$-invariant model on this dataset; whereas the model is able to achieve a 100% accuracy on training, it does poorly with 75% on test dataset, showing that it is not enough to be $\mathcal{G}_{\mathcal{I}}$-invariant to achieve CG-invariance.

## B  PROOFS

### B.1  GENERATING ANY $T \in \mathcal{G}_{\mathcal{D} \cup \mathcal{I}}$ USING NOISES $U_{\mathcal{I}}$ AND $U_{\mathcal{D}}$

The structural causal model for $X$ in Equation (2) requires that any $T \in \mathcal{G}_{\mathcal{D} \cup \mathcal{I}}$ can be indexed by the hidden background variables $U_{\mathcal{I}}$ and $U_{\mathcal{D}}$. We first interpret $U_{\mathcal{D}}$ (or $U_{\mathcal{I}}$) as the random seed of a random number generator that gives an ordered sequence of transformations of $\mathcal{G}_{\mathcal{D}}$ (or $\mathcal{G}_{\mathcal{I}}$). We assume that these background noise variables can generate any sequence of transformations from within their respective groups. Let $T_{\mathcal{D}}^{(1)}, \ldots, T_{\mathcal{D}}^{(a)}$ and $T_{\mathcal{I}}^{(1)}, \ldots, T_{\mathcal{I}}^{(b)}$ be those ordered sequences respectively generated by $U_{\mathcal{D}}$ and $U_{\mathcal{I}}$. Then we can obtain a transformation in $\mathcal{G}_{\mathcal{D} \cup \mathcal{I}}$ by interleaving these two sequences (in order): $T_{U_{\mathcal{D}}, U_{\mathcal{I}}} = T_{\mathcal{I}}^{(1)} \circ T_{\mathcal{D}}^{(1)} \circ T_{\mathcal{I}}^{(2)} \circ \ldots$. Note that $U_{\mathcal{I}}$ and $U_{\mathcal{D}}$ can always sample the identity transformation from the respective groups in the corresponding sequences, i.e., $T_{\mathcal{I}}^{(i)}$ or $T_{\mathcal{D}}^{(i)}$ can be identity.

Now, it is a known result in group theory that any $T \in \langle \mathcal{G}_{\mathcal{D}} \cup \mathcal{G}_{\mathcal{I}} \rangle$ is such that $T = T_1 \circ T_2 \circ T_3 \circ \ldots$, where $T_i$ is in either $\mathcal{G}_{\mathcal{I}}$ or $\mathcal{G}_{\mathcal{D}}$. Then, if $T_1 \in \mathcal{G}_{\mathcal{D}}$, we can write $T_1 = T_{\mathcal{I}}^{(1)} \circ T_{\mathcal{D}}^{(1)}$ with $T_{\mathcal{I}}^{(1)} = T_{\text{identity}} \in \mathcal{G}_{\mathcal{I}}$ and $T_{\mathcal{D}}^{(1)} = T_1 \in \mathcal{G}_{\mathcal{D}}$. Continuing in a similar fashion, we can find two sequences of transformations, one from $\mathcal{G}_{\mathcal{I}}$ and the other from $\mathcal{G}_{\mathcal{D}}$, such that interleaving and composing the resultant sequence of transformations gives us any transformation from $\mathcal{G}_{\mathcal{D} \cup \mathcal{I}}$. This property of the noises to appropriately index any $T \in \mathcal{G}_{\mathcal{D} \cup \mathcal{I}}$ will be used in the proof of Theorems 1 and 2.

## B.2 PROOF OF THEOREMS 1 AND 2

**Theorem 1** (CG-invariance is stronger than G-invariance). *Let the vector* $(X, X_{U_\mathcal{I} \leftarrow \widetilde{U}_\mathcal{I}})$ *denote the counterfactual coupling of the observed variable $X$ given in Definition 1. For a representation* $\Gamma : \mathcal{X} \to \mathbb{R}^d$, $d \geq 1$, *let*

$$G\text{-inv} : \ \forall T_\mathcal{I} \in \mathcal{G}_\mathcal{I}, \ \ \Gamma(X) = \Gamma(T_\mathcal{I} \circ X),$$
$$CG\text{-inv} : \ \Gamma(X) = \Gamma(X_{U_\mathcal{I} \leftarrow \widetilde{U}_\mathcal{I}}),$$

*denote the conditions on $\Gamma$ for $\mathcal{G}_\mathcal{I}$-invariance and CG-invariance respectively. Then, CG-inv $\implies$ G-inv, but G-inv $\centernot\implies$ CG-inv.*

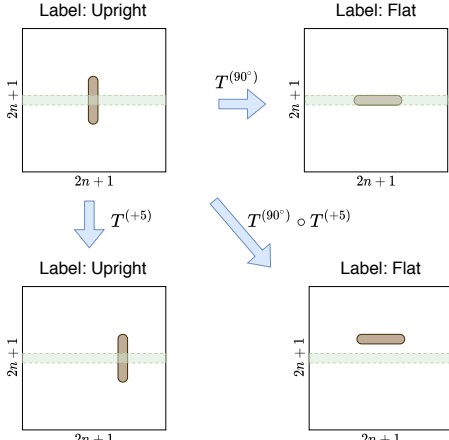

Figure 3: Counterexample to show that $\mathcal{G}_\mathcal{I}$-invariance does not imply CG-invariance. Given images of a rod (shown in brown), we wish to predict the orientation of the rod, i.e., whether the rod is upright or flat. In this example, we have $\mathcal{G}_\mathcal{D} = \mathcal{G}_{\text{rot}}$ and $\mathcal{G}_\mathcal{I} = \mathcal{G}_{\text{h-translate}}$ as any horizontal translation does not affect the orientation of the rod. $\Gamma : \mathcal{X} \to \mathbb{R}$ sums the pixel values across the green shaded region, and is clearly G-invariant to horizontal translations. However, $\Gamma$ is not CG-invariant.

*Proof.* First, we will show that CG-invariance $\implies$ G-invariance, i.e., for any CG-invariant representation $\Gamma : \mathcal{X} \to \mathbb{R}^d$, we will show that $\Gamma$ is also G-invariant to $\mathcal{G}_\mathcal{I}$.

Consider any $u \in \text{supp}(U_\mathcal{I})$ and say the input was generated as $X_{U_\mathcal{I} \leftarrow u} = T_{U_\mathcal{D}, U_\mathcal{I} \leftarrow u} \circ X^{(\text{hid})}$. In other words, $U_\mathcal{I}$ took the value $u$ in the structural causal equation for generating the observed input (Equation (2)). We will prove G-invariance for this input $X_{U_\mathcal{I} \leftarrow u}$, i.e., $\Gamma(T_\mathcal{I}^\dagger \circ X_{U_\mathcal{I} \leftarrow u}) = \Gamma(X_{U_\mathcal{I} \leftarrow u})$ for any $T_\mathcal{I}^\dagger \in \mathcal{G}_\mathcal{I}$.

Recall that $T_{U_\mathcal{D}, U_\mathcal{I} \leftarrow u}$ was generated by interleaving two separate sequences of transformations obtained via the background variables $U_\mathcal{D}$ and $U_\mathcal{I}$ respectively (Appendix B.1). In other words, we can write $T_{U_\mathcal{D}, U_\mathcal{I} \leftarrow u} = T_\mathcal{I}^{(1)} \circ T_\mathcal{D}^{(1)} \circ T_\mathcal{I}^{(2)} \circ \ldots \circ T^{(*)}$, where $T_\mathcal{I}^{(i)} \in \mathcal{G}_\mathcal{I}$ and $T_\mathcal{D}^{(j)} \in \mathcal{G}_\mathcal{D}$ and $T^{(*)}$ depends upon which of the respective sequences before interleaving is longer. Then, $T_\mathcal{I}^\dagger \circ T_{U_\mathcal{D}, U_\mathcal{I} \leftarrow u} = T_\mathcal{I}^\dagger \circ T_\mathcal{I}^{(1)} \circ T_\mathcal{D}^{(1)} \circ T_\mathcal{I}^{(2)} \circ \ldots \circ T^{(*)}$. Further, if we write $T_\mathcal{I}'^{(1)} = T_\mathcal{I}^\dagger \circ T_\mathcal{I}^{(1)}$, then we have $T_\mathcal{I}^\dagger \circ T_{U_\mathcal{D}, U_\mathcal{I} \leftarrow u} = T_\mathcal{I}'^{(1)} \circ T_\mathcal{D}^{(1)} \circ T_\mathcal{I}^{(2)} \circ \ldots \circ T^{(*)}$.

Now we can find a $\widetilde{u}$ such that $U_\mathcal{I} \leftarrow \widetilde{u}$ generates the sequence of transformations $T_\mathcal{I}'^{(1)}, T_\mathcal{I}^{(2)}, \ldots$. Interleaving this sequence with the sequence generated by $U_\mathcal{D}$, we get $T_{U_\mathcal{D}, U_\mathcal{I} \leftarrow \widetilde{u}} = T_\mathcal{I}'^{(1)} \circ T_\mathcal{D}^{(1)} \circ T_\mathcal{I}^{(2)} \circ \ldots \circ T^{(*)}$. Denote $X_{U_\mathcal{I} \leftarrow \widetilde{u}} = T_{U_\mathcal{D}, U_\mathcal{I} \leftarrow \widetilde{u}} \circ X^{(\text{hid})}$. Since $\Gamma$ is CG-invariant, we have from

Definition 2 that

$$\begin{aligned}
\Gamma(X_{U_{\mathcal{I}} \leftarrow u}) &= \Gamma(X_{U_{\mathcal{I}} \leftarrow \widetilde{u}}) \\
&= \Gamma(T_{U_{\mathcal{D}}, U_{\mathcal{I}} \leftarrow \widetilde{u}} \circ X^{(\text{hid})}) \\
&= \Gamma(T_{\mathcal{I}}^{\dagger} \circ T_{U_{\mathcal{D}}, U_{\mathcal{I}} \leftarrow u} \circ X^{(\text{hid})}) \qquad\qquad \text{(from construction of } \widetilde{u}) \\
&= \Gamma(T_{\mathcal{I}}^{\dagger} \circ X_{U_{\mathcal{I}} \leftarrow u}) \, .
\end{aligned}$$

Since this holds for all $u \in \text{supp}(U_{\mathcal{I}})$, we have that $\Gamma(X) = \Gamma(T_{\mathcal{I}}^{\dagger} \circ X)$.

Next, we will show G-invariance $\not\Longrightarrow$ CG-invariance by constructing a counterexample. Let $X^{(\text{hid})} \in \mathbb{R}^{(2n+1) \times (2n+1)}$ be the $(2n+1) \times (2n+1)$ grayscale image of an upright rod as shown in Figure 3. Consider two groups that act on this image: the rotation group $\mathcal{G}_{\text{rot}} = \{T^{(k)}\}_{k \in \{0°, 90°, 180°, 270°\}}$ and the cyclic horizontal-translation group $\mathcal{G}_{\text{h-translate}} = \{T^{(+u)}\}_{u \in \mathbb{Z}_n}$. Let $\mathcal{G}_{\mathcal{D}} = \mathcal{G}_{\text{rot}}$ and $\mathcal{G}_{\mathcal{I}} = \mathcal{G}_{\text{h-translate}}$ and the label of the image $Y$ deterministically given by the orientation of the rod: upright ($Y = 0$) or flat ($Y = 1$). The top row of Figure 3 depicts the data in training which is transformed by $\mathcal{G}_{\text{h-translate}}$ only via the identity $T^{(+0)}$ (i.e., no translation).

Now consider a representation $\Gamma : \mathbb{R}^{(2n+1) \times (2n+1)} \to \mathbb{R}$ such that $\Gamma(X) = \sum_{i=1}^{2n+1} X_{n,i}$ finds the sum of the middle row of the image. Note that **(a)** $\Gamma$ is able to distinguish between the labels for the training data, and **(b)** $\Gamma$ is $\mathcal{G}_{\text{h-translate}}$-invariant.

We can define the random variables $U_{\mathcal{I}}$ and $\widetilde{U}_{\mathcal{I}}$ such that $X = T^{(90°)} \circ X^{(\text{hid})}$ and $X_{U_{\mathcal{I}} \leftarrow \widetilde{U}_{\mathcal{I}}} = T^{(90°)} \circ T^{(+5)} \circ X^{(\text{hid})}$. Then, as shown in Figure 3, $\Gamma(X_{U_{\mathcal{I}} \leftarrow \widetilde{U}_{\mathcal{I}}}) = \Gamma(T^{(90°)} \circ T^{(+5)} \circ X^{(\text{hid})}) \neq \Gamma(T^{(90°)} \circ X^{(\text{hid})})$, thus showing that $\Gamma$ is not CG-invariant.

$\square$

**Theorem 2.** *If $\mathcal{G}_{\mathcal{I}}$ is a normal subgroup of $\mathcal{G}_{\mathcal{D} \cup \mathcal{I}}$, then CG-inv $\iff$ G-inv.*

*Proof.* The proof that CG-invariance $\implies$ G-invariance (from Theorem 1) still holds here. We only need to prove the converse: G-invariance $\implies$ CG-invariance when $\mathcal{G}_{\mathcal{I}}$ is a normal subgroup of $\mathcal{G}_{\mathcal{D} \cup \mathcal{I}}$. We begin with a representation $\Gamma$ that is $\mathcal{G}_{\mathcal{I}}$-invariant and consider the simpler case when $U_{\mathcal{D}}$ generates a transformation sequence of length 1 (from $\mathcal{G}_{\mathcal{D}}$). In other words, $X$ is obtained by: $X = T_{\mathcal{I}}^{(1)} \circ T_{\mathcal{D}} \circ T_{\mathcal{I}}^{(2)} \circ X^{(\text{hid})}$ for arbitrary transformations $T_{\mathcal{D}} \in \mathcal{G}_{\mathcal{D}}$ and $T_{\mathcal{I}}^{(1)}, T_{\mathcal{I}}^{(2)} \in \mathcal{G}_{\mathcal{I}}$.

Then for any $\widetilde{U}_{\mathcal{I}}$, we have that $X_{U_{\mathcal{I}} \leftarrow \widetilde{U}_{\mathcal{I}}} = \widetilde{T}_{\mathcal{I}}^{(1)} \circ T_{\mathcal{D}} \circ \widetilde{T}_{\mathcal{I}}^{(2)} \circ X^{(\text{hid})}$ with $\widetilde{T}_{\mathcal{I}}^{(1)}, \widetilde{T}_{\mathcal{I}}^{(2)} \in \mathcal{G}_{\mathcal{I}}$. Note that $\widetilde{U}_{\mathcal{I}}$ only affects the transformations from $\mathcal{G}_{\mathcal{I}}$. The condition for CG-invariance with respect to $\mathcal{G}_{\mathcal{I}}$ requires that

**requirement:** $\quad \Gamma(X) = \Gamma(T_{\mathcal{I}}^{(1)} \circ T_{\mathcal{D}} \circ T_{\mathcal{I}}^{(2)} \circ X^{(\text{hid})}) = \Gamma(\widetilde{T}_{\mathcal{I}}^{(1)} \circ T_{\mathcal{D}} \circ \widetilde{T}_{\mathcal{I}}^{(2)} \circ X^{(\text{hid})}) = \Gamma(X_{U_{\mathcal{I}} \leftarrow \widetilde{U}_{\mathcal{I}}}) \, .$

$\hfill (13)$

Since $\mathcal{G}_{\mathcal{I}}$ is a normal subgroup of $\mathcal{G}_{\mathcal{D} \cup \mathcal{I}}$ and $\mathcal{G}_{\mathcal{D}} \leq \mathcal{G}_{\mathcal{D} \cup \mathcal{I}}$, we have

$$\forall T_{\mathcal{D}} \in \mathcal{G}_{\mathcal{D}}, \ \ \forall T_{\mathcal{I}} \in \mathcal{G}_{\mathcal{I}}, \ \ T_{\mathcal{D}} \circ T_{\mathcal{I}} \circ T_{\mathcal{D}}^{-1} \in \mathcal{G}_{\mathcal{I}},$$

or equivalently,

$$\begin{aligned}
\forall T_{\mathcal{D}} \in \mathcal{G}_{\mathcal{D}}, \ \ &\forall T_{\mathcal{I}} \in \mathcal{G}_{\mathcal{I}}, \ \ \exists T_{\mathcal{I}}', \ \ \text{s.t.,} \\
&T_{\mathcal{D}} \circ T_{\mathcal{I}} \circ T_{\mathcal{D}}^{-1} = T_{\mathcal{I}}' \\
\implies \quad &T_{\mathcal{D}} \circ T_{\mathcal{I}} = T_{\mathcal{I}}' \circ T_{\mathcal{D}}
\end{aligned} \qquad (14)$$

(A special case is when the groups $\mathcal{G}_{\mathcal{D}}$ and $\mathcal{G}_{\mathcal{I}}$ commute, as then $T_{\mathcal{D}} \circ T_{\mathcal{I}} = T_{\mathcal{I}} \circ T_{\mathcal{D}}$.)

Then,

$$\begin{aligned}
\Gamma(X) &= \Gamma(T_{\mathcal{I}}^{(1)} \circ T_{\mathcal{D}} \circ T_{\mathcal{I}}^{(2)} \circ X^{(\text{hid})}) \\
&= \Gamma(T_{\mathcal{D}} \circ T_{\mathcal{I}}^{(2)} \circ X^{(\text{hid})}) && (\Gamma \text{ is invariant to } \mathcal{G}_{\mathcal{I}}) \\
&= \Gamma(T_{\mathcal{I}}' \circ T_{\mathcal{D}} \circ X^{(\text{hid})}) && (\text{there exists such a } T_{\mathcal{I}}' \in \mathcal{G}_{\mathcal{I}}) \\
&= \Gamma(T_{\mathcal{D}} \circ X^{(\text{hid})}) && (\Gamma \text{ is invariant to } \mathcal{G}_{\mathcal{I}})
\end{aligned}$$

Similarly, we can prove for the coupled variable that $\Gamma(X_{U_{\mathcal{I}} \leftarrow \widetilde{U}_{\mathcal{I}}}) = \Gamma(\widetilde{T}_{\mathcal{I}}^{(1)} \circ T_{\mathcal{D}} \circ \widetilde{T}_{\mathcal{I}}^{(2)} \circ X^{\text{(hid)}}) = \Gamma(T_{\mathcal{D}} \circ X^{\text{(hid)}})$, thus satisfying the requirement of CG-invariance in Equation (13).

Extension to the case when $U_{\mathcal{D}}$ generates transformation sequences of length greater than one is trivial. Any transformation $T_{U_{\mathcal{D}}, U_{\mathcal{I}}} = T_{\mathcal{I}}^{(1)} \circ T_{\mathcal{D}}^{(1)} \circ T_{\mathcal{I}}^{(2)} \cdots \circ T^{(*)}$ can be written in the form $T_{\mathcal{I}}^{\dagger} \circ T_{\mathcal{D}}^{(1)} \circ T_{\mathcal{D}}^{(2)} \circ \cdots$ by repeatedly applying the normal subgroup property in Equation (14). Then $\Gamma(T_{U_{\mathcal{D}}, U_{\mathcal{I}}} \circ X^{\text{(hid)}}) = \Gamma(T_{\mathcal{I}}^{\dagger} \circ T_{\mathcal{D}}^{(1)} \circ T_{\mathcal{D}}^{(2)} \circ \cdots \circ X^{\text{(hid)}}) = \Gamma(T_{\mathcal{D}}^{(1)} \circ T_{\mathcal{D}}^{(2)} \circ \cdots \circ X^{\text{(hid)}})$ as $\Gamma$ is $\mathcal{G}_{\mathcal{I}}$-invariant. Using a similar argument, we can show for the coupled variable that $\Gamma(T_{U_{\mathcal{D}}, \widetilde{U}_{\mathcal{I}}} \circ X^{\text{(hid)}}) = \Gamma(T_{\mathcal{D}}^{(1)} \circ T_{\mathcal{D}}^{(2)} \circ \cdots \circ X^{\text{(hid)}})$, thus proving that $\Gamma$ is CG-invariant, i.e., $\Gamma(X) = \Gamma(X_{U_{\mathcal{I}} \leftarrow \widetilde{U}_{\mathcal{I}}})$. $\quad\square$

### B.3 PROOFS OF LEMMA 1, LEMMA 2 AND THEOREM 3

**Lemma 1** (Reynolds operator (Mumford et al. (1994), Definition 1.5)). *Let $\mathcal{G}$ be a (finite) linear automorphism group over $vec(\mathcal{X})$. Then,*

$$\overline{T} = \frac{1}{|\mathcal{G}|} \sum_{T \in \mathcal{G}} T \tag{7}$$

*is a $\mathcal{G}$-invariant linear automorphism, i.e., $\forall T_{\dagger} \in \mathcal{G}$ and $\forall \boldsymbol{x} \in vec(\mathcal{X})$, it must be that $\overline{T}(T_{\dagger}\boldsymbol{x}) = \overline{T}\boldsymbol{x}$.*

*Proof.* Consider an arbitrary transformation $T_{\dagger} \in \mathcal{G}$. Then

$$\begin{aligned}
\overline{T} \circ T_{\dagger} &= \frac{1}{|\mathcal{G}|} \sum_{T \in \mathcal{G}} T \circ T_{\dagger} \\
&= \frac{1}{|\mathcal{G}|} \sum_{T' \in \mathcal{G}_{\dagger}} T' ,
\end{aligned}$$

where we define $\mathcal{G}_{\dagger} = \{T \circ T_{\dagger} : \forall T \in \mathcal{G}\}$. Now, in order to prove $\overline{T} \circ T_{\dagger} = \overline{T}$, we only need to show that $\mathcal{G}_{\dagger} = \mathcal{G}$. Since groups are closed under compositions, we have $\forall T \in \mathcal{G}, T \circ T_{\dagger} \in \mathcal{G}$, and thus $\mathcal{G}_{\dagger} \subseteq \mathcal{G}$. Finally, since $T_{\dagger}$ is a bijection and $T_a \circ T_{\dagger} = T_b \circ T_{\dagger}$ only if $T_a = T_b$ for any $T_a, T_b \in \mathcal{G}$, it must be that $|\mathcal{G}_{\dagger}| = |\mathcal{G}|$. Hence, $\mathcal{G}_{\dagger} = \mathcal{G}$.

$\square$

**Lemma 2.** *If $\mathcal{W}$ denotes the left eigenspace corresponding to the eigenvalue 1 of the Reynolds operator $\overline{T}$ for the group $\mathcal{G}$, then $\forall b \in \mathbb{R}$, the linear transformation $\gamma(\boldsymbol{x}; \boldsymbol{w}, b) = \boldsymbol{w}^T \boldsymbol{x} + b$ is invariant to all transformations $T \in \mathcal{G}$, i.e., $\gamma(T\boldsymbol{x}; \boldsymbol{w}, b) = \gamma(\boldsymbol{x}; \boldsymbol{w}, b)$, **if and only if** $\boldsymbol{w} \in \mathcal{W}$.*

*Proof. Sufficiency:* Let $\{\boldsymbol{w}_i^T\}_{i=1}^{d_W}$ be the set of left eigenvectors of $\overline{T}$ with eigenvalue 1 and constitute the orthogonal basis for $\mathcal{W}$. Consider any non-zero $\boldsymbol{w}' \in \mathcal{W}$, then

$$(\boldsymbol{w}')^T = \sum_{i=1}^{d_W} \alpha_i \boldsymbol{w}_i^T = \sum_{i=1}^{d_W} \alpha_i \boldsymbol{w}_i^T \overline{T} \tag{15}$$

for some coefficients $\{\alpha_i\}_{i=1}^{d_W}$, where we used the fact that $\boldsymbol{w}_i^T \overline{T} = \boldsymbol{w}_i^T, 1 \leq i \leq d_W$. For any $\boldsymbol{x} \in \text{vec}(\mathcal{X})$ and any $T \in \mathcal{G}$ we have,

$$\begin{aligned}
\gamma(T\boldsymbol{x}; \boldsymbol{w}', b) &= (\boldsymbol{w}')^T (T\boldsymbol{x}) + b \\
&= \sum_{i=1}^{d_W} \alpha_i \boldsymbol{w}_i^T \overline{T}(T\boldsymbol{x}) + b && \text{(using Equation (15))} \\
&= \sum_{i=1}^{d_W} \alpha_i \boldsymbol{w}_i^T \overline{T}\boldsymbol{x} + b && \text{(from Lemma 1)} \\
&= \gamma(\boldsymbol{x}; \boldsymbol{w}', b)
\end{aligned}$$

*Necessity:* Given a non-zero $\boldsymbol{w} \in \mathcal{W}$ and $b \in \mathbb{R}$, let $\gamma(T\boldsymbol{x}; \boldsymbol{w}, b) = \gamma(\boldsymbol{x}; \boldsymbol{w}, b)$ for all $\boldsymbol{x} \in \text{vec}(\mathcal{X})$ and all $T \in \mathcal{G}$. Then,

$$\boldsymbol{w}^T T \boldsymbol{x} = \boldsymbol{w}^T \boldsymbol{x}, \quad \forall \boldsymbol{x}, \forall T$$

$$\implies \boldsymbol{w}^T T = \boldsymbol{w}^T, \quad \forall T$$

$$\implies \boldsymbol{w}^T \sum_{T \in \mathcal{G}} T = |\mathcal{G}| \boldsymbol{w}^T \qquad \text{(summing over all } T \in \mathcal{G})$$

$$\implies \boldsymbol{w}^T \overline{T} = \boldsymbol{w}^T.$$

Hence proved that $\boldsymbol{w}^T$ is a left eigenvector of $\overline{T}$ with eigenvalue 1.

$\square$

**Theorem 3** (G-invariant subspace bases can be partially ordered by invariance strength). *Let $\mathcal{W}_i \subseteq vec(\mathcal{X})$ be the left eigenspace corresponding to the eigenvalue 1 of the Reynolds operator $\overline{T}_i$ for group $\mathcal{G}_i$, $i = 1, \ldots, m$. We construct the invariant subspace partitions*

$$\widetilde{\mathcal{B}}_M = \bigcap_{i \in M} \mathcal{W}_i; \quad \mathcal{B}_M = orth_{\mathcal{B}_{\supsetneq M}}(\widetilde{\mathcal{B}}_M), \quad \forall M \in \wp(\{1, \ldots, m\}) \setminus \emptyset, \qquad (8)$$

*where $\wp$ is the power set, $\mathcal{B}_{\supsetneq M} = \bigoplus_{N \supsetneq M} \mathcal{B}_N$, $orth_{\mathcal{A}_1}(\mathcal{A}_2)$ removes from the subspace $\mathcal{A}_2$ its orthogonal projection onto the subspace $\mathcal{A}_1$, and $\bigoplus$ is the direct sum operator.* **Then***, the linear transformation $\gamma(\boldsymbol{x}; \boldsymbol{w}, b) = \boldsymbol{w}^T \boldsymbol{x} + b$, $b \in \mathbb{R}$, $\forall \boldsymbol{w} \in \mathcal{B}_M \setminus \{\boldsymbol{0}\}$, is $\mathcal{G}_M$-invariant but not $\mathcal{G}_j$-invariant $\forall j \in \{1, \ldots, m\} \setminus M$.*

*Proof.* Throughout this proof, we will slightly abuse notation by calling a $\boldsymbol{w} \in \text{vec}(\mathcal{X})$ as $\mathcal{G}$-invariant for some group $\mathcal{G}$, where we mean the transformation $\gamma(\cdot; \boldsymbol{w}, b), b \in \mathbb{R}$ is $\mathcal{G}$-invariant.

Consider the subspace $\mathcal{B}_{\supsetneq M} = \bigoplus_{N \supsetneq M} \mathcal{B}_N$, where $\bigoplus$ is the direct sum operator. Essentially, $\mathcal{B}_{\supsetneq M}$ is the direct sum of all the subspaces corresponding to the strict supersets of $M$. Using induction on the size of $M$, we first show that $\mathcal{B}_{\supsetneq M} = \bigoplus_{N \supsetneq M} \widetilde{\mathcal{B}}_N$. The statement trivially holds for $\mathcal{B}_{\supsetneq \{1, \ldots, m\}}$. Then the induction hypothesis is: for all sets $M$ such that $|M| > k$, we have $\mathcal{B}_{\supsetneq M} = \bigoplus_{N \supsetneq M} \widetilde{\mathcal{B}}_N$. We prove that the statement holds for any set $M$ with $|M| = k$ as follows,

$$\mathcal{B}_{\supsetneq M} = \bigoplus_{N \supsetneq M} \mathcal{B}_N$$

$$= \bigoplus_{\substack{N \supsetneq M \\ |N| = |M| + 1}} (\mathcal{B}_N \oplus \mathcal{B}_{\supsetneq N})$$

$$= \bigoplus_{\substack{N \supsetneq M \\ |N| = |M| + 1}} \left( orth_{\mathcal{B}_{\supsetneq N}}(\widetilde{\mathcal{B}}_N) \oplus \mathcal{B}_{\supsetneq N} \right) \qquad \text{(Definition of } \mathcal{B}_N)$$

$$= \bigoplus_{\substack{N \supsetneq M \\ |N| = |M| + 1}} \left( \widetilde{\mathcal{B}}_N \oplus \mathcal{B}_{\supsetneq N} \right) \quad \text{(For vector subspaces } V \text{ and } W, \, orth_W(V) \oplus W = V \oplus W)$$

$$= \bigoplus_{\substack{N \supsetneq M \\ |N| = |M| + 1}} \left( \widetilde{\mathcal{B}}_N \oplus \widetilde{\mathcal{B}}_{\supsetneq N} \right) \qquad \text{(Inductive hypothesis holds for sets } N \text{ as } |N| > k)$$

$$= \bigoplus_{N \supsetneq M} \widetilde{\mathcal{B}}_N.$$

This proves our claim that $\mathcal{B}_{\supsetneq M} = \bigoplus_{N \supsetneq M} \mathcal{B}_N = \bigoplus_{N \supsetneq M} \widetilde{\mathcal{B}}_N$.

Now we are ready to prove the theorem. We begin by showing that any nonzero $\boldsymbol{w} \in \text{vec}(\mathcal{X})$ is $\mathcal{G}_M$-invariant where $\mathcal{G}_M = \langle \cup_{i \in M} \mathcal{G}_i \rangle$ iff $\boldsymbol{w} \in \widetilde{\mathcal{B}}_M$. Since $\boldsymbol{w} \in \widetilde{\mathcal{B}}_M \iff \boldsymbol{w} \in \mathcal{W}_i, \, \forall i \in M$, we have from Lemma 2 that any $\boldsymbol{w} \in \widetilde{\mathcal{B}}_M$ is $\mathcal{G}_i$-invariant for all $i \in M$. Then it is easy to see that

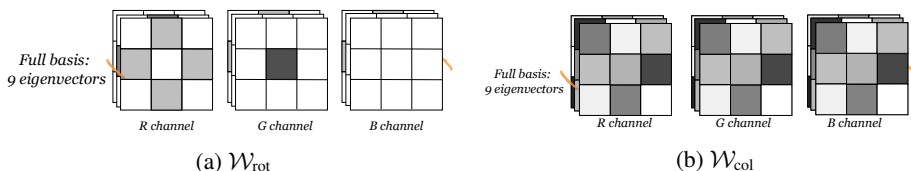

(a) $\mathcal{W}_{\text{rot}}$        (b) $\mathcal{W}_{\text{col}}$

Figure 4: **(a)** 1-eigenspace of the Reynolds operator for the rotation group. The eigenspace has nine basis vectors $v \in \mathbb{R}^{27}$ (stacked). We are representing these eigenvectors in $\mathbb{R}^{3 \times 3 \times 3}$ instead to emphasize that these are **rotation-invariant**. **(b)** 1-eigenspace of the Reynolds operator for the color-permutation group. The eigenspace again has nine basis vectors $v \in \mathbb{R}^{27}$ but we represent them in $\mathbb{R}^{3 \times 3 \times 3}$ to emphasize that these are **invariant to permutations of color channels**.

any nonzero $\boldsymbol{w}$ is $\mathcal{G}_M$-invariant iff it is $\mathcal{G}_i$-invariant for all $i \in M$. It is possible to have $\widetilde{\mathcal{B}}_M = \{\mathbf{0}\}$ implying that there is no nonzero $\boldsymbol{w} \in \text{vec}(\mathcal{X})$ that is $\mathcal{G}_M$-invariant.

Next note that for all $N \supsetneq M$, we have $\widetilde{\mathcal{B}}_N \subseteq \widetilde{\mathcal{B}}_M$ (using the definition of $\widetilde{\mathcal{B}}_M$). Then, their direct sum is the smallest subspace containing all such $\widetilde{\mathcal{B}}_N$ and thus, $\bigoplus_{N \supsetneq M} \widetilde{\mathcal{B}}_N \subseteq \widetilde{\mathcal{B}}_M$. From our claim earlier, this implies that $\mathcal{B}_{\supsetneq M} = \bigoplus_{N \supsetneq M} \widetilde{\mathcal{B}}_N \subseteq \widetilde{\mathcal{B}}_M$. Finally, we have $\mathcal{B}_M = \text{orth}_{\mathcal{B}_{\supsetneq M}}(\widetilde{\mathcal{B}}_M) \subseteq \widetilde{\mathcal{B}}_M$ for all $M$. Thus, we have proved that any nonzero $\boldsymbol{w} \in \mathcal{B}_M$ also lies in $\widetilde{\mathcal{B}}_M$ and hence is invariant to $\mathcal{G}_M$.

In the sequel, we will prove that any $\boldsymbol{w} \in \mathcal{B}_M$ is **not** $\mathcal{G}_j$-invariant for any $j \in \{1, \ldots, m\} \setminus M$. Let $P \supsetneq M$. Then it is clear that $\mathcal{B}_{\supsetneq M} = \bigoplus_{N \supsetneq M} \widetilde{\mathcal{B}}_N \supseteq \widetilde{\mathcal{B}}_P$, which implies from the first part of our proof that any $\boldsymbol{w} \in \text{vec}(\mathcal{X})$ that is $\mathcal{G}_P$-invariant lies inside $\mathcal{B}_{\supsetneq M}$. The orthogonalization step ensures that $\mathcal{B}_M \perp \mathcal{B}_{\supsetneq M}$ and thus, $\mathcal{B}_M \perp \widetilde{\mathcal{B}}_P$ and $\mathcal{B}_M \cap \widetilde{\mathcal{B}}_P = \{\mathbf{0}\}$. Hence there is no nonzero $\boldsymbol{w} \in \mathcal{B}_M$ such that $\boldsymbol{w}$ is $\mathcal{G}_P$-invariant. This applies for all supersets $P \supsetneq M$.

Finally, we consider supersets of $M$ of the form $P' = M \cup \{j\}$ for $j \in \{1, \ldots, m\} \setminus M$. If a nonzero $\boldsymbol{w} \in \mathcal{B}_M$ is invariant to $\mathcal{G}_j$, then it will hold that $\boldsymbol{w}$ is invariant to $\mathcal{G}_{P'}, P' \supsetneq M$, resulting in a contradiction. Hence, we have that if $\mathcal{B}_M \neq \{\mathbf{0}\}$, any $\boldsymbol{w} \in \mathcal{B}_M \setminus \{\mathbf{0}\}$ is $\mathcal{G}_M$-invariant but not $\mathcal{G}_j$-invariant for any $j \in \{1, \ldots, m\} \setminus M$.

$\square$

## C    EXAMPLE CONSTRUCTION OF CG-INVARIANT NEURONS

In this section, we will present a detailed example of the construction of CG-invariant neurons. Consider a $3 \times 3$ image with 3 channels, thus $\mathcal{X} = \mathbb{R}^{3 \times 3 \times 3}$. Then, a convolutional filter $\boldsymbol{w} \in \mathcal{X} = \mathbb{R}^{3 \times 3 \times 3}$ multiplies elementwise with the image $\boldsymbol{x} \in \mathcal{X}$.

Consider $m = 2$ groups $\mathcal{G}_{\text{rot}}$ and $\mathcal{G}_{\text{col}}$, the former rotates the image patch by 90-degree multiples and the latter permutes the color channels of the image. Our goal is to enforce invariance to rotation and color channel unless contradicted by training data. Note that $\text{vec}(\mathcal{X}) = \mathbb{R}^{27}$.

**Step 1: Construct 1-eigenspace of Reynolds operator for each group.** Since we only consider linear automorphism groups, each transformation $T$ in the group can be written as $T(x) = \boldsymbol{T}\boldsymbol{x}$, where $\boldsymbol{T}$ is a matrix of size $\mathbb{R}^{27 \times 27}$ and $\boldsymbol{x} \in \text{vec}(\mathcal{X}) = \mathbb{R}^{27}$. Given a group, we can directly use Lemma 1 to construct the Reynolds operator by averaging over all the linear transformations (or corresponding matrices) in the group. Then, we can use standard methods in linear algebra to find the 1-eigenspace of the Reynolds operator (i.e., find the eigenvectors with corresponding eigenvalues equal to 1).

Let $\mathcal{W}_{\text{rot}}$ and $\mathcal{W}_{\text{col}}$ be the 1-eigenspaces of the Reynolds operator of the groups $\mathcal{G}_{\text{rot}}$ and $\mathcal{G}_{\text{col}}$ respectively. Figure 4 shows these eigenspaces with the eigenvectors arranged in $\mathbb{R}^{3 \times 3 \times 3}$ instead of $\mathbb{R}^{27}$. The figure shows that the eigenvectors in $\mathcal{W}_{\text{rot}}$ are invariant to rotations of 90-degree multiples whereas the eigenvectors in $\mathcal{W}_{\text{col}}$ have the same values across the RGB channels, and thus are invariant to permutation of these channels. Lemma 2 proves this invariance-property for the 1-eigenspaces of the Reynolds operator of **any** finite linear automorphism group.

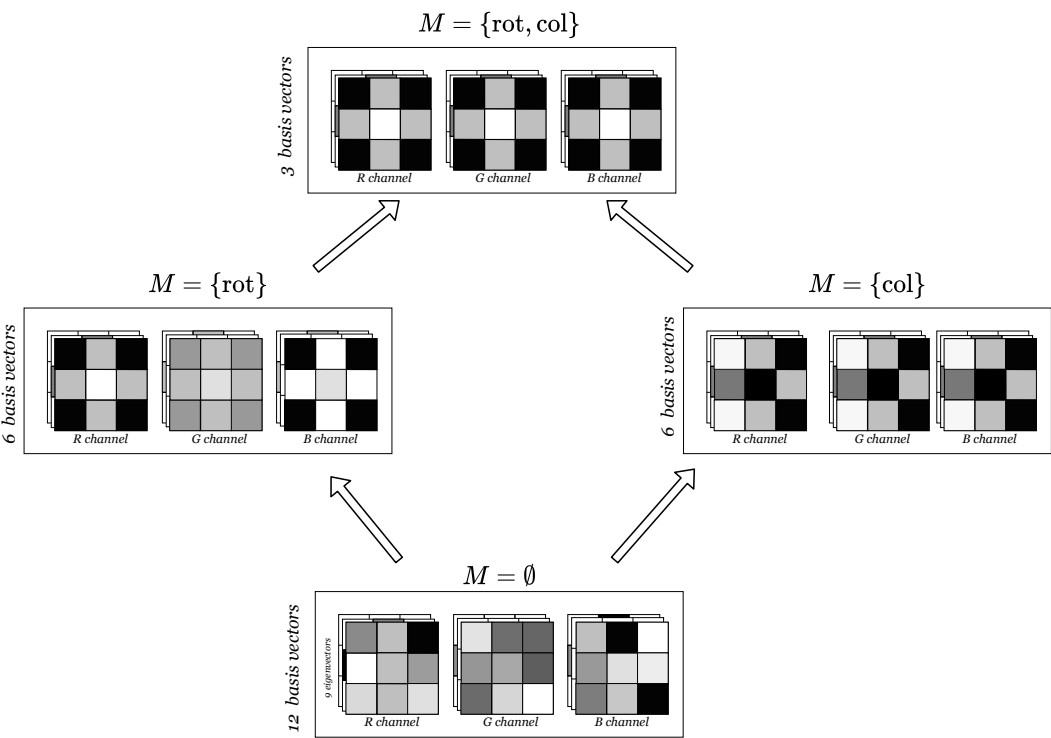

Figure 5: The subspaces $\mathcal{B}_M$ for all $M \subseteq \{\text{rot}, \text{col}\}$. For instance, $\mathcal{B}_{\{\text{rot,col}\}}$ on the top has 3 basis vectors (represented in $\mathbb{R}^{3 \times 3 \times 3}$) and each of these vectors are **both rotation-invariant and channel-permutation invariant**. On the other hand, $\mathcal{B}_{\{\text{rot}\}}$ (of dimension 6) is rotation invariant but strictly not channel-permutation invariant. Finally, the vectors in $\mathcal{B}_\emptyset$ are neither rotation-invariant nor channel-permutation invariant. All the basis vectors together cover the entire space $\mathbb{R}^{27}$ (i.e., $\dim(\mathcal{B}_{\{\text{rot,col}\}}) + \dim(\mathcal{B}_{\{\text{rot}\}}) + \dim(\mathcal{B}_{\{\text{col}\}}) + \dim(\mathcal{B}_\emptyset) = 3 + 6 + 6 + 12 = 27$).

**Step 2: Construct $\mathcal{B}_M$ for all $M \subseteq \{\text{rot}, \text{col}\}$.** Now, given $\mathcal{W}_{\text{rot}}$ and $\mathcal{W}_{\text{col}}$, we will construct basis for the subspaces $\mathcal{B}_M$ for all $M \subseteq \{\text{rot}, \text{col}\}$ using Theorem 3.

1. Set $M = \{\text{rot}, \text{col}\}$.

$$\widetilde{\mathcal{B}}_{\{\text{rot,col}\}} = \mathcal{W}_{\text{rot}} \cap \mathcal{W}_{\text{col}}$$
$$\mathcal{B}_{\{\text{rot,col}\}} = \widetilde{\mathcal{B}}_{\{\text{rot,col}\}} . \qquad\qquad (\text{because } \mathcal{B}_{\supsetneq \{\text{rot,col}\}} = \{\mathbf{0}\})$$

The intersection of subspaces $\mathcal{W}_{\text{rot}} \cap \mathcal{W}_{\text{col}}$ can be computed using standard methods in linear algebra. The subspace $\mathcal{B}_{\{\text{rot,col}\}}$ with 3 basis vectors is visualized in the topmost level of Figure 5. As before the basis vectors of the subspace are represented in $\mathbb{R}^{3 \times 3 \times 3}$. It is clear that the basis vectors are **invariant to both rotation and permutation of the channels**. This property will hold for any linear combination of the basis vectors, i.e., for any $\boldsymbol{w} \in \mathcal{B}_{\{\text{rot,col}\}}$.

2. Set $M = \{\text{rot}\}$.

$$\widetilde{\mathcal{B}}_{\{\text{rot}\}} = \mathcal{W}_{\text{rot}}$$
$$\mathcal{B}_{\{\text{rot}\}} = \text{orth}_{\mathcal{B}_{\supsetneq\{\text{rot}\}}}(\widetilde{\mathcal{B}}_{\{\text{rot}\}})$$
$$= \text{orth}_{\mathcal{B}_{\{\text{rot,col}\}}}(\widetilde{\mathcal{B}}_{\{\text{rot}\}}) \qquad\qquad (\text{because } \mathcal{B}_{\supsetneq\{\text{rot}\}} = \mathcal{B}_{\{\text{rot,col}\}})$$

The subspace $\widetilde{\mathcal{B}}_{\{\text{rot}\}}$ consists of all vectors that are invariant to rotation but also includes vectors that are invariant to both rotation and channel-permutation. Thus, we need to remove from $\widetilde{\mathcal{B}}_{\{\text{rot}\}}$ the projection of $\widetilde{\mathcal{B}}_{\{\text{rot}\}}$ on $\mathcal{B}_{\{\text{rot,col}\}}$.

The subspace $\mathcal{B}_{\{\text{rot}\}}$ with 6 basis vectors is visualized in middle level of Figure 5. It is clear that the basis vectors are **invariant to rotation but not invariant to channel-permutations**. Again, this property holds for any linear combination of the basis vectors.

3. Set $M = \{\text{col}\}$.

$$\widetilde{\mathcal{B}}_{\{\text{col}\}} = \mathcal{W}_{\text{col}}$$

$$\mathcal{B}_{\{\text{col}\}} = \text{orth}_{\mathcal{B}_{\supsetneq\{\text{col}\}}}(\widetilde{\mathcal{B}}_{\{\text{col}\}})$$

$$= \text{orth}_{\mathcal{B}_{\{\text{rot,col}\}}}(\widetilde{\mathcal{B}}_{\{\text{col}\}}) \qquad\qquad (\text{because } \mathcal{B}_{\supsetneq\{\text{col}\}} = \mathcal{B}_{\{\text{rot,col}\}})$$

The subspace $\mathcal{B}_{\{\text{col}\}}$ is obtained in a similar fashion. $\mathcal{B}_{\{\text{col}\}}$ has 6 basis vectors and is visualized in middle level of Figure 5. It is clear that the basis vectors are **invariant to channel-permutations but not invariant to rotation**. This property holds for any linear combination of the basis vectors.

4. Set $M = \emptyset$.

$$\mathcal{B}_{\emptyset} = \text{orth}_{\mathcal{B}_{\supseteq\emptyset}}(\text{vec}(\mathcal{X}))\,,$$

where $\mathcal{B}_{\supseteq\emptyset} = \mathcal{B}_{\{\text{rot,col}\}} \oplus \mathcal{B}_{\{\text{rot}\}} \oplus \mathcal{B}_{\{\text{col}\}}$. The subspace $\mathcal{B}_{\emptyset}$ represents the rest of the space that is **neither rotation-invariant nor channel-permutation-invariant**. This subspace has 12 basis vectors and is visualized in the bottommost level of Figure 5.

Finally, we have $B = 4$ subspaces (enumerated above) with a total of 27 basis vectors covering the entire space $\text{vec}(\mathcal{X}) = \mathbb{R}^{27}$.

**Step 3: Neuron construction.** For each subspace $\mathcal{B}_M$, $M \subseteq \{\text{rot}, \text{col}\}$, we denote $\boldsymbol{B}_M$ as the corresponding matrix with columns as the basis vectors of the subspace $\mathcal{B}_M$. As described above any linear combination of the basis vectors of $\mathcal{B}_M$ are invariant to all groups indexed by $M$ and nothing more (e.g., $\mathcal{B}_{\{\text{rot}\}}$ consists of vectors invariant to rotation but not invariant to channel-permutation).

In the following, we consider a single neuron and drop the subscript $h$ from $\boldsymbol{\omega}_{M,h}$ (where $h$ represented the $h$-th neuron in Equation (10)). Recall that $\boldsymbol{\omega}_M \in \mathbb{R}^{d_M}$ are the learnable parameters of the neuron corresponding to each basis vector of the subspace $\mathcal{B}_M$, and $d_M$ is the dimension of the subspace $\mathcal{B}_M$. Then, $\boldsymbol{\omega}_{\{\text{rot,col}\}} \in \mathbb{R}^3$ represents the coefficients in the linear combination of the basis vectors in $\boldsymbol{B}_{\{\text{rot,col}\}}$. The linear combination is given by the matrix-vector product $\boldsymbol{B}_{\{\text{rot,col}\}}\boldsymbol{\omega}_{\{\text{rot,col}\}}$. Similarly, $\boldsymbol{\omega}_{\{\text{rot}\}} \in \mathbb{R}^6$, $\boldsymbol{\omega}_{\{\text{col}\}} \in \mathbb{R}^6$, $\boldsymbol{\omega}_{\emptyset} \in \mathbb{R}^{12}$ represent the coefficients of the basis vectors in the columns of $\boldsymbol{B}_{\{\text{rot}\}}$, $\boldsymbol{B}_{\{\text{col}\}}$ and $\boldsymbol{B}_{\emptyset}$ respectively.

Then, a CG-invariant neuron is given by,

$$\Gamma(\boldsymbol{x}) = \boldsymbol{x}^T \boldsymbol{w} + b\,,$$

where

$$\boldsymbol{w} = \boldsymbol{B}_{\{\text{rot,col}\}}\boldsymbol{\omega}_{\{\text{rot,col}\}} + \boldsymbol{B}_{\{\text{rot}\}}\boldsymbol{\omega}_{\{\text{rot}\}} + \boldsymbol{B}_{\{\text{col}\}}\boldsymbol{\omega}_{\{\text{col}\}} + \boldsymbol{B}_{\emptyset}\boldsymbol{\omega}_{\emptyset}\,,$$

and $\boldsymbol{\omega}_{\{\text{rot,col}\}}$, $\boldsymbol{\omega}_{\{\text{rot}\}}$, $\boldsymbol{\omega}_{\{\text{col}\}}$, $\boldsymbol{\omega}_{\emptyset}$, $b \in \mathbb{R}$ are the only learnable parameters. The total number of parameters is 28, same as that of the standard neuron with input $\boldsymbol{x} \in \mathbb{R}^{27}$.

Now, if for example the optimization finds $\boldsymbol{\omega}_{\{\text{rot,col}\}} \neq \boldsymbol{0}$, $\boldsymbol{\omega}_{\{\text{rot}\}} = \boldsymbol{0}$, $\boldsymbol{\omega}_{\{\text{col}\}} = \boldsymbol{0}$ and $\boldsymbol{\omega}_{\emptyset} = \boldsymbol{0}$, then the neuron $\Gamma(\cdot)$ is invariant to both rotation and channel-permutation.

Our regularization in Equation (11) forces the optimization to find maximum invariance as long as training performance is unaffected. A more comprehensive example of the computation of the penalty is given in Appendix F.

## D  PSEUDOCODE FOR THEOREM 3

We present the algorithm for Theorem 3 in Algorithm 1. The loops in the algorithm iterate over the different subsets $M \subseteq \{1, \ldots, m\}$ in descending order of their sizes. The worst-case complexity of the algorithm is exponential in $m$ (to iterate over all subsets). However, since the algorithm stops after finding all the basis for the space $\text{vec}(\mathcal{X})$, it is unclear if the worst-case runtime occurs in practice.

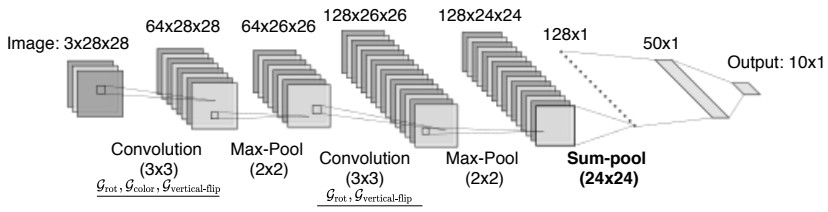

Figure 6: An example architecture of CG-invariant CNN architecture.

Moreover, the algorithm only needs to run once for a given collection of groups and the results can be reused in all experiments.

---

**Algorithm 1:** Procedure to construct basis for the subspaces $\mathcal{B}_M$ of Theorem 3.

---

**Input:** Left 1-eigenspaces of the Reynolds operator $\mathcal{W}_1, \mathcal{W}_2, \ldots, \mathcal{W}_m$ for groups
$\quad\quad \mathcal{G}_1, \mathcal{G}_2, \ldots, \mathcal{G}_m$ respectively.
**Result:** Basis for nonzero subspaces $\mathcal{B}_{M_1}, \mathcal{B}_{M_2}, \ldots, \mathcal{B}_{M_B}$, with $B \leq d_\mathcal{X}$ and
$\quad\quad M_i \subseteq \{1, \ldots, m\}$.
```
// Initialization
```
$l \leftarrow m$ ;
$\mathcal{C} \leftarrow \emptyset$ ;
$k \leftarrow 1$ ;                                     `// A counter for the subspaces.`
**while** $l \geq 0$ **do**
$\quad$ /\* $l$ will denote the size of subsets $M \subseteq \{1,\ldots,m\}$, denoting
$\quad\quad$ the level of invariance.                                     \*/
$\quad \mathcal{P}_l \leftarrow \{M : |M| = l,\ M \subseteq \{1, \ldots, m\}\}$ ;
$\quad$ **for** $M$ *in* $\mathcal{P}_l$ **do**
$\quad\quad$ **if** $M \neq \emptyset$ **then**
$\quad\quad\quad \widetilde{\mathcal{B}}_M \leftarrow \cap_{i \in M} \mathcal{W}_i$ ;          /\* Intersection of 1-eigenspaces.  \*/
$\quad\quad$ **else**
$\quad\quad\quad \widetilde{\mathcal{B}}_M \leftarrow \text{vec}(\mathcal{X})$ ;                /\* Used to find the subspace $\mathcal{B}_\emptyset$.  \*/
$\quad\quad$ **end**
$\quad\quad$ `// Direct sum of subspaces of supersets of M.`
$\quad\quad \mathcal{B}_{\supsetneq M} = \bigoplus_{N \supsetneq M} \mathcal{B}_N$ ;
$\quad\quad \mathcal{B}_M \leftarrow \text{orth}_{\mathcal{B}_{\supsetneq M}}(\widetilde{\mathcal{B}}_M)$ ;
$\quad\quad$ **if** $\mathcal{B}_M \neq \mathbf{0}$ **then**
$\quad\quad\quad M_k \leftarrow M$ ;             /\* Record current subspace to return \*/
$\quad\quad\quad k \leftarrow k + 1$ ;
$\quad\quad$ **end**
$\quad\quad \mathcal{C} \leftarrow \mathcal{C} \oplus \mathcal{B}_M$ ;
$\quad\quad$ **if** $dim(\mathcal{C}) = dim(vec(\mathcal{X}))$ **then**
$\quad\quad\quad$ break while ;             /\* Found basis for the entire space.  \*/
$\quad\quad$ **end**
$\quad$ **end**
$\quad l \leftarrow l - 1$
**end**
$B \leftarrow k - 1$ ;                                     /\* Number of subspaces.  \*/
**return** $\mathcal{B}_{M_1}, \mathcal{B}_{M_2}, \ldots, \mathcal{B}_{M_B}$ ;

---

# E ARCHITECTURES

## E.1 IMAGES

An example CG-invariant CNN architecture is depicted in Figure 6. Majority of the CNN architecture remains the same with the exception that the filters are obtained using the bases of the subspaces obtained in Theorem 3 for the given set of groups. Figure 5 shows example subspaces along with

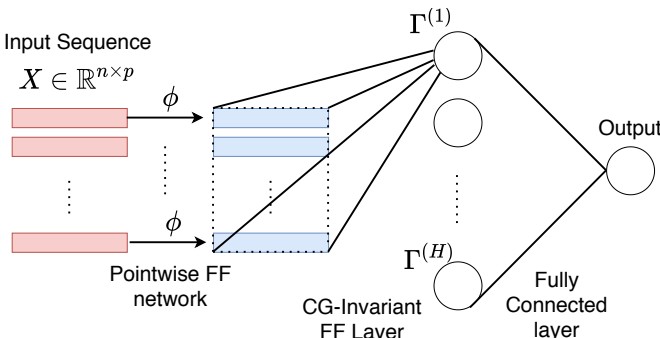

Figure 7: An example architecture of CG-invariant feedforward network.

their basis vectors when the groups are just $\mathcal{G}_{\text{rot}}$ and $\mathcal{G}_{\text{color}}$, and the kernel size is $3 \times 3$ applied over an input with 3 channels. One can similarly obtain these subspaces for other groups, different kernel sizes and different number of input channels. Then, the filter is obtained as a linear combination of these basis vectors, where the coefficients form the learnable parameters. The G-invariance of the filter then depends upon which of these coefficients are nonzero. Once the filter is obtained, it is convolved with the image or the feature maps. This will ensure that the model can be CG-invariant to transformations of *smaller patches* in the image if needed.

Max-pooling layers function in the standard way. After all the convolutional and max-pooling layers, we use a sum-pooling layer over the entire channel to ensure that the model can be invariant to the transformations (e.g., rotations) on the *whole image* if needed. Finally, any number of dense layers can be added after the sum-pooling layer.

In our experiments, we use the three groups $\mathcal{G}_{\text{rot}}$, $\mathcal{G}_{\text{color}}$ and $\mathcal{G}_{\text{vertical-flip}}$ to construct the subspaces for the filters of the first convolutional layer, but remove $\mathcal{G}_{\text{color}}$ in the further layers as we do not wish to be invariant to channel permutation after the first layer.

### E.2 Sequences

A CG-invariant architecture for sequences is depicted in Figure 7. Consider a sequence $\boldsymbol{X} = [\boldsymbol{x}_1, \dots, \boldsymbol{x}_n] \in \mathbb{R}^{p \times n}$ of length $n$ and groups $\mathcal{G}_1, \dots, \mathcal{G}_m$ as before. In the following discussion, we will assume that the groups are permutation groups over the sequence elements. However, one could also consider other groups over $\boldsymbol{X}$.

First, each element of the sequence is passed through a shared feedforward network $\phi$ that returns a representation $\boldsymbol{Z} \in \mathbb{R}^{p' \times n}$. Then, Theorem 3 finds the bases for $\mathcal{B}_M$, $M \subseteq \{1, \dots, m\}$ until all the $p'n$ basis vectors are found covering the space $\mathbb{R}^{p' \times n}$. The weight vectors for the $h$-th neuron of the CG-invariant layer is obtained as a linear combination of these basis vectors via the learnable parameters $\Omega$ (Equation (10)). Finally, any number of dense layers can be stacked after the CG-invariant layer for the final output.

## F Regularization

### F.1 Example

Figure 8 shows an example computation of the penalty in Equation (12). The example considers an image task with $m = 3$ groups: $\mathcal{G}_{\text{rot}}, \mathcal{G}_{\text{col}}, \mathcal{G}_{\text{vflip}}$. Each cell in the figure shows one subset $M \subseteq \{\text{rot}, \text{col}, \text{vflip}\}$. The subsets are arranged according to their levels of invariance, i.e., by the size of $|M|$. For example, the topmost cell $\{\text{rot}, \text{col}, \text{vflip}\}$ denotes the subspace with all the invariances whereas the bottommost cell $\emptyset$ denotes the subspace with no invariance.

The colors indicate the state of the parameters $\Omega$ at a single point in the optimization. The cells are colored green or red depending on whether the subspace is used or unused respectively, i.e., whether the parameters corresponding to the subspace are nonzero or not. The least invariant subspaces used

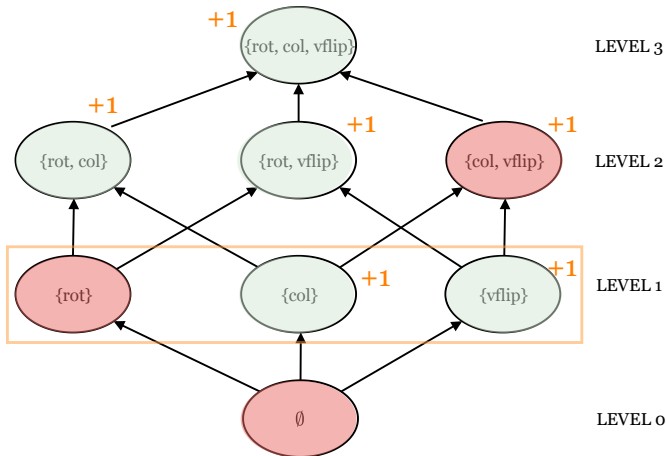

Figure 8: (Best viewed in color) Describing the computation of the penalty. The cells denote different subsets $M \subseteq \{\text{rot}, \text{col}, \text{vflip}\}$. Red colored cells denote that the parameters corresponding to these subspaces are zero (i.e., the subspaces are unused) and the green colored cells denote otherwise (i.e., the subspaces are used). In this example, the least invariant subspaces used are in Level 1. The penalty counts all the subspaces (used or unused) that are in higher levels (i.e., with $|M| > 1$) and adds it to the number of subspaces of the same level that are used.

at this point are in **Level 1** (i.e., invariant to a single group). The penalty counts **(a)** all subspaces with higher levels of invariance irrespective of whether the subspace is used or not, and **(b)** counts all the used subspaces with the same level of invariance. The former penalizes the use of subspaces lower in the partial order and ensures that subspaces with higher levels of invariance are used. The latter approximates the effort to reach a higher level of invariance.

## F.2 DIFFERENTIABLE APPROXIMATION

Recall that the regularization penalty $R(\mathbf{\Omega})$ in Equation (12) is given by,

$$R(\mathbf{\Omega}) = f_l(\mathbf{\Omega}) := |\{M_i : |M_i| > l, \ 1 \le i \le B\}| + \sum_{\substack{i:|M_i|=l \\ 1 \le i \le B}} \mathbf{1}\{\|\boldsymbol{\omega}_{M_i,\cdot}\|_2^2 > 0\}, \qquad (16)$$

where $l = \min\{|M_i| \cdot \mathbf{1}\{\|\boldsymbol{\omega}_{M_i,\cdot}\|_2^2 > 0\}, \ 1 \le i \le B\}$.

$R(\mathbf{\Omega})$ is clearly discrete but can be approximated by a differentiable formula. First, we replace the indicator function $\mathbf{1}\{z > 0\}$ in Equation (16) with the approximation $\tilde{\mathbf{1}}\{z > 0\} = \tau z / (\tau z + 1)$, where $\tau \ge 1$ is a temperature hyperparameter.

Then, in order to obtain $R(\mathbf{\Omega}) = f_l(\mathbf{\Omega})$ for the minimum $l$ defined in Equation (16), we use the following recursion: $R(\mathbf{\Omega}) = R_m(\mathbf{\Omega})$, and

$$R_l(\mathbf{\Omega}) = (1 - \beta_l(\mathbf{\Omega})) \cdot R_{l-1}(\mathbf{\Omega}) + f_l(\mathbf{\Omega})\beta_l(\mathbf{\Omega}) \quad l = 1, \dots, m,$$

with the base case $R_0(\mathbf{\Omega}) = 0$, and $\beta_l(\mathbf{\Omega}) = \tilde{\mathbf{1}}\{\sum_{N_i:|N_i|=l, \ 1 \le i \le B} \|\boldsymbol{\omega}_{N_i,\cdot}\|_2^2 > 0\}$. $\beta_l(\mathbf{\Omega})$ is approximately one if at least one neuron $h$ has nonzero $\boldsymbol{\omega}_{N_i,h}$ parameters for some $N_i \subseteq \{1, \dots, m\}$ of size $l$ (i.e., with $l$ groups). Then the recursion finds $f_l(\mathbf{\Omega})$ with $l$ defined as the size of the least invariant subspace used.

## F.3 LIMITATION OF $R(\Omega)$

As explained in Section 4.2, there could be overgroups (out of the total $2^m$ groups considered) with different levels of invariance, but penalized similarly by Equation (12). This scenario arises only in cases when Theorem 3 does not construct subspace basis for all the $2^m$ overgroups, i.e., the basis for $\text{vec}(\mathcal{X})$ is found prior to that. In this section, we provide such an example scenario with sequence inputs and the transposition groups considered in Section 6.

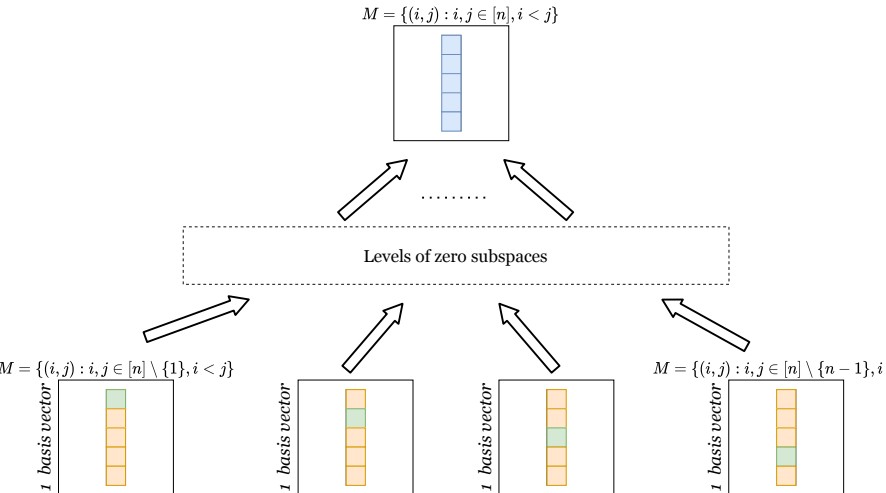

Figure 9: (Best viewed in color) The subspaces $\mathcal{B}_M$ for different $M \subseteq \{(i,j)\}_{1 \leq i < j \leq n}$ indexing the $m = \binom{n}{2}$ transposition groups $\mathcal{G}_{i,j}$ over sequences of length $n = 5$ and dimension $d = 1$. Each of the subspaces $\mathcal{B}_M$ is of dimension 1. For each basis vector shown above, elements sharing the same color have the same value. At the topmost level, we have the subspace with most invariance, i.e., **invariant to the full permutation group** $\mathbb{S}_n$. Following many levels with empty subspaces, we have subspaces $\mathcal{B}_{M_p}$ for $M_p = \{(i,j) \mid i,j \in [n] \setminus \{p\}, i < j\}$, where $[n] = \{1, \ldots, n\}$. In other words, the subspace $\mathcal{B}_{M_p}$ **is invariant to all transpositions except those that move index** $p$. Note that we have covered the entire space $\mathbb{R}^n$ with these $n$ independent subspaces of dimension 1.

Let $X \in \mathcal{X} = \mathbb{R}^n$ be a 1-dimensional sequence of length $n$. The transposition groups are $\{\mathcal{G}_{i,j}\}_{1 \leq i < j \leq n}$, where $\mathcal{G}_{i,j} = \{T_{\text{identity}}, T_{i,j}\}$ and $T_{i,j}$ swaps positions $i$ and $j$ in the sequence. Given these $m = \binom{n}{2}$ groups, we can use Lemmas 1 and 2, and Theorem 3 to find the invariant subspaces $\mathcal{B}_M$ for subsets $M \subseteq \{(i,j) \mid 1 \leq i < j \leq n\}$ indexing the transposition groups. The basis vectors for these subspaces constructed for sequence length $n = 5$ are visualized in Figure 9.

There are $n$ 1-dimensional subspaces. Let the vectors $\boldsymbol{b}_{\setminus \emptyset}, \boldsymbol{b}_{\setminus \{1\}} \ldots \boldsymbol{b}_{\setminus \{n-1\}}$ denote these $n$ basis vectors. The notation $\setminus A$ means that the vector has the same value for all positions $k \in \{1, \ldots, n\} \setminus A$ (cf. Figure 9). Let $n = 5$ and note that any weight vector $\boldsymbol{\omega} \in \mathbb{R}^5$ can be written as,

$$\boldsymbol{\omega} = \alpha_1 \boldsymbol{b}_{\setminus \emptyset} + \alpha_2 \boldsymbol{b}_{\setminus \{1\}} + \alpha_3 \boldsymbol{b}_{\setminus \{2\}} + \alpha_4 \boldsymbol{b}_{\setminus \{3\}} + \alpha_5 \boldsymbol{b}_{\setminus \{4\}} . \tag{17}$$

where $\boldsymbol{\alpha} \in \mathbb{R}^5$.

Let $\boldsymbol{\alpha}' = (1, 0, 1, 0, 1)^T$. From a quick read of Figure 9, we see that the weight $\boldsymbol{\omega}'$ obtained by substituting $\boldsymbol{\alpha}'$ in Equation (17) is such that $\omega_1' = \omega_3' = \omega_5'$ and $\omega_2' = w_4'$. For any input $\boldsymbol{x} \in \mathbb{R}^5$, the neuron $\sigma(\boldsymbol{\omega}'^T \boldsymbol{x} + b)$ is invariant to any permutation of $x_1, x_3$ and $x_5$, and, transposition of $x_2$ and $x_4$. The penalty $R(\boldsymbol{\omega}') = 3$ as there are 2 subspaces used at the lowest level and there is 1 subspace above the lowest level (see Equation (12)).

Now let $\boldsymbol{\alpha}'' = (1, 0, 1, 0, 1.5)^T$. The weight $\boldsymbol{\omega}''$ obtained by substituting $\boldsymbol{\alpha}''$ in Equation (17) is such that $\omega_1'' = \omega_3'' = \omega_5''$ but $\omega_2'' \neq \omega_4''$. For input $\boldsymbol{x} \in \mathbb{R}^5$, the neuron $\sigma(\boldsymbol{\omega}''^T \boldsymbol{x} + b)$ is invariant to any permutation of $x_1'', x_3''$ and $x_5''$, but *sensitive* to the transposition of $x_2$ and $x_4$. The penalty $R(\boldsymbol{\omega}'') = 3$ as the same subspaces are used as before.

In the first case, with all the parameters being equal (especially $\alpha_3' = \alpha_5'$), $\boldsymbol{\omega}'$ lies in a smaller (more invariant) subspace of $\text{span}(\boldsymbol{b}_{\setminus \emptyset}, \boldsymbol{b}_{\setminus \{2\}}, \boldsymbol{b}_{\setminus \{4\}})$. In the second case, since $\alpha_3'' \neq \alpha_5''$, the same does not hold for $\boldsymbol{\omega}''$. The penalty $R(\cdot)$, which only counts the subspaces used (in this case, $\boldsymbol{b}_{\setminus \emptyset}, \boldsymbol{b}_{\setminus \{2\}}$ and $\boldsymbol{b}_{\setminus \{4\}}$), is unable to distinguish between these two weight vectors $\boldsymbol{\omega}'$ and $\boldsymbol{\omega}''$, one clearly more invariant than the other.

In this specific case with transposition groups over sequences, one could add another penalty term that regularizes the parameters $\alpha_i$ to share the same value (e.g., entropy regularization of the parameters). We leave further investigation into the general scenario with other groups for future work.

# G    DATASETS AND EMPIRICAL RESULTS

## G.1    IMAGES

**Datasets.**    We consider the standard MNIST dataset and its subset MNIST-34 that contains only the digits 3 & 4 alone. We chose to experiment on the MNIST-34 dataset since it does not have digits that can be confused with a rotation transformation (e.g., 6 and 9) or are invariant to some rotations (e.g., 0, 1 and 8), thus avoiding any confounding factors while testing our hypothesis. We also experiment on the full MNIST dataset to depict the scenario when the data does contain these contradictions. First, we modify all the images in the dataset to have three RGB color channels and color each digit *red* initially, i.e., all active pixels in the digit are set to (255, 0, 0). We sample $X^{(\mathrm{hid})}$ from this dataset with the target digit as its original label.

**Groups.**    We consider $m = 3$ linear automorphism groups on images: the rotation group $\mathcal{G}_{\mathrm{rot}} = \{T^{(0°)}, T^{(90°)}, T^{(180°)}, T^{(270°)}\}$ that rotates the entire image by multiples of $90°$, the channel-permutation group $\mathcal{G}_{\mathrm{color}} = \{T^{\alpha}\}_{\alpha \in \mathbb{S}_3}$ that permutes the three RGB channels of the image, and the vertical flip group $\mathcal{G}_{\mathrm{vertical\text{-}flip}} = \{T^{(0)}, T^{(v)}\}$ that vertically flips the image.

**Tasks.**    For both MNIST and MNIST-34 datasets, we consider 4 classification tasks where each task represents the case when the target $Y$ is invariant to a different subset of $\{\mathcal{G}_{\mathrm{rot}}, \mathcal{G}_{\mathrm{vertical\text{-}flip}}, \mathcal{G}_{\mathrm{color}}\}$, i.e., invariant to all three groups, to two, to one, invariant to none (and sensitive to the remaining groups). We consider the following subsets $\mathcal{I}$: **i)** {rot, color, vertical-flip}, **ii)** {rot, vertical-flip}, **iii)** {color}, **iv)** $\emptyset$, and generate $\mathcal{G}_{\mathcal{I}} = \langle \cup_{i \in \mathcal{I}} \mathcal{G}_i \rangle$ as the *join* of the respective groups. Setting $\mathcal{D} = \{\mathrm{rot}, \mathrm{color}, \mathrm{vertical\text{-}flip}\} \setminus \mathcal{I}$, we generate $\mathcal{G}_{\mathcal{D}} = \langle \cup_{j \in \mathcal{D}} \mathcal{G}_j \rangle$ from the join of groups in the complement set (our choices ensure that $\mathcal{G}_{\mathcal{I}} \trianglelefteq \mathcal{G}_{\mathcal{D} \cup \mathcal{I}}$, thus satisfying the conditions of Theorem 2).

*Training data:* $X^{(\mathrm{hid})}$ is the canonically ordered (standard) image in the MNIST datasets. Recall that the training data is sampled via an economical data generation process. Thus the training data consists only of images under transformations that have an effect on the label, i.e., transformations from $\mathcal{G}_{\mathcal{D}}$.

Recall from Equation (2) that the observed input is obtained as $X = T_{U_{\mathcal{I}}, U_{\mathcal{D}}} \circ X^{(\mathrm{hid})}$, a transformation of the canonical input $X^{(\mathrm{hid})}$. Since $\mathcal{G}_{\mathcal{I}} \trianglelefteq \mathcal{G}_{\mathcal{D}}$ (by construction), we have that any $T_{U_{\mathcal{I}}, U_{\mathcal{D}}} = T_{U'_{\mathcal{I}}|U_{\mathcal{D}}} \circ T_{U_{\mathcal{D}}}$, i.e., the transformation can be decomposed into one transformation from $\mathcal{G}_{\mathcal{D}}$ followed by another transformation from $\mathcal{G}_{\mathcal{I}}$. $U'_{\mathcal{I}} \mid U_{\mathcal{D}}$ in the subscript indicates that the transformation $T_{U'_{\mathcal{I}}|U_{\mathcal{D}}} \in \mathcal{G}_{\mathcal{I}}$ also depends on $U_{\mathcal{D}}$. Under the assumption of economic sampling of training data, in all our experiments we sample a single value for $T_{U'_{\mathcal{I}}|U_{\mathcal{D}}} \in \mathcal{G}_{\mathcal{I}}$: we simply use $T_{U'_{\mathcal{I}}|U_{\mathcal{D}}} = T_{\mathrm{identity}}$ (one could consider any other transformation in $\mathcal{G}_{\mathcal{I}}$ as well).

In conclusion, we obtain the observed image $X$ in the training data by applying a random transformation from $\mathcal{G}_{\mathcal{D}}$ to $X^{(\mathrm{hid})}$ and then applying a constant transformation (e.g., $T_{\mathrm{identity}}$) from $\mathcal{G}_{\mathcal{I}}$ to the result. The task is to predict the original label of the image (i.e., the digit) and the transformation $T_{U_{\mathcal{D}}}$ that was applied to obtain $X$ (recall from Equation (3) that $Y$ is a function of both $X^{(\mathrm{hid})}$ and $U_{\mathcal{D}}$).

For instance, if $\mathcal{G}_{\mathcal{I}} = \mathcal{G}_{\mathrm{rot, vertical\text{-}flip}}$ and $\mathcal{G}_{\mathcal{D}} = \mathcal{G}_{\mathrm{color}}$, then the training data consists of **upright** and **unflipped** images (as $T_{U'_{\mathcal{I}}|U_{\mathcal{D}}}$ is chosen to be identity transformation) with different permutations of the color channels (since random transformations are sampled from $\mathcal{G}_{\mathcal{D}}$) resulting in digits with different colors. Then, the task is to predict the digit and its color.

*Extrapolation task:* The extrapolated test data consists of samples from the coupled random variable $X_{U_{\mathcal{I}} \leftarrow \widetilde{U}_{\mathcal{I}}}$ (Definition 1). Unlike the training data that was economically sampled (i.e., with a single transformation from $\mathcal{G}_{\mathcal{I}}$), the extrapolated test data is obtained via the full range of transformations in $\mathcal{G}_{\mathcal{I}}$. Recall from Definition 1 that $X_{U_{\mathcal{I}} \leftarrow \widetilde{U}_{\mathcal{I}}} = T_{\widetilde{U}_{\mathcal{I}}, U_{\mathcal{D}}} \circ X^{(\mathrm{hid})}$. As before, we decompose $T_{\widetilde{U}_{\mathcal{I}}, U_{\mathcal{D}}} = T_{\widetilde{U}'_{\mathcal{I}}|U_{\mathcal{D}}} \circ T_{U_{\mathcal{D}}}$. However, there is no economic sampling for the test data: $T_{\widetilde{U}'_{\mathcal{I}}|U_{\mathcal{D}}}$ and $T_{U_{\mathcal{D}}}$ are sampled randomly from $\mathcal{G}_{\mathcal{I}}$ and $\mathcal{G}_{\mathcal{D}}$ respectively.

Table 2: **(MNIST-34.)** Validation and Extrapolation test accuracies (%) with 95% confidence intervals for different CG-regularization strength $\lambda$ in Equation (11). $\lambda$ is chosen only based on the validation accuracy: maximum $\lambda$ with validation accuracy within 5% of the best validation accuracy (**bold** values indicate the performance of this choice of $\lambda$).

| | | $\mathcal{I} \subseteq \{\text{rot, color, vflip}\}$ | | | | | | | |
| | | rot,color,vflip | | rot,vflip | | color | | $\emptyset$ | |
| Model | $\lambda$ | Val. acc (%) | Test acc (%) | Val. acc (%) | Test acc (%) | Val. acc (%) | Test acc (%) | Val. acc (%) | Test acc (%) |
|---|---|---|---|---|---|---|---|---|---|
| VGG + CG-reg | 0.0 | 99.94 ( 0.17) | 49.51 ( 2.36) | 99.83 ( 0.09) | 43.57 ( 3.69) | 97.15 ( 0.27) | 15.71 ( 5.46) | 95.65 ( 0.39) | 96.30 ( 0.68) |
| | 0.1 | 99.92 ( 0.09) | 78.72 (25.75) | 99.86 ( 0.21) | 73.98 (16.33) | 96.48 ( 0.77) | 96.27 ( 1.01) | 94.95 ( 0.54) | 95.56 ( 0.61) |
| | 1.0 | 99.71 ( 0.22) | 85.42 (29.66) | 99.77 ( 0.25) | 75.64 (20.52) | 96.12 ( 1.26) | 96.20 ( 1.11) | 94.01 ( 1.51) | 94.42 ( 1.38) |
| | 2.0 | 99.55 ( 0.33) | 94.88 ( 0.84) | 99.56 ( 0.66) | 82.59 (28.92) | 95.23 ( 0.99) | 95.61 ( 1.75) | **94.05 ( 1.50)** | **94.49 ( 1.49)** |
| | 10.0 | **99.00 ( 1.18)** | **94.89 ( 7.49)** | **98.43 ( 2.00)** | **95.78 ( 7.11)** | **93.34 ( 8.42)** | **94.16 ( 6.43)** | 88.42 (19.36) | 88.68 (20.13) |

Table 3: **(MNIST.)** Validation and Extrapolation test accuracies (%) with 95% confidence intervals for different CG-regularization strength $\lambda$ in Equation (11). $\lambda$ is chosen only based on the validation accuracy: maximum $\lambda$ with validation accuracy within 5% of the best validation accuracy (**bold** values indicate the performance of this choice of $\lambda$).

| | | $\mathcal{I} \subseteq \{\text{rot, color, vflip}\}$ | | | | | | | |
| | | rot,color,vflip | | rot,vflip | | color | | $\emptyset$ | |
| Model | $\lambda$ | Val. acc (%) | Test acc (%) | Val. acc (%) | Test acc (%) | Val. acc (%) | Test acc (%) | Val. acc (%) | Test acc (%) |
|---|---|---|---|---|---|---|---|---|---|
| VGG + CG-reg | 0.0 | 99.17 ( 0.17) | 11.93 ( 1.87) | 98.80 ( 0.14) | 25.81 ( 0.92) | 91.50 ( 0.35) | 4.19 ( 2.12) | 91.80 ( 0.60) | 91.60 ( 0.32) |
| | 0.1 | 98.62 ( 0.05) | 29.48 ( 0.98) | 98.34 ( 0.17) | 30.12 ( 4.05) | 90.11 ( 0.58) | 87.23 ( 3.68) | 88.30 ( 1.21) | 88.48 ( 1.17) |
| | 1.0 | 98.49 ( 0.24) | 44.23 (15.45) | 98.29 ( 0.21) | 40.13 ( 4.83) | 90.24 ( 0.46) | 90.23 ( 0.99) | 88.76 ( 1.28) | 88.65 ( 1.30) |
| | 2.0 | 98.45 ( 0.13) | 55.24 ( 2.29) | 98.34 ( 0.34) | 47.14 (15.17) | 89.98 ( 0.24) | 89.71 ( 0.89) | 89.50 ( 1.35) | 89.45 ( 1.43) |
| | 10.0 | **97.76 ( 0.74)** | **64.99 ( 2.76)** | **95.21 ( 6.55)** | **62.68 ( 6.02)** | **88.80 ( 2.11)** | **88.69 ( 2.11)** | **90.54 ( 1.04)** | **90.89 ( 0.43)** |

In conclusion, we obtain the observed image $X$ in the test data by applying a random transformation from $\mathcal{G}_{\mathcal{D}}$ to $X^{(\text{hid})}$ and then applying a random transformation from $\mathcal{G}_{\mathcal{I}}$ to the result. The task is the same as in the training data: to predict the original label of the image (i.e., the digit) and the transformation $T_{U_{\mathcal{D}}}$ that was applied to obtain $X$. Note that the label does not depend on the transformation $T_{\widetilde{U}'_{\mathcal{I}}|U_{\mathcal{D}}} \in \mathcal{G}_{\mathcal{I}}$ that was applied.

Once again, if $\mathcal{G}_{\mathcal{I}} = \mathcal{G}_{\text{rot, vertical-flip}}$ and $\mathcal{G}_{\mathcal{D}} = \mathcal{G}_{\text{color}}$, then the extrapolated test data consists of images randomly **rotated**, **flipped** and **channel permuted**, while the task is the same: predict the digit and its color.

In order to evaluate the models, we use 5-fold cross-validation procedure as follows. We divide the training and test datasets that are pre-split in MNIST and MNIST-34 datasets into 5 folds each. We use the above procedure to transform the training data and the test data. Then in each iteration $i$ of the cross-validation procedure, we leave out $i$-th fold of the transformed training data and $i$-th fold of the extrapolated test data. Further, we use 20% of the training data as validation data for hyperparameter tuning and early stopping.

**Baselines and Architecture.** For all methods, we use a VGG architecture (Simonyan and Zisserman, 2014) with 8 convolutional layers each having 128 channels except the first layer which has 64 channels. All convolutional layers have a receptive field of size $3 \times 3$, stride 1 and padding 1. A max-pooling layer is added after every two convolutional layers. Two feedforward layers at the end give the final output. We compare our approach with the standard CNNs and Group-equivariant CNNs (G-CNNs) (Cohen and Welling, 2016) with the $p4m$ group. We modified G-CNN such that it has invariances to all the 3 groups strictly enforced via a) coset-pooling (Cohen and Welling, 2016) after each layer and b) adding together the 3 input RGB channels. For our approach, we replace the standard convolutional layer in the VGG architecture by CG-invariant layers with bases constructed from $\mathcal{G}_{\text{rot}}, \mathcal{G}_{\text{color}}$ and $\mathcal{G}_{\text{vertical-flip}}$. An example architecture with only 2 convolutional layers is shown in Figure 6.

We optimize all models using SGD with momentum with learning rate in $\{10^{-2}, 10^{-3}, 10^{-4}\}$ and a batch size of 64. We use early stopping on validation loss to select the best model. Further, we use validation loss to select the best set of hyperparameters for each model. We choose the maximum value of $\lambda$ with validation accuracy within a 5% threshold of the maximum validation accuracy

Table 4: Sequence tasks. The first column defines the target $Y$ for a given sequence $(X_i)_{i=1}^{10}$. The second column denotes $\mathcal{G}_\mathcal{I}$, the group of transformations to which $Y$ is invariant. Recall that $\mathcal{G}_\mathcal{I}$ is constructed as the join of a subset of $\binom{10}{2}$ transposition groups.

| Target $Y$ | $\mathcal{G}_\mathcal{I}$ | $\mathcal{G}_\mathcal{D}$ |
|---|---|---|
| $Y_{\text{task-1}} = \sum_{i=1}^{10} X_i$ | $\langle \{\mathcal{G}_{i,j}\}_{1 \leq i < j \leq n} \rangle$ | $\{\text{Id}\}$ |
| $Y_{\text{task-2}} = \sum_{i=2}^{10} X_i$ | $\langle \{\mathcal{G}_{i,j}\}_{2 \leq i < j \leq n} \rangle$ | $\{\text{Id}\}$ |
| $Y_{\text{task-3}} = \sum_{i=1}^{5} (X_{2i} - X_{2i-1})$ | $\langle \{\mathcal{G}_{i,i+2k}\}_{1 \leq i < i+2k \leq n} \rangle$ | $\{\text{Id}\}$ |
| $Y_{\text{task-4}} = \sum_{i=1}^{10} \prod_{j=1}^{i} \mathbf{1}(X_j \geq 20)$ | $\{\text{Id}\}$ | $\langle \{\mathcal{G}_{i,j}\}_{1 \leq i < j \leq n} \rangle$ |

obtained from any value of $\lambda$. Tables 2 and 3 show the effect of regularization strength on the performance of the model. We observe that $\lambda = 10$ performs considerably well across all tasks.

## G.2 SEQUENCES

**Datasets.** For sequence tasks, we generate $X^{(\text{hid})} = (X_i)_{i=1}^{10}$ as a sequence of $n = 10$ canonically ordered integers uniformly sampled with replacement from a fixed vocabulary set $\{1, \ldots, 99\}$. The canonical ordering is fixed for a given set of integers sampled: the corresponding sequence $X^{(\text{hid})}$ is always either in an increasing order or a decreasing order.

**Groups.** We consider $m = \binom{n}{2}$ permutation groups for all the pair-wise permutations: $\mathcal{G}_{1,2}, \mathcal{G}_{2,3}, \mathcal{G}_{1,3}, \ldots, \mathcal{G}_{n-1,n}$, where $\mathcal{G}_{i,j} := \{T_{\text{identity}}, T_{i,j}\}$ and $T_{i,j}$ swaps positions $i$ and $j$ in the sequence. For $\mathcal{I} \subseteq \{(1,2), (2,3), (1,3), \ldots, (n-1,n)\}$, $\mathcal{G}_\mathcal{I}$ is defined as before as the join $\langle \cup_{(i,j) \in \mathcal{I}} \mathcal{G}_{i,j} \rangle$. We choose 4 different subsets $\mathcal{I}$ of the given $m$ groups indicated by the second column of Table 4. For our choices of $\mathcal{I} \neq \emptyset$, we set $\mathcal{D} = \emptyset$ to ensure that $\mathcal{G}_\mathcal{I} \trianglelefteq \mathcal{G}_{\mathcal{D} \cup \mathcal{I}}$, i.e., $\mathcal{G}_\mathcal{I}$ is a normal subgroup of $\mathcal{G}_{\mathcal{D} \cup \mathcal{I}}$.

**Tasks.** The label for the sequence $X^{(\text{hid})}$ is obtained by applying an arithmetic function to $X^{(\text{hid})}$ that is invariant to the chosen group $\mathcal{G}_\mathcal{I}$. The arithmetic functions are given in the first column of Table 4. $Y_{\text{task-1}}$ is invariant to any permutation of the input elements $X_i, 1 \leq i \leq n$. $Y_{\text{task-2}}$ is invariant to any permutation of input elements $X_i$ with indices $i > 1$ but sensitive to permutations that move $X_1$. $Y_{\text{task-3}}$ is invariant to permutations that move elements at even indices to even indices and elements at odd indices to odd indices respectively. Finally, $Y_{\text{task-4}}$ is sensitive to all permutations (i.e., no invariance).

*Training data:* Recall that $X^{(\text{hid})}$ is in a sorted order. Since the training data is sampled economically, it consists only of sequences under transformations that have an effect on the label, i.e., transformations from $\mathcal{G}_\mathcal{D}$. The observed input is obtained as $X = T_{U_\mathcal{I}, U_\mathcal{D}} \circ X^{(\text{hid})}$, a transformation of the sorted input $X^{(\text{hid})}$. Since $\mathcal{G}_\mathcal{I} \trianglelefteq \mathcal{G}_\mathcal{D}$ (by construction), we have that any $T_{U_\mathcal{I}, U_\mathcal{D}} = T_{U'_\mathcal{I} | U_\mathcal{D}} \circ T_{U_\mathcal{D}}$, i.e., the transformation can be decomposed into one transformation from $\mathcal{G}_\mathcal{D}$ followed by another transformation from $\mathcal{G}_\mathcal{I}$. $U'_\mathcal{I} \mid U_\mathcal{D}$ in the subscript indicates that the transformation $T_{U'_\mathcal{I} | U_\mathcal{D}} \in \mathcal{G}_\mathcal{I}$ also depends on $U_\mathcal{D}$. Under the assumption of economic sampling of training data, in all our experiments we sample a single value for $T_{U'_\mathcal{I} | U_\mathcal{D}} \in \mathcal{G}_\mathcal{I}$: we simply use $T_{U'_\mathcal{I} | U_\mathcal{D}} = T_{\text{identity}}$.

In conclusion, we obtain the observed sequence $X$ in the training data by applying a random transformation $T_{U_\mathcal{D}} \in \mathcal{G}_\mathcal{D}$ to $X^{(\text{hid})}$ and then applying a constant transformation (e.g., $T_{\text{identity}}$) from $\mathcal{G}_\mathcal{I}$ to the result. The target $Y$ is computed by applying the arithmetic function corresponding to the task (see Table 4) to $T_{U_\mathcal{D}} \circ X^{(\text{hid})}$ (recall from Equation (3) that $Y$ is a function of both $X^{(\text{hid})}$ and $U_\mathcal{D}$).

*Extrapolation task:* The extrapolated test data consists of samples from the coupled random variable $X_{U_\mathcal{I} \leftarrow \tilde{U}_\mathcal{I}}$ (Definition 1). Unlike the training data that was economically sampled (i.e., with a single transformation from $\mathcal{G}_\mathcal{I}$), the extrapolated test data is obtained via the full range of transformations in $\mathcal{G}_\mathcal{I}$. Recall from Definition 1 that $X_{U_\mathcal{I} \leftarrow \tilde{U}_\mathcal{I}} = T_{\tilde{U}_\mathcal{I}, U_\mathcal{D}} \circ X^{(\text{hid})}$. As before, we decompose $T_{\tilde{U}_\mathcal{I}, U_\mathcal{D}} = T_{\tilde{U}'_\mathcal{I} | U_\mathcal{D}} \circ T_{U_\mathcal{D}}$. However, there is no economic sampling for the test data: $T_{\tilde{U}'_\mathcal{I} | U_\mathcal{D}}$ and $T_{U_\mathcal{D}}$ are sampled randomly from $\mathcal{G}_\mathcal{I}$ and $\mathcal{G}_\mathcal{D}$ respectively.

Table 5: (**Sequence tasks**) Extrapolation test accuracies (%) with 95% confidence intervals for all the models (**bold** means $p < 0.05$ significant). The standard sequence models cannot extrapolate when $\mathcal{I} \neq \emptyset$ whereas the forced G-invariant models cannot unlearn the invariances and fail when $\mathcal{I} \subsetneq \{1, \dots, m\}$.

| Model | $\langle\{\mathcal{G}_{i,j}\}_{1\leq i<j\leq n}\rangle$ | $\mathcal{G}_\mathcal{I}$ $\langle\{\mathcal{G}_{i,j}\}_{2\leq i<j\leq n}\rangle$ | $\langle\{\mathcal{G}_{i,i+2k}\}_{1\leq i<i+2k\leq n}\rangle$ | {Id} |
|---|---|---|---|---|
| DeepSets (Zaheer et al., 2017) | **100.00 ( 0.00)** | 2.36 ( 2.37) | 0.97 ( 0.60) | 16.12 ( 8.21) |
| Janossy pooling (Murphy et al., 2018) | 96.64 ( 3.13) | 9.55 ( 1.61) | 0.78 ( 0.52) | 21.22 ( 2.94) |
| Set Transformer (Lee et al., 2019) | **99.57 ( 0.33)** | 10.68 ( 1.49) | 0.75 ( 0.28) | 23.38 ( 1.88) |
| Transformer (Vaswani et al., 2017) | 20.26 (32.08) | 12.15 (16.05) | 0.85 ( 0.37) | **100.00 ( 0.00)** |
| GRU (Cho et al., 2014) | 0.48 ( 0.48) | 0.47 ( 0.38) | 0.90 ( 0.77) | **99.41 ( 1.58)** |
| FF + CG-reg. (ours) | **100.00 ( 0.00)** | **42.08 (18.99)** | **71.85 (26.61)** | 95.70 ( 3.05) |

Table 6: (**Sequence Tasks**) Validation and Extrapolation test accuracies (%) with 95% confidence intervals for different CG-regularization strength $\lambda$ in Equation (11). $\lambda$ is chosen only based on the validation accuracy: maximum $\lambda$ with validation accuracy within 5% of the best validation accuracy (**bold** values indicate the performance of this choice of $\lambda$).

| Model | $\lambda$ | $\langle\{\mathcal{G}_{i,j}\}_{1\leq i<j\leq n}\rangle$ Val. acc (%) | Test acc (%) | $\mathcal{G}_\mathcal{I}$ $\langle\{\mathcal{G}_{i,j}\}_{2\leq i<j\leq n}\rangle$ Val. acc (%) | Test acc (%) | $\langle\{\mathcal{G}_{i,i+2k}\}_{1\leq i<i+2k\leq n}\rangle$ Val. acc (%) | Test acc (%) | {Id} Val. acc (%) | Test acc (%) |
|---|---|---|---|---|---|---|---|---|---|
| FF + CG-reg. | 0.0 | 80.80 (84.15) | 54.34 (87.19) | 100.00 ( 0.00) | 80.36 (74.04) | 99.95 ( 0.10) | 22.78 (14.52) | 99.92 ( 0.27) | 99.86 ( 0.15) |
| | 0.1 | 80.83 (84.04) | 59.18 (91.12) | 100.00 ( 0.00) | 65.13 (80.43) | 99.99 ( 0.05) | 60.66 (49.83) | 99.95 ( 0.10) | 99.98 ( 0.05) |
| | 1.0 | 80.72 (84.48) | 80.03 (87.52) | 99.04 ( 4.22) | 61.81 (66.24) | 100.00 ( 0.00) | 68.34 (36.98) | 99.81 ( 0.55) | 99.76 ( 0.65) |
| | 2.0 | 82.56 (72.85) | 63.16 (99.44) | 100.00 ( 0.00) | 77.97 (48.87) | 99.99 ( 0.05) | 69.20 (31.40) | 99.46 ( 0.53) | 99.37 ( 0.53) |
| | 10.0 | 80.97 (74.10) | 62.83 (100.17) | **98.14 ( 2.71)** | **42.08 (18.99)** | 100.00 ( 0.00) | 71.85 (26.61) | **95.56 ( 3.34)** | **95.70 ( 3.05)** |
| | 100.0 | **100.00 ( 0.00)** | **100.00 ( 0.00)** | 15.65 ( 3.63) | 2.29 ( 0.96) | 93.42 (14.90) | 27.64 (24.85) | 65.92 (10.38) | 65.42 (10.30) |

In conclusion, we obtain the observed sequence $X$ in the test data by applying a random transformation $T_{U_\mathcal{D}} \in \mathcal{G}_\mathcal{D}$ to $X^{(\mathrm{hid})}$ and then applying a random transformation from $\mathcal{G}_\mathcal{I}$ to the result. The target $Y$ is computed in a similar fashion as in the training data by applying the appropriate arithmetic function to $T_{U_\mathcal{D}} \circ X^{(\mathrm{hid})}$. Note that $Y$ is invariant to $\mathcal{G}_\mathcal{I}$.

Example: Consider the the first row of Table 4 with $\mathcal{I} = \{(i,j)\}_{1\leq i<j\leq n}$, i.e., it contains all the $m = \binom{n}{2}$ groups. Then, the group $\mathcal{G}_\mathcal{I}$ is simply the full permutation group over $n$ elements. The target is defined as the sum of elements (which is fully permutation-invariant). The sequences in the training data are always sorted (because of the economic sampling of training data), whereas the sequences in test data have arbitrarily different permutations (by sampling random transformations from $\mathcal{G}_\mathcal{I}$). The task is simply to compute the sum of the elements of the sequence.

Sizes of the training data and the extrapolated test data are fixed at 8000 and 2000 respectively. We repeat all the experiments for 5 different random seeds.

**Baselines and Architecture.** We compare our approach with a) standard sequence models, specifically Transformers (Vaswani et al., 2017) and GRUs (Cho et al., 2014), and b) forced permutation-invariant set models, specifically DeepSets (Zaheer et al., 2017), SetTransformer (Lee et al., 2019) and Janossy Pooling (Murphy et al., 2018). An example of the proposed CG-invariant feedforward architecture is depicted in Figure 7.

We optimize all models using Adam (Kingma and Ba, 2014) with an initial learning rate in $\{10^{-2}, 10^{-3}, 10^{-4}\}$ and a batch size of 128. We use validation loss for early-stopping and to select the best hyperparameters for all models. Once again, we choose the best value for the CG-regularization strength $\lambda$ by choosing the maximum value of $\lambda$ with validation accuracy within 5% of the maximum validation accuracy obtained from any $\lambda$. Table 6 shows the effect of regularization strength on the performance of the model. We observe that although $\lambda = 10$ performs comparably to the rest in validation accuracy and is chosen consistently, it does not achieve the best possible extrapolation accuracy.

Table 5 shows the complete set of results for all the models. The table clearly shows the issue with standard sequence models (cannot extrapolate when $\mathcal{I} \neq \emptyset$) and the issue with forced G-invariant models (fail when $\mathcal{I} \subsetneq \{1, \ldots, m\}$). In Table 4 of the main text, we show the results for the best model out of all the permutation-invariant models in the column Best FF+G-inv.

