# OpenReview forum: "Neural Networks for Learning Counterfactual G-Invariances from Single Environments"
_ICLR.cc/2021/Conference — ICLR 2021 Poster_

### Official Review · AnonReviewer3 · 2020-10-28
**Seems like a good paper, although I found it hard to follow in places (updated)**

**Rating:** 7
**Confidence:** 3

**Review:**

POST REBUTTAL UPDATE: I am increasing my confidence in this paper from 2 to 3 - I still believe the paper can use some more clarity but enough points have been explained and updated in the draft for me to feel more confident in my evaluation.  I think the ideas in this paper are quite interesting - for this reason I continue to recommend acceptance.

Summary: This paper describes an approach to embedded invariances in learned neural networks through defining linear automorphism groups. They define a fair amount of theoretical machinery for this task, proposing a model of image generation which includes a number of transformations from these defined groups, and defining the notion of a counterfactual G-invariance for the task. They discuss a method for practically learning a useful invariance even if that exact invariance you wish to have is unknown, by ordering subspaces which may be invariant, and discusses the practicalities of embedding these into neural architectures. Some experiments are described in an MNIST setting and on simulated experiments, showing success at embedding these invariances in toy-ish settings.

Recommendation: I’m recommending acceptance for the paper, since the ideas seem interesting, there appears to  be theoretical contribution and empirical evidence, and it is obviously written with care. My hesitance comes on two fronts: I may be lacking some background in the relevant group theory/invariance literature, and it is hard for me to understand a number of important ideas due to the information density in the writing (a result of ICLR restrictions but also some fault of the authors).

Strong points:
-	The paper is very detailed and interesting – while I am not particularly familiar with the group theory side of the literature, it seems like a good idea from a robustness perspective and the authors lay out their ideas carefully
-	The notion of assuming an invariance unless contradicted is interesting at a high level, and provides some meat to the oft-discussed notion of “extrapolation from a single environment”
-	I appreciate the bridge made from the theory to the practical implementation
-	The experiments mostly back up the point the authors assert in their theory – larger experiments for a mostly theoretical paper like this are not necessarily required

Weak points:
-	My sense is the authors are having a lot of trouble fitting their ideas into the 8 pages. I sympathize but I also think they can do better on this front – the first two pages can be much more compact, with more space to explicate complex ideas that get swept over quickly
-	Ideas which do not receive enough attention or explanation (but should) include: Eq 2, Theorem 3, the design of neural network weights, and even the bare minimum of experimental details. I know the format is short but some of this stuff is necessary, and I believe the authors can do better in terms of fitting important information in. As it is I am confused about some central ideas, even after checking the appendix
-	The notion of “forbidding examples in the learner’s statistical model” carries some intuitive weight but is not precise – what is this model for a discriminative classifier? This can be more clearly explicated
-	It’s not clear how much is packed in the “economical data generation” assumption, or how that is really connected to the method. Please be more clear.
-	The definition of $T_{{U_D, U_I}}$ is not really clear – need another sentence on this indexing in the main body, as well as discussion of ordering!
-	Def 1: you don’t actually define what it means for 2 variables to be counterfactually coupled, you just show what a counterfactual coupling is. Please reword this definition.
-	Thm 3 – this is incredibly dense and I have a lot of trouble parsing this. I’m not sure how to interpret the direct sum – you’re combining all the subspaces which are supersets of M? Also not sure about the end – this is not G_j -invariant for j \in M-bar. What is M-bar? Is that a Reynolds operator? Then what does it mean for j to be an element of it?
-	How should I pick my groups G_1 … m? Not clear if this is important
-	Bottom of p6: not clear how these neuron weights are specified – does it matter which layer we are in? what does the product of B_m \omega_M,h mean – it looks like a subspace times a real number
-	Sec 5: I really am confused about the relationship between architecture and which invariances can be realizable. There’s not a lot of explanation on this.
-	Sec 6: it’s very hard to interpret anything in this section without the appendix – work more to make it stand alone
-	Proof of Thm 1: maybe I am missing some group theory background but I don’t understand this. For instance, there is no explanation of why this $\tilde{U}_I$ can always be found to couple X^cf with X^obs.
-	Proof of Lemma 1: Again, may be missing some background. But neither x nor $\bar{T}$ is mentioned in this proof, so I don’t know where it is going – please be more verbose.

Clarifications:
-	Middle of p3 – we can “compose rotations and image flips” – do you mean a union of the two sets of transformations? That’s what is shown in the notation but I may be missing some group theory background here
-	Top of p4: not clear how $G_D \cap G_I \neq 0$ is possible – is the idea that $G_i \cap G_j$ might be nonempty?
-	Below Eq 3: “the training data may contaion on a few samples of the variable” – do you mean only a few values of the variable may be observed?
-	Below Eq 3: you use the term environment without defining it, not sure how to interpret it in this context
-	Below Eq 7: should the samples of $\hat{Y}  \ X^{(obs)}$ be Y instead?
-	Below Eq 7: in (i) you say you don’t know the group – do you mean for which group Y should be considered invariant? If so, state that explicitly
-	It’s not clear how the statistical assumption at the end of Sec 3 really fits in with the argument, if you’re going to use it need more here
-	Top of p6 – you say the eigenspace of the Reynolds operator gives us a way to build an invariant NN, but the Lemma is about a linear operator. Need more description here
-	You say the method in Thm 3 is fast but the power set should grow exponentially – is that a problem?

Other feedback:
-	Not sure why you define g as going to the image of the probability – can’t it just be [0, 1]?
-	Not sure “unseen is forbidden” is quite right – wouldn’t it be “unseen is irrelevant” or something?
-	Specify whether Thm 1 only applies to linear automorphisms or if it is more general
-	The recursion at the bottom of Sec 4 is unreadable – just put this in the appendix
-	Bottom of p17 – describe more how the transformations are sampled. I shouldn’t have to work so hard to understand the experiments

---

> ### Author Response · Authors · 2020-11-17
> **Updated Manuscript (in blue) & Clarifications (Part 3/3)**
>
> Q14) “'the training data may contain only a few samples of the variable' – do you mean only a few values of the variable may be observed?"
>
> A14. We have clarified as follows: If the support of $U\_\\mathcal{I}$ is a singleton set $\\{c\\}$ for some constant $c$, then $(Y, X^\\text{(obs)})$ are said to be sampled using an economical data generation process. In other words, the training data can contain just one value for the variable $U\_\\mathcal{I}$ since the outputs $Y$ do not depend on $U\_\\mathcal{I}$.
>
> Q15) “Below Eq 3: you use the term environment without defining it, not sure how to interpret it in this context”
>
> A15. In our context, different environments correspond to different supports of $U\_\\mathcal{I}$. The economical data generation process with a singleton support for $U\_\\mathcal{I}$ results in a single environment. For example, if $Y$ is invariant to 90-degree rotations, then the set of upright images form one environment and the set of upside-down images can form another environment.
>
> Q16) “Below Eq 7: should the samples of $\\hat{Y} | X^{(obs)}$ be Y instead?”
>
> A16. This was a typo. We have updated the notation in Section 3: $Y | X^\\text{(tr)}$ refers to the distribution in training and $Y | X^\\text{(te)}$ refers to the distribution in test.
>
>
>
> Q17) “Below Eq 7 (new equation number (6)): in (i) you say you don’t know the group – do you mean for which group Y should be considered invariant?”
>
> A17. Yes, we do not know $\\mathcal{I}$ (or $\\mathcal{G}\_\\mathcal{I}$), thus we do not know which group $Y$ should be invariant to.
>
>
> Q18) “It’s not clear how the statistical assumption at the end of Sec 3 really fits in with the argument, if you’re going to use it need more here”
>
> A18. We have clarified below Definition 1 that $\\tilde{U}\_\\mathcal{I}$ can have very different support than $U\_\\mathcal{I}$. Since $\\tilde{U}\_\\mathcal{I}$, and thus $X^\\text{(cf)}$, are not observed, the statistical assumption at the end of Section 3 hinders learning of $\\Gamma\_\\text{true}$. We believe that this is clear to the reader in the updated manuscript.
>
>
>
> Q19) "Top of p6 – you say the eigenspace of the Reynolds operator gives us a way to build an invariant NN, but the Lemma is about a linear operator."
>
> A19. The Lemma describes a single neuron. If the first layer of a neural network is composed of these neurons and thus, is G-invariant, then we can use dense layers and non-linear activations on top of the first layer to obtain a G-invariant neural network.
>
>
> Q20) “You say the method in Thm 3 is fast but the power set should grow exponentially – is that a problem?”
>
> A20. Practically, this does not pose a problem for two reasons: **(1)** Theorem 3 will stop after finding $\\text{dim}(\\text{vec}(\\mathcal{X}))$ number of bases (which does not depend on the number of groups), so it need not iterate over the entire power set. However, it is unclear whether the worst-case exponential runtime can actually happen in practice. **(2)** Regardless, for a collection of groups, we would only need to compute the bases once and reuse them for all experiments.
> We have added this to the paper.
>
>
>
> Other feedback:
>
> Q21) "Not sure why you define g as going to the image of the probability – can’t it just be [0, 1]?"
>
> A21. $g$ returns a probability measure over the entire space of $Y$. In classification for example, $g$ returns the probabilities for all the classes.
>
>
>
> Q22) “Not sure ‘unseen is forbidden’ is quite right – wouldn’t it be ‘unseen is irrelevant’ or something?”
>
> A22. We have replaced the phrase with ‘unseen-is-unknown’ to represent the fact that examples not explicitly observed with infinitely many training examples have undefined/unknown outcomes in the learner’s model.
>
>
>
> Q23) “Specify whether Thm 1 only applies to linear automorphisms or if it is more general”
>
> A23. Theorems 1 & 2 also apply to groups with non-linear automorphisms. However, to be consistent with the rest of the paper, we only discuss linear automorphisms.
>
> Q24) “The recursion at the bottom of Sec 4 is unreadable – just put this in the appendix”
>
> A24. We have moved the differentiable approximation of the penalty to the Appendix and updated Section 4 with an intuition for the penalty instead.
>
> Q25) “Bottom of p17 – describe more how the transformations are sampled. I shouldn’t have to work so hard to understand the experiments”
>
> A25. We have updated Appendix F.1 further clarifying how the transformations are sampled to construct the training and test datasets.

---

> ### Author Response · Authors · 2020-11-17
> **Updated Manuscript (in blue) & Clarifications (Part 2/3)**
>
> Q6) “Thm 3 - this is incredibly dense ... I’m not sure how to interpret the direct sum – you’re combining all the subspaces which are supersets of M? What is M-bar?”
>
> A6. We have updated the text around Theorem 3 to describe the two goals of the Theorem: We wish to construct bases $\\mathcal{B}\_M$ for all $M \\subseteq \\{1,\\ldots,m\\}$ such that any weight vector $\\mathbf{w} \\in \\mathcal{B}\_M$ is **(a)** invariant to the groups $\\mathcal{G}\_i$ for $i\\in M$, and **(b)** not invariant to any group $j \\in \\{1,\\ldots,m\\}\\setminus M$}.
>
>  $\\mathcal{B}\_{\\supsetneq M}$ is computed as the direct sum of all the subspaces that correspond to the supersets of $M$. The reason is that we wish $\\mathcal{B}\_M$ to be invariant to groups indexed by $M$ but none of the groups outside $M$. Thus, we need to remove from from $\\tilde{\\mathcal{B}}\_M$ all weight vectors that are invariant to more groups in addition to those indexed by $M$ (i.e., supersets of $M$) . $\\bar{M}$ denotes the set complement $\\{1,\\ldots,m\\}\\setminus M$. Then, the theorem says that any nonzero vector $\\mathbf{w} \\in \\mathcal{B}\_M$ is invariant to all groups in $\\mathcal{G}\_i ~\\forall i\\in M$ but not invariant to $\\mathcal{G}\_j ~\\forall j\\in \\{1,\\ldots,m\\}\\setminus M$ . We have updated the notation to avoid confusion.
> We have added these clarifications to the paper and refer the reader to a pseudocode in the Appendix.
>
>
>
> Q7) “How should I pick my groups G\_1 … m? Not clear if this is important”
>
> A7. A practitioner can pick any collection of groups that they wish to be invariant to. Our method will choose invariance to the largest overgroup without harming training accuracy. However, it is important that the overgroup representing the true invariance of the task satisfies Theorem 2, otherwise the method will not extrapolate (as G-invariance does not imply CG-invariance).
>
>
>
> Q8) “Bottom of p6: not clear how these neuron weights are specified – does it matter which layer we are in? what does the product of B\_m \\omega\_M,h mean – it looks like a subspace times a real number”
>
> A8. Equation 10 only describes a single CG-invariant layer with $H$ neurons. Different architectures can be built using the neurons defined in Equation 10 for various types of input (architectures for images and sequences are described in Section 5). The parameter $\\omega\_{M\_i, h} \\in \\mathbb{R}^{d\_{M\_i} \\times 1}$ is a vector of dimension equal to the dimension of the subspace $\\mathcal{B}\_{M\_i}$, i.e., there is one scalar weight in $\\omega\_{M\_i, h}$ for every basis vector of $\\mathcal{B}\_{M\_i}$. The matrix-vector product $\\mathbf{B}\_{M\_i} \\omega\_{M\_i,h}$ simply gives a linear combination of these basis vectors (the basis vectors are the columns of matrix $\\mathbf{B}\_{M\_i}$). Any such linear combination is in the subspace $\\mathcal{B}\_{M\_i}$ and is invariant to $\\mathcal{G}\_{M\_i}$ (Lemma 2).
> We have clarified this before presenting Equation (10).
>
>
>
> Q9) “Sec 5..There’s not a lot of explanation on this.. “
>
> A9. We have streamlined Section 5 and focus more on the details of the architectures.
>
>
>
> Q10) “Sec 6... work more to make it stand alone”
>
> A10. We have added a running example in the results section (Section 6) to better clarify the experimental details.
>
>
>
> Q11) “Proofs of Theorem 1 and Lemma 1”
>
> A11. We have added more explanations in the proofs to help the reader less familiar with group theory.
>
>
>
> Clarifications:
>
> Q12) "We can 'compose rotations and image flips' – do you mean a union of the two sets of transformations?"
>
> A12. Join of two groups is the smallest set of transformations that contains the union of the groups and also satisfies the group properties. The join is not simply a union of the two groups: for example, join of $G\_\\text{rot}$ and $G\_\\text{flip}$ will also include transformations such as $T^{(90^\\circ)} \\circ T\_\\text{horiz-flip}$ which is not present in either of these groups. We use $\\langle \\cdot \\rangle$ to emphasize this distinction.
>
>
>
> Q13) “not clear how GD∩GI≠0 is possible – is the idea that Gi∩Gj might be nonempty?”
>
> A13. First, we have fixed a typo: $G\_D \\cap G\_I$ always has the identity transformation and is never empty. In general however, $G\_D \\cap G\_I$ can also have transformations other than the identity, for example when $G\_i \\cap G\_j$ is nonempty. While this can make the extrapolation task very hard to solve, the SCM is describing the most general scenario.

---

> > ### Comment · AnonReviewer3 · 2020-11-20
> > **Response**
> >
> > I have some confusion around Thm 3 still. In the paragraph previous you write "alas, we do not know G_I". But then, clearly an input to Thm 3 is the eigenspaces of the Reynolds operators of a group - is this a different group to G_I? What am I missing here?

---

> > > ### Author Response · Authors · 2020-11-20
> > > **Clarification**
> > >
> > > - We don't know the set $I$
> > > - Hence, we don't know $G_I$ because we don't know $I$
> > >
> > > Does this help clarify Theorem 3?
> > >
> > > We will clarify that in the next revision.

---

> > > > ### Comment · AnonReviewer3 · 2020-11-23
> > > > **Response**
> > > >
> > > > Yes I think this helps - it seems like there is a connection here between the set I and the learned ordering of the basis of invariance, would be good to draw that thread out a little for clarity's sake (if so).

---

> ### Author Response · Authors · 2020-11-17
> **Updated Manuscript (in blue) & Clarifications (Part 1/3)**
>
> We thank the reviewer for the constructive comments and helpful feedback. We would like to emphasize the timeliness of our contribution w.r.t. recent findings about DNNs extrapolations (learning spurious correlations and shortcut learning (see our Section 6, Arjovsky et al. (2019), D’Amour et al. (2020), de Haan et al. (2019) , Geirhos et al. (2020),McCoy et al. (2019), and Schölkopf (2019) among many others)), with a special emphasis in the large-scale study that came out this week from Google (D’Amour et al. (2020)).
>
>
>
>
>
> Q1) "Ideas which do not receive enough attention or explanation (but should) include: Eq 2, Theorem 3, the design of neural network weights, and even the bare minimum of experimental details."
>
> A1. We have updated the manuscript adding more details on these ideas. The text below Equation (2) discusses the indexing of $T\_{U\_\\mathcal{D}, U\_\\mathcal{I}}$. We have updated the text around Theorem 3 to clarify the objectives of the theorem and present a pseudocode in the Appendix. We have added clarifications before the neuron definition in Equation (10) and also clarified the intuitions behind the regularization penalty (Equation 12). In Section 6, we provide a running example to clarify the experimental details.
>
>
>
> Q2) “The notion of 'forbidding examples in the learner’s statistical model' carries some intuitive weight but is not precise – what is this model for a discriminative classifier? This can be more clearly explicated”
>
> A2. We have further clarified our meaning in the paper. Our theory gives it a precise mathematical meaning, that we have now emphasized in the updated manuscript. A better description is “unseen-is-unknown” rather than “unseen-is-forbidden”. Thanks for the suggestion.
>
>
>
> Q3) “It’s not clear how much is packed in the 'economical data generation' assumption, or how that is really connected to the method. Please be more clear.”
>
> A3. We do not need to assume it. The method works without economical data generation. Economical data generation gives an intuition of why one may get only one environment in the training data. We have made the notion more clear in the theory (that the environment $U\_\\mathcal{I}$ in training could be deterministic (a single value in the support) while in test, $\\tilde{U}\_\\mathcal{I}$ could have larger support).
>
>
>
>
> Q4) "The definition of TUD,UI is not really clear – need another sentence on this indexing in the main body, as well as discussion of ordering!"
>
> A4. We have added a paragraph describing the indexing in the main text. We interpret $U\_\\mathcal{D}$ and $U\_\\mathcal{I}$ as the random seeds of a random number generator that generate ordered sequences of transformations from $\\mathcal{G}\_\\mathcal{D}$ and $\\mathcal{G}\_\\mathcal{I}$ respectively. If these ordered sequences are, say, $T^{(1)}\_\\mathcal{D}, \\ldots, T^{(a)}\_\\mathcal{D}$ and $T^{(1)}\_\\mathcal{I}, \\ldots, T^{(b)}\_\\mathcal{I}$, then $T\_{U\_\\mathcal{D},U\_\\mathcal{I}}$ is the transformation obtained after interleaving the two sequences of transformations and composing them in order: $T\_{U\_\\mathcal{D}, U\_\\mathcal{I}} = T^{(1)}\_\\mathcal{I} \\circ T^{(1)}\_\\mathcal{D} \\circ T^{(2)}\_\\mathcal{I} \\circ \\ldots$. Note that $T^{(i)}\_\\mathcal{I}$ or $T^{(i)}\_\\mathcal{D}$ could be identity transformations. Appendix B.1 shows that this indexing is surjective, i.e., it can index every transformation in $\\mathcal{G}\_{\\mathcal{D}\\cup \\mathcal{I}}$.
>
>
>
> Q5) "Def 1: you don’t actually define what it means for 2 variables to be counterfactually coupled, you just show what a counterfactual coupling is. Please reword this definition."
>
> A5. We have added a standard example of two coupled random variables (dice rolls) before presenting Definition 1. We have reworded the definition to explicitly describe the coupling of $X^\\text{(obs)}$ and $X^\\text{(cf)}$. A visual interpretation is shown in the SCM graph of Figure 1.

---

> > ### Comment · AnonReviewer3 · 2020-11-20
> > **Response**
> >
> > A3 - Okay - it might be good to rephrase this bit then, to make it sound like less of an assumption and more of an intuition-building exercise
> >
> > A5 - I think I understand counterfactual coupling now but I'm not sure that the example helps much. For instance, in the example, there are four variables (D1, D2, and their respective modified versions) and I don't quite see how it maps onto X_cf and X_obs in defn 1. Also, it's not clear to me why the marginal distributions must remain the same.
> >
> > Proof of Lemma 1 is more understandable now, thanks.

---

> > > ### Author Response · Authors · 2020-11-20
> > > **Thanks (further clarifications)**
> > >
> > > Your comments were really helpful. Thanks!
> > >
> > > A3: Thanks. That is a good point, we will rephrase it in the next version to make it clear it is not an assumption.
> > >
> > > A5: Would it help if we had called dice D as U (hidden variable)? Then the die are D'1 = (U + 2) mod 6 + 1 and D'2 = (U + 1) mod 6 + 1 ?
> > >
> > > - It may be more clear if we think of X(obs) = X(train) and X(cf) = X(test) (we describe like this later in the paper) where (train) means sampled from the training distribution (and test means sampled from test distribution).
> > > - The test has a different environment than training. So, for each test example, there is a hypothetical training-distribution example generated with the same U_D, U_Y, U_u but using U_I (train environment) rather than \tilde{U}_I (test environment)
> > > - The marginals must remain the same since we want the X(train) and X(test) to be sampled by their respective train and test distributions.
> > > - We avoided saying "train" and "test" (opting for "obs" and "cf") in the definition to avoid confusion about the test data, since we don't observe it during training. At training time, the test data is just a counterfactual variable (a hypothetical).

---

> > > > ### Comment · AnonReviewer3 · 2020-11-23
> > > > **Response**
> > > >
> > > > I think the U notation is better, yes.
> > > >
> > > > Is the marginal that remains the same the marginal over U? Or is it the marginal of D'1 must equal the Marginal of D'2? In the paper it sounds like you're saying the latter, but the former seems more reasonable to me. If the latter, then I'm not sure this should be part of the definition but rather an extra assumption you're making (although I'm still not sure why you need it). If the former then that makes sense, just needs clarification.

---

> > > > > ### Author Response · Authors · 2020-11-25
> > > > > **Clarified the example in the paper**
> > > > >
> > > > > Yes, the marginal over $U$ remains the same. Although the marginals of $D_1^\dagger$ and $D_2^\dagger$ turn out to be same in the example, this is not an assumption in general: $X^\text{(obs)}$ and $X^\text{(cf)}$ can have different marginal distributions. We have modified the example in the paper to avoid confusion: now $D_1^\dagger$ has the marginal of a 6-sided die roll whereas $D_2^\dagger$ has the marginal of a 12-sided die roll.

---

### Official Review · AnonReviewer2 · 2020-10-29
**Great idea, but presentation has room for improvement**

**Rating:** 7
**Confidence:** 2

**Review:**

Summary:
A method is given for training neural networks in the presence of a group of transformations, such that the network weights are invariant with respect to any transformation on the inputs which doesn't contradict the training data. Experiments on MNIST and toy sequence data are used to verify that this training method leads to improved extrapolation of predictions to unseen environments.

Strengths:
The method introduced seems promising, as it doesn't require training data to exhibit symmetry, but can still verify (or reject) the invariance of data with respect to a collection of candidate symmetry groups. The training method also seems quite lightweight, and shouldn't require significant additional resources to check for the presence or absence of symmetry.

Although the experiments are a bit limited, the authors include a detailed experiments section in the appendix with a larger selection of baselines and more information about choosing the magnitude of the applied CG-regularization.

Critiques:
The explanation of the results has a lot of room for improvement, and I would recommend the authors revise the writing to follow standard best practices, such as defining/explaining new variables when they are introduced, giving the steps associated with novel algorithms, etc. I give a few specific examples below where this lack of clarity makes the authors' results hard to understand, but there are many other examples of this not listed.

The description of the underlying causal model in section 3 (from the end of page 3 to the start of page 5) is hard to follow owing to a lack of explanation in many places. For example, the overgroups $G_D$ and $G_I$ are introduced without any insight into the distinction between these groups, or what role they play in the context of extrapolation tasks.

Similarly, the central concept of CG-invariance lacks some crucial details in its definition (Def 2), making the following material harder to follow. In particular, it isn't stated if CG-invariance is defined relative to a specific $\tilde{U}_I$, or else requires Eq. 6 to hold for any choice of $\tilde{U}_I$ (the latent distribution which determines the counterfactual samples $X^{cf}$).

Theorem 3 is difficult to make sense of, with the subspaces $B_M$ appearing at first glance to be circularly defined (the projection in Eq. 9 used to define the $B_M$ is itself defined in terms of these subspaces). The text below and above Theorem 3 helps to interpret this circularity as an inductive algorithm for calculating these subspaces, but it would have been much clearer to define this algorithm explicitly in terms of pseudocode (along with a runtime), and then reference this definition in Theorem 3.

Recommendation:
Although the techniques seem like a timely and useful contribution, the poor presentation makes these techniques difficult to follow, and limits the usefulness of the paper for readers. I'm recommending a weak accept, but this can be improved by clarifying the presentation and making the results easier for readers to understand.


**UPDATE AFTER THE REBUTTAL:** The new material in the paper clarifies things quite a bit, especially the intuitive explanations appearing below Equation 2 and at the bottom of page 4. Thank you for adding that, I have changed my score accordingly :)

---

> ### Author Response · Authors · 2020-11-17
> **Updated Manuscript (in blue) & Clarifications**
>
> We thank the reviewer for the constructive comments and helpful feedback. We would like to emphasize the timeliness of our contribution w.r.t. recent findings about DNNs extrapolations (learning spurious correlations and shortcut learning (see our Section 6, Arjovsky et al. (2019), D’Amour et al. (2020), de Haan et al. (2019) , Geirhos et al. (2020),McCoy et al. (2019), and Schölkopf (2019) among many others)), with a special emphasis in the large-scale study that came out this week from Google (D’Amour et al. (2020)).
>
> Q1) “The explanation of the results has a lot of room for improvement, and I would recommend the authors revise the writing to follow standard best practices, such as defining/explaining new variables when they are introduced, giving the steps associated with novel algorithms, etc. I give a few specific examples below where this lack of clarity makes the authors' results hard to understand, but there are many other examples of this not listed.”
>
> A1. We have provided additional explanations throughout the paper. We have streamlined Section 3 and added the graph of the SCM (Figure 1). In Section 4, we have updated the text around Theorem 3 to clarify the objectives of the theorem and present a pseudocode in the Appendix. We have also clarified the intuitions behind the regularization penalty (Equation 12) and show an example computation of the penalty in Figure 6 of the Supplementary Material. We have updated Section 5 with more details about the architectures. In Section 6, we provide a running example to clarify the experimental details.
>
>
>
> Q2) “The description of the underlying causal model in section 3 (from the end of page 3 to the start of page 5) is hard to follow owing to a lack of explanation in many places. For example, the overgroups GD and GI are introduced without any insight into the distinction between these groups, or what role they play in the context of extrapolation tasks.”
>
> A2. We have updated this section making it easier to follow and have included a graph of the structural causal model in Figure 1. We have clarified the role of groups $\\mathcal{G}\_D$ and $\\mathcal{G}\_I$: the target variable is defined as being invariant to the group $\\mathcal{G}\_I$ but dependent on the group $\\mathcal{G}\_D$.
>
>
>
> Q3) “Similarly, the central concept of CG-invariance lacks some crucial details in its definition (Def 2), making the following material harder to follow. In particular, it isn't stated if CG-invariance is defined relative to a specific U\~I, or else requires Eq. 6 to hold for any choice of U\~I (the latent distribution which determines the counterfactual samples Xcf).”
>
> A3. We have updated Definition 1 (Counterfactual coupling) to make it clear that $X^\\text{(cf)}$ is obtained by counterfactually replacing $U\_\\mathcal{I}$ by any choice of $\\tilde{U}\_\\mathcal{I}$ in the SCM equations. Then Definition 2 (CG-invariant representations) is tied to that choice of $\\tilde{U}\_\\mathcal{I}$ from Definition 1.
>
>
>
> Q4) Theorem 3 is difficult to make sense of, with the subspaces BM appearing at first glance to be circularly defined (the projection in Eq. 9 used to define the BM is itself defined in terms of these subspaces). The text below and above Theorem 3 helps to interpret this circularity as an inductive algorithm for calculating these subspaces, but it would have been much clearer to define this algorithm explicitly in terms of pseudocode (along with a runtime), and then reference this definition in Theorem 3.
>
> A4. We have updated the text around Theorem 3 to describe the two goals of the Theorem: We wish to construct bases $\\mathcal{B}\_M$ for all $M \\subseteq \\{1,\\ldots,m\\}$ such that any weight vector $\\mathbf{w} \\in \\mathcal{B}\_M$ is **(a)** invariant to the groups $\\mathcal{G}\_i$ for $i\\in M$, and **(b)** not invariant to any group $j \\in \\{1,\\ldots,m\\}\\setminus M$}.
> $\\tilde{\\mathcal{B}}\_M$ contains all the vectors $\\mathbf{w}$ that are invariant to $\\mathcal{G}\_M$ but could also contain vectors that are invariant to some overgroup of $\\mathcal{G}\_M$. Thus, each step of our inductive method performs a Gram-Schmidt orthogonalization in order to satisfy condition **(b)** above: we need to remove from $\\tilde{\\mathcal{B}}\_M$ all weight vectors that are invariant to more groups in addition to those indexed by $M$ (i.e., supersets of $M$). We also refer the reader to a pseudocode in Appendix C.

---

### Official Review · AnonReviewer4 · 2020-10-29
**[Official Review]: A nice idea being somewhat oversold; weak experimental evaluation**

**Rating:** 5
**Confidence:** 4

**Review:**

This paper proposes an interesting and potentially quite impactful and valuable idea, which I believe is novel.
The idea is: instead of specifying invariances by hand in the architecture of a network, we can instead specify a set of possible invariances, and regularize the model to favor more invariance.
The authors describe how to structure and regularize a DNN in this way, and provide proof-of-concept experiments.
The experiments show that the proposed method outperforms networks that are fully-invariant or non-invariant when the true data is partially-invariant.

Unfortunately, there are a number of weaknesses which lead me to recommend against acceptance.  In no particular order:
1) The crucial "unseen is forbidden" hypothesis is vague and seems to be a bit of a strawman.
2) The framing of the paper seems to oversell the method in a way that makes the contribution less clear.
3) The writing is not very clear.
4) The experiments seem to be only proof-of-concept in scenarios where the method is designed to work.
5) The method seems to incur an exponential cost, but this is not discussed.

Elaborating:
1) The authors claim that, because DNN behavior is undefined on unseen datapoints, the "unseen-is-forbidden learning hypothesis is currently preventing neural networks from assuming symmetric extrapolations without evidence."  This claim is stated in various forms several times, but never made very precise, and it is crucial in motivating the authors' approach.  Roughly, I take the authors to be claiming that (i) the correct way to "extrapolate" is to assume that: transformations that were not observed to change the target distribution should be assumed to NOT change the target distribution, (ii) DNNs will not extrapolate in this way by default, and must be explicitly designed to do so.
These claims (or whatever the authors actually mean) need(s) to be stated explicitly, and with appropriate modesty.  After all, both (i) and (ii) seem contentious.
The claim about an "economical data generating process" supports (i), but is itself somewhat vague and dubious, and should be discussed in the introduction as motivation for (i).

2) The authors claim that their method can discover invariances without any data supporting them.  And their abstract claims: "Any invariance to transformation groups is mandatory even without evidence, unless the learner deems it inconsistent with the training data."  But in reality, the authors specify a small number of possible invariances which the method selects among (in a soft way).  And the data is used to guide this selection process.  So in reality, the designer is in charge of specifying a (restricted) set of (possible) invariances.  So like previous works on enforcing invariances, it places a  burden on the designer to identify plausible invariances.  Overall, I found the framing in the work to be "the model discovers invariances by itself without any data!" whereas a more neutral version would be "instead of enforcing a set of invariances, we propose a set of *possible* invariances, and assume that any input transformations that are not observed to affect the label should be enforced"

3) Besides the above issues (vagueness of "unseen-is-forbidden" and related discussion (1), overselling (2)), there were several other issues of clarity.  The paper is not poorly written overall, but is much harder to read and understand than it needs to be.  Some specific issues are:
- The results in Section 4 are presented with insufficient context or intuition.  Theorems are stated without any proof intuition and should reference proofs in the appendix.  The intuition for the penalty arrived at (eqn13) is unclear.
- The flow is sometimes unclear.  For instance, "Learning CG-invariant representations without knowledge of G_I. " should be a subsection, not a (latex) paragraph, and should explain what the point of the subsection is before diving in.  The authors seem to be using (latex) paragraphs (i.e. beginning with bolded phrases) as subsections and paragraphs beginning with italicized phrases as (latex) paragraphs.  I suspect the paper was edited to fit into 8 pages without removing sufficient content.  This impedes the flow and sacrifices clarity.
- I think a graph showing the data generating process would be much clearer than the current explanations (e.g. eqn4/5)
- it is unclear what equation 7 is saying... the text above makes it seem like a definition of a goal, but the following paragraph treats it as an assertion that the goal is possible to achieve.
...Overall, I recommend stripping out some of the mathematical details and using more words and diagrams in the main text to describe the underlying issues/motivations/methods.
The overall story should be made clearer (e.g. by addressing (1) and (2)), and more space should be devoted to linking each part of the paper into the overall story.

4) The experiments are synthetic tasks where the correct invariance group is included in the set of invariances being searched over.  I don't think that showing that this method can bring some benefits on a real task is an absolute requirement, given the novelty of the approach.  But without more meaningful results, the paper is held to a much higher standard.  Even for synthetic experiments, these are rather weak; for instance, it would be interesting to see whether/how the method degrades when we consider much larger sets of possible invariances.

5) It seems like the method might require including a set of parameters for each of the possible 2^m invariances.  Is this in fact the case?  If not, why not?  If so, it should be discussed as a limitation.


-------------------------------
Suggestions/Questions:
- In Section 4 paragraph 1, are G-invariance and G_I-invariance used interchangeably?  This was confusing.
- say what I and D are as soon as they are introduced (top of page 4).
- Typo: "a somewhat a"
- Why a "nonpolynomial" activation function?
- The definition of "almost surely" at the bottom of page 4 is not correct (it is possible to sample probability 0 events), and also it should say that samples of Gamma(X^(obs)/(cf)) (not X^(obs)/(cf)) are equal with probability 1 (these are not the same statement!).
- "level of invariance" and "non-extrapolated validation accuracy", and several other phrases are not defined and should probably be replaced by something more clear and explicit.
- It seems like you might need to assume that that different x^(hid) can't be used to generate the same x^(obs) or x^(cf).  If so, this should be explicit.

---

> ### Author Response · Authors · 2020-11-17
> **Title: Unintended interpretation, Updated Manuscript (in blue) & Clarifications (Part 3/3)**
>
>
> Suggestions/Questions:
> We have updated the paper to incorporate the reviewer’s suggestions, adding clarifications wherever necessary.
>
> Q6) “In Section 4 paragraph 1, are G-invariance and G\_I-invariance used interchangeably? This was confusing.”
>
> A6. We have replaced the occurrences of $\\mathcal{G}\_\\mathcal{I}$-invariance with G-invariances in Section 4 para 1.
>
>
>
> Q7) “say what I and D are as soon as they are introduced (top of page 4).”
>
> A7. We have clarified that the target variable will be invariant to the groups indexed by $\\mathcal{I}$ but dependent on the groups indexed by $\\mathcal{D}$.
>
>
>
> Q8) “Why a ‘nonpolynomial' activation function?”
>
> A8. The theorems that prove universal approximation power of DNNs argue that activations should be nonpolynomial (Leshno (1993)), since DNNs with nonlinear polynomial activations are not expressive enough.
>
> Reference: Leshno, Moshe, et al. "Multilayer feedforward networks with a nonpolynomial activation function can approximate any function." *Neural networks* 6.6 (1993): 861-867.
>
>
>
> Q9) “The definition of ‘almost surely’ at the bottom of page 4 is not correct”
>
> A9. We have changed the sentence to: ‘the equality is true for any sample of $X^\\text{(cf)}$ and $X^\\text{(obs)}$ except for a set of measure zero.’
>
>
>
> Q10) "'level of invariance' and 'non-extrapolated validation accuracy', and several other phrases are not defined and should probably be replaced by something more clear and explicit."
>
> A10) We define the level of invariance of a subspace $\\mathcal{B}\_M$ as the size of $M \\subseteq \\{1,\\ldots,m\\}$, i.e., the number of groups that any $\\mathbf{w} \\in \\mathcal{B}\_M$ is invariant to. We have removed the phrase "non-extrapolated validation accuracy"; we now refer to it as validation accuracy on held-out training data (sampled with no knowledge of the extrapolation task).
>
>
>
> Q11) “It seems like you might need to assume that that different x^(hid) can't be used to generate the same x^(obs) or x^(cf). If so, this should be explicit.”
>
> A11. We do not need this assumption. For example, consider two different $X^\\text{(hid)}$ corresponding to the canonical forms of the digits 6 and 9 respectively. If we transform one of them by a 90-degree rotation and the other with no rotation, then we get the same $X^\\text{(obs)}$ for both. The task cannot be perfectly solved, but this is fine as the SCM is describing the most general scenario.

---

> ### Author Response · Authors · 2020-11-17
> **Unintended interpretation, Updated Manuscript (in blue) & Clarifications (Part 2/3)**
>
> Q3) (i). “The results in Section 4 are presented with insufficient context or intuition… Theorems are stated without any proof intuition… intuition for the penalty arrived at (eqn13) is unclear."
>
> A3. (i) We have added intuitive explanations for both Theorem 3 and the regularization penalty.  We show an example computation of the penalty in Figure 6 of the Supplementary Material. We have also added main ideas of the proofs for all the theorems (the proofs of the lemmas use group-theoretic properties and are more algebraic in nature) and refer the reader to the proofs in the Appendix.
>
>
>
> Q3) (ii) “The flow is sometimes unclear. For instance, 'Learning CG-invariant representations without knowledge of G\_I.' should be a subsection, not a (latex) paragraph, and should explain what the point of the subsection is before diving in.”
>
> A3. (ii) We have divided Sections 3 and 4 into subsections to improve readability.
>
>
>
> Q3) (iii) “Graph showing the structural casual model...”
>
> A3. (iii) We have added a graph of the structural causal model in Figure 1.
>
>
>
> Q3) (iv) "it is unclear what equation 7 is saying... the text above makes it seem like a definition of a goal, but the following paragraph treats it as an assertion that the goal is possible to achieve."
>
> A3. (iv) We have replaced the equation with two equations (now numbered (5) and (6)) to clarify our statement: If $g\_\\text{true}$ and $\\Gamma\_\\text{true}$ approximate the training distribution $Y | X^\\text{(tr)}$, and $\\Gamma\_\\text{true}$ is CG-invariant, then the model extrapolates to the test (counterfactual) distribution $Y | X^\\text{(te)}$.
>
>
>
> Q4) “Even for synthetic experiments, these are rather weak; for instance, it would be interesting to see whether/how the method degrades when we consider much larger sets of possible invariances.”
>
> A4. For sequence tasks, we consider 45 pairwise permutation groups. The method is able to learn invariance to groups generated by any subset of these groups. These experiments indicate that the performance does not degrade even with larger sets of possible groups.
>
> For images, the only other linear automorphism group we can think of is cyclic translation (translation that wraps around the edges). However, this group is **(a)** not something that happens with natural images, **(b)** leads to the violation of normal subgroup property required in Theorem 2 (for G-invariance to imply CG-invariance). In fact, proof of Theorem 1 uses the cyclic translation group and rotation group to show that G-invariance does not always imply CG-invariance. Note that image translations (non-cyclic) for fixed-size images are not invertible and thus do not form a group.
>
>
>
> Q5) “It seems like the method might require including a set of parameters for each of the possible 2^m invariances. Is this in fact the case? If not, why not? If so, it should be discussed as a limitation.”
>
> A5. This is not the case. The number of parameters of the new neuron is the same as the old neuron as the method in Theorem 3 returns $d\_\\mathcal{X} = \\text{dim}(\\text{vec}(\\mathcal{X}))$ basis vectors across all subspaces. The rest of the subspaces are zero subspaces. There is nearly no computational penalty during the optimization. We have added this clarification in the last paragraph of Section 4.1 and have updated Equation (10) to emphasize that there are only $B \\leq d\_\\mathcal{X}$ nonzero subspaces.

---

> ### Author Response · Authors · 2020-11-17
> **Unintended interpretation, Updated Manuscript (in blue) & Clarifications (Part 1/3)**
>
> We thank the reviewer for the constructive comments and helpful feedback. We believe the reviewer gave an unintended interpretation of our claims, arguing we overstated our contributions. We feel we did not, and we are here to clarify.
>
> Q1) “(i) the correct way to 'extrapolate' is to assume that: transformations that were not observed to change the target distribution should be assumed to NOT change the target distribution (ii) DNNs will not extrapolate in this way by default, and must be explicitly designed to do so. These claims (i) and (ii) need to be stated explicitly, and with appropriate modesty. After all, both (i) and (ii) seem contentious.“
>
> A1. It is unfair to say we “overstate the contributions or created strawman arguments”, when it all looks like an unintended interpretation.
>
> When we say “This unseen-is-forbidden learning hypothesis is currently preventing neural networks from assuming symmetric extrapolations without evidence” we want to convey that, if one does not actively teach a DNN the symmetry, there is no guarantee the DNN will simply spontaneously learn it (in our experiments, they do not. We never found one example where the DNN was able without being forced to). Right above, we write “the prediction $P(Y^\\text{(te)}=C|X^\\text{(te)} = (B,A))$ is undefined, since $P(X^\\text{(tr)} = (B,A)) = 0$.” That means, the DNN is free to have any output for $(B,A)$, since the optimization is not pushing any specific outcome for the output of $(B,A)$. There is nothing contentious. We will change the language to be more clear.
>
> Regarding extrapolations, there is growing evidence that DNNs have extrapolation shortcomings, such as “spurious correlations” and “shortcut learning” (see Arjovsky et al. (2019), de Haan et al. (2019) , Geirhos et al. (2020),McCoy et al. (2019), and Schölkopf (2019) among many other works (see updated paper)). This week, a wide-scale study at Google (D’Amour et al. (2020)) concludes “that structural failure modes [of DNNs], [... are] a misalignment between the predictor learned by empirical risk minimization and the causal structure of the desired predictor (Schölkopf, 2019; Arjovsky et al., 2019), [...] being a key failure mode for machine learning models [in deployment].” Underspecification Presents Challenges for Credibility in Modern Machine Learning, https://arxiv.org/abs/2011.03395. Currently, the highest-score paper at ICLR https://openreview.net/forum?id=UH-cmocLJC is a paper about extrapolations in graph neural networks. Clearly, not everybody in the community believes these are issues, and certainly not an issue in all applications. But there is mounting evidence that extrapolation is one of the biggest challenges ahead. We believe our work is an important contribution in understanding how we can include causal structure in DNNs.
>
> We do not claim to have the “only correct way to extrapolate”. In fact, there is really no “correct way to extrapolate”, since extrapolations are tied to a causal model. Different causal models will give different ways to extrapolate. Rather, we show a clear case of single-environment extrapolations tied to group transformations, the first of its kind.
>
> Our experimental setting is not “designed to work with our method”, rather, it is designed to validate the theory and method, since a method that cannot perform our task will clearly fail and one that can clearly succeeds.
>
>
>
> Q2) "The framing is the model discovers invariances by itself without any data!"
>
> A2. There is confusion again interpreting our claims. When we say "Any invariance to transformation groups is mandatory even without evidence, unless the learner deems it inconsistent with the training data." we mean “Any invariance to (known) transformation groups is mandatory…”. In causality, there is provably no way to infer counterfactuals without a clear causal model (Pearl 2009). For us, the causal model must describe the group transformations that can affect the data. We have clarified this to avoid incorrect interpretations.
>
> We like the reviewer’s proposed sentence "instead of enforcing a set of invariances, we propose a set of *possible* invariances, and assume that any input transformations that are not observed to affect the label should be enforced", but the challenge is the potential *incorrect* interpretation “that the user must propose all overgroup invariances”, which is not required. We changed all sentences to make clear that the groups are *known*.

---

> ### Comment · AnonReviewer4 · 2020-11-22
> **[Official Reviewer]: Good intentions don't compensate for poor presentation**
>
> What matters here is how your work comes across to the reader, not your intentions.
> It is not "unfair" of me to point out flaws in the presentation.
> I also never said the claims are overstated, I said they are *oversold*.
>
> Now, I'll address the content of your response, and outline my remaining confusions and concerns.
>
> 1/2/4) Overall, I remain unsatisfied with the framing of this work.
> When you say "the prediction is undefined", this is false; DNNs define a function over their entire input domain.  The idea that a DNN "is free to have any output" for an unseen example is also false, because DNNs have limited capacity.  I think these claims need to be rethought, not rephrased.  I'm familiar with the lines of work that you mention, but I don't consider them conclusive evidence against DNNs being able to generalize appropriately, when given sufficient (and sufficiently diverse) data.
> Of course, there can be generalization issues related to causality.  You said: "For us, the causal model must describe the group transformations that can affect the data."  This seems like another way of saying what I said: that the designer must specify possible invariances.  I think my framing is more general.  After all, you can apply this technique when the groups are unknown using a best guess.  I'd like to see experiments of that nature, or at least experiments that show how the method performs when the possible invariances specified by the designer are not well-matched to the actual invariances in the data.  Having the set of possible invariances be known *a priori* seems like a strong assumption, which could easily be violated.  **Do you agree?  If not, why not?**  Note that my question is about the general applicability of this method, not about specific types of data, such as images, or sequences of tokens.
>
> Can you please:
> * Explain and justify the causal assumptions you make?
> * Explain how you use them to derive which group transformations can affect the data?
> * Discuss the significance of these assumptions and derivations as contributions?
>
> 3) I'm not satisfied with the updates.  I believe the following questions should be answered in the main text:
> * Why are these results interesting?
> * What role do they play in your work?
> Not addressing these questions is an example of what I mean by "insufficient context".
>
> 5) The construction is still unclear to me, and the updates didn't help me much.
> Equation 10 looks like the standard equation for computing the activation of a neuron, except that it constructs the weight using this sum over $B$ elements.  So it seems like this method might require a factor of $B$ more parameters than a standard neural network, and I am not sure why you say: "The number of parameters of the new neuron is the same as the old neuron".  Can you provide a clear, simple, and detailed example of how this construction works?
>
>
>
>
> Regarding the caption for the new Figure 1: you make it sound like only P(X) changes, but can't P(Y|X) change as well?

---

> > ### Comment · AnonReviewer4 · 2020-11-22
> > **Clarity of paper worse than I remembered**
> >
> > I spent a long time reading this paper during my initial review.  As I mentioned, I find the paper much harder to read and understand than it needs to be.  Revisiting it now, it is worse than I remember.  I think the main issue is that the technical aspects are not clearly explained using words.
> >
> > I recommend the authors spend more effort on clarifying the structure of the work, and how all of the pieces fit together.  I feel like readers will need to struggle unnecessarily to understand this work.  While I think the authors have made some improvements to the presentation, I think it's unlikely I will be able to recommend acceptance without significant improvements in clarity.
> >
> > The other reviewers also noted issues with clarity and even though they have raised their scores, they have low confidence.  This leads me to believe that none of us have understood the paper thoroughly (although maybe I am wrong; I have only skimmed the other reviews), and I don't think that is our fault.  My high confidence is not because I understand the paper thoroughly, but because I am confident that the clarity is insufficient.

---

> > > ### Author Response · Authors · 2020-11-25
> > > **Summary of updates to paper since initial submission**
> > >
> > > Since our initial submission, we have provided additional explanations throughout the paper.
> > >
> > > We have streamlined Section 3 and added the graph of the SCM (Figure 1). We have provided an example of coupling before Definition 1 (Counterfactual coupling) and reworded the definition to improve clarity. The paragraph added after Definition 2 clearly describes the train and the test data.
> > >
> > >
> > >
> > > Every result in Section 4 is presented along with the role it plays (in initial submission as well), for instance:
> > >
> > > - Before Theorem 1: “Theorem 1 below shows that CG-invariances (Definition 2) are stronger than G-invariances. After that, Theorem 2 defines conditions under which G-invariances suffice as CG-invariances.”
> > > - After Lemma 2: "The above property of the Reynolds operator can be leveraged to build neural networks that adhere to particular group symmetries, as done by Yarotsky (2018) and van der Pol et al. (2020). If we knew $\mathcal{G}_\mathcal{I}$, restricting the parameters of each neuron to the left 1-eigenspace of the Reynolds operator of $\mathcal{G}_\mathcal{I}$ would give us a way to build a $\mathcal{G}_\mathcal{I}$-invariant neural network.”
> > > - Before Theorem 3: “Alas, we do not know $\mathcal{I}$, and consequently we do not know $\mathcal{G}_I$. Instead, we want to construct bases for the complete space such that they are partially ordered by their invariance strength: From most invariant bases to least.”
> > >
> > > We present a pseudocode for Theorem 3 in Appendix D. We have now added a detailed step-by-step example showing the construction of a CG-invariant neuron in Appendix C.  It also includes visualization of the subspaces found by Theorem 3 for the same example (Figure 5).
> > >
> > > Below Equation (12), we have also clarified the intuitions behind the regularization penalty and show an example computation of the penalty in Figure 8 (Appendix F).
> > >
> > > Finally, we have updated Section 6 with a running example to clarify the experimental details. Due to space constraints, complete details of the experiments have been deferred to Appendix G.

---

> > ### Author Response · Authors · 2020-11-25
> > **Clarifications (Part 3/3)**
> >
> > Q6) "I'm not satisfied with the updates. I believe the following questions should be answered in the main text: Why are these results interesting?"
> >
> > A6)
> > - Single-environment extrapolation is a significant step towards bringing counterfactual learning to representation learning.
> > - Theorems 1 & 2 are extremely important, showing that counterfactual extrapolations are not the same as G-invariant representations.
> > - Existing works on G-invariances consider only a couple of linear automorphism groups with hand-selected bases (since it is difficult to hand-design bases for multiple groups). We show how to automatically create bases for all overgroups of a set of groups.
> >
> >
> > Q7) "What role do they play in your work? Not addressing these questions is an example of what I mean by 'insufficient context'".
> >
> > A7) Lemma 1 and 2 allow us to construct neural networks strictly adhering to a given transformation group. Paragraph after Lemma 2 in the paper clarifies this:
> >
> > “The above property of the Reynolds operator can be leveraged to build neural networks that adhere to particular group symmetries, as done by Yarotsky (2018) and van der Pol et al. (2020). If we knew $\\mathcal{G}\_\\mathcal{I}$, restricting the parameters of each neuron to the left 1-eigenspace of the Reynolds operator of $\\mathcal{G}\_\\mathcal{I}$ would give us a way to build a $\\mathcal{G}\_\\mathcal{I}$-invariant neural network.”
> >
> > However, we do not know $\\mathcal{G}\_\\mathcal{I}$. Thus, with the help of the eigenspaces found in Lemma 2, the role of Theorem 3 is to construct subspaces with different invariances (i.e., for all $M \\subseteq \\{1,\\ldots,m\\}$) partially ordered by their invariance strength. Paragraph before Theorem 3 in the paper clarifies this:
> >
> > “Alas, we do not know $\\mathcal{I}$, and consequently we do not know $\\mathcal{G}\_\\mathcal{I}$. Instead, we want to construct bases for the complete space such that they are partially ordered by their invariance strength: From most invariant bases to least. In other words, we construct bases for subspaces $\\mathcal{B}\_M$ for $M \\subseteq \\{1,\\ldots,m\\}$ such that any weight vector $\\mathbf{w} \\in \\mathcal{B}\_M$ is **(a)** invariant to the groups $\mathcal{G}_i$ for $i\in M$, and **(b)** not invariant to any group $j \\in \\{1,\\ldots,m\\}\\setminus M$.
> >
> > Later, we will use this partial order to define a regularization term for our method.”
> >
> >
> >
> >
> > Q8) "The construction is still unclear to me, and the updates didn't help me much. Equation 10 looks like the standard equation for computing the activation of a neuron, except that it constructs the weight using this sum over B elements. So it seems like this method might require a factor of B more parameters than a standard neural network, and I am not sure why you say: "The number of parameters of the new neuron is the same as the old neuron". Can you provide a clear, simple, and detailed example of how this construction works?"
> >
> > A7) Let the input $x \in \mathbb{R}^3$. Since the subspaces found in Theorem 3 are all orthogonal to each other, there are only 3 basis vectors for the space $\text{vec}(\mathcal{X}) = \mathbb{R}^3$. For our neuron, there is one parameter for each of these basis vectors and a bias parameter, thus a total of 4 parameters. In a standard neuron, there are 4 parameters as well.
> >
> > **In Appendix C, we have provided a detailed step-by-step example describing the construction of the neuron. We also visualize the subspaces of Theorem 3 for the same example in Figure 5.**
> >
> >
> >
> > Q9) "Regarding the caption for the new Figure 1: you make it sound like only P(X) changes, but can't P(Y|X) change as well?"
> >
> > A9) Our framework follows the independent casual mechanism principle (Schölkopf et al., 2012, Peters et al., 2017, Schölkopf, 2019): a mechanism describing a variable given its causes is independent of all other mechanisms describing other variables.
> >
> > In our context, the mechanism to generate the data changes from $X^\\text{(obs)} | X^\\text{(hid)}$ to $X^\\text{(cf)} | X^\\text{(hid)}$. However, the mechanism $Y | X^\\text{(hid)}$, representing the underlying task, is not influenced by this change and remains the same. We have clarified this in the paper.
> >
> >
> > Schölkopf, Bernhard, et al. "On causal and anticausal learning." *arXiv preprint arXiv:1206.6471* (2012).
> >
> > J. Peters, D. Janzing, and B. Schölkopf. “Elements of Causal Inference - Foundations and Learning Algorithms”. MIT Press (2017).
> >
> > Schölkopf, Bernhard. "Causality for machine learning." *arXiv preprint arXiv:1911.10500* (2019).

---

> > ### Author Response · Authors · 2020-11-25
> > **Clarifications (Part 2/3)**
> >
> > Q3) "Can you please: Explain and justify the causal assumptions you make?"
> >
> > A3) Assumptions:
> >
> > - We assume the training and test data *may be* transformed by a set of linear automorphism groups.
> > - Maybe the training and test data are not transformed by any of them.
> > - Maybe the training and test data are transformed by all of them.
> > - Maybe the training and test data are transformed by a subset of them.
> > - We do not know which transformations or transformation groups have been applied to the training data or which ones will be applied to the (unobserved) test data.
> > - Maybe the training data has a different set of transformations than the test data. In this case, we want our classifier to be invariant to them, if possible.
> > - In some cases this is impossible with the tools we use (Theorem 1), but if it is possible* (Theorem 2), we introduce a method to do it.
> > - *Theorem 2: gives a condition that is sufficient but we do not know if it is necessary.
> >
> >
> >
> > Q4) "Explain how you use them to derive which group transformations can affect the data?"
> >
> > A4) The automorphism groups are not derived from the training data. Specifically, because the training data may not contain any transformations from the groups that we may need to be invariant to. Hence, the researcher needs to provide which transformation groups must be considered by the method. Moreover, our method is not tied to any specific transformation group. Anyone is free to essentially use any set of linear automorphism groups with our method.
> >
> >
> >
> > Q5) "Discuss the significance of these assumptions and derivations as contributions?"
> >
> > A5) Existing work in environment-invariant classifiers either forces the invariances or assumes the training data contains them. Bernhard Schölkopf’s work (and co-authors) has a number of examples in these two categories (we cite the ones most relevant to our task), including (Locatello et al., 2019) which uses causality describe the shortcomings of disentanglement methods (After observing x, we can construct infinitely many generative models which have the same marginal distribution of x), and one assuming group invariances in the causal model (Besserve et al., 2018). Single environment invariances, however, have never been addressed. It requires a rather complex set of techniques. However, as we show, it is possible.

---

> > ### Author Response · Authors · 2020-11-25
> > **Clarifications (Part 1/3)**
> >
> > Q1) "Overall, I remain unsatisfied with the framing of this work.
> > When you say 'the prediction is undefined', this is false; DNNs define a function over their entire input domain.  The idea that a DNN 'is free to have any output' for an unseen example is also false, because DNNs have limited capacity.  I think these claims need to be rethought, not rephrased. I'm familiar with the lines of work that you mention, but I don't consider them conclusive evidence against DNNs being able to generalize appropriately, when given sufficient (and sufficiently diverse) data."
> >
> >
> > A1)
> >
> > - We understand the concern and have changed “undefined” to “underspecified”, which is a technical term that should clarify the reviewer’s concern.
> > - Our argument is not controversial. It is the same argument made in domain adaptation (covariate shift adaptation, to be more exact) (Ben-David et al., 2006; Ben-David et al., 2010; Glorot et al., 2011; Long et al., 2015). We also give specific examples related to symmetries. The difference between covariate shift adaptation and environment-invariant work (as described in our Related Work) is that covariate shift adaptation is a data-driven approach, not a structural causal one (hence, it needs the shifted test data $X^\\text{(te)}$ for the adaptation). Zero-shot learning is similar to covariate shift adaptation, but through an observable proxy variable.
> > - It would be controversial, however, if we equated counterfactual learning with domain adaptation and zero-shot learning, since these approaches rely on data and adapt to a specific covariate shift. Hence, we tried to avoid the parallel. Counterfactual models do not need test data because the potential shift is described in the structural causal equations, such that the resulting counterfactual model would adapt to any such potential shift.
> >
> > References:
> >
> > - Ben-David, Shai, John Blitzer, Koby Crammer, and Fernando Pereira. "Analysis of representations for domain adaptation." Advances in neural information processing systems 19 (2006): 137-144.
> > - Ben-David, Shai, John Blitzer, Koby Crammer, Alex Kulesza, Fernando Pereira, and Jennifer Wortman Vaughan. "A theory of learning from different domains." Machine learning 79, no. 1-2 (2010): 151-175.
> > - Glorot, Xavier, Antoine Bordes, and Yoshua Bengio. "Domain adaptation for large-scale sentiment classification: A deep learning approach." In ICML. 2011.
> > - Long, Mingsheng, Yue Cao, Jianmin Wang, and Michael Jordan. "Learning transferable features with deep adaptation networks." In International conference on machine learning, pp. 97-105. PMLR, 2015.
> >
> >
> >
> > Q2) "Of course, there can be generalization issues related to causality. You said: "For us, the causal model must describe the group transformations that can affect the data." This seems like another way of saying what I said: that the designer must specify possible invariances. I think my framing is more general.  After all, you can apply this technique when the groups are unknown using a best guess.  I'd like to see experiments of that nature, or at least experiments that show how the method performs when the possible invariances specified by the designer are not well-matched to the actual invariances in the data. Having the set of possible invariances be known a priori seems like a strong assumption, which could easily be violated. Do you agree? If not, why not? Note that my question is about the general applicability of this method, not about specific types of data, such as images, or sequences of tokens."
> >
> > A2) **“Do you agree? If not, why not?”** We will politely disagree, because something in this case is better than nothing.
> > First, we clarify that the transformations need not be present in the training data. Then, if the set of possible transformations specified by the designer only contains a subset of the transformations that are seen during test time, using our method is still better than not using it, since it will extrapolate at least to a subset of the test examples that use the transformations given in the causal model. Clearly, the examples with transformations that are never known to the method and applied during test time will be missed (it is unclear to us how one could adapt to something that it has never seen in the training data and was never described to the model).
> >
> > We can add experiments with missing transformations in a group if the reviewer feels these are necessary, but they will essentially show what we are describing above. No surprises.

---

### Author Response · Authors · 2021-03-18
**Updates to the final version**

1. We have modified the title to emphasize the fact that our approach tackles *counterfactual* extrapolations via G-invariances.
2. We have updated the notation for the counterfactual variable from $X^\text{(cf)}$ to $X_{U_\mathcal{I}\leftarrow\tilde{U}_\mathcal{I}}$ in order to emphasize that it is constructed by counterfactually replacing $U_\mathcal{I}$ with $\tilde{U}_\mathcal{I}$ in the data generation process of the observed variable $X$.
3. We have added a paragraph in the main text (and a section in the Supplementary Material) discussing a limitation of the proposed regularization (Equation (12)).
4. We have made the code publicly available at: https://github.com/PurdueMINDS/NN_CGInvariance

---

### Decision · Program_Chairs · 2021-01-07
**Final Decision**

**Decision:**

Accept (Poster)

**Comment:**

Pros:
- All reviewers agreed that the idea was particularly interesting/novel. I personally appreciated the perspective of unlearning invariances that prove inconsistent with the training data, rather than learning invariances that are demonstrated by the training data.
- The authors significantly improved clarity during the rebuttal period, and two out of three reviewers raised scores or confidence as a result.

Cons:
- There were significant concerns raised by reviewers about clarity of presentation, and some concern around whether the specific instantiation of the high level idea was the most sensible. From a *lightweight* reading of the paper on my part, I also feel that the writing style is unnecessarily dense, though I believe the underlying ideas are solid.
- One of the reviewers (AnonReviewer4) continues to have serious concerns. I believe the authors and AnonReviewer4 may have both become more entrenched in their positions during the discussion, in a way that wasn't particularly productive.

This paper is borderline score-wise. I believe it is particularly important to reward and encourage unusually novel work. Primarily for this reason I bias my decision upwards, and recommend acceptance.

nit: belive --> believe